



# Subglacial upwelling in winter/spring increases under-ice primary production

Tobias Reiner Vonnahme[1], Emma Persson[1], Ulrike Dietrich[1], Eva Hejdukova[2], Christine Dybwad[1], Josef Elster[3], Melissa Chierici[4,5], Rolf Gradinger[1]

[1] Department of Arctic and Marine Biology, UiT – The Arctic University of Norway, Tromsø, Norway
[2] Department of Ecology, Faculty of Science, Charles University, Prague, Czech Republic
[3] University of South Bohemia, České Budějovice, and Institute of Botany ASCR, Třeboň, Czech Republic
[4] Institute of Marine Research, Tromsø, Norway
[5] University Centre in Svalbard (UNIS), Longyearbyen, Svalbard, Norway

*Correspondence to*: Tobias R. Vonnahme (Tobias.Vonnahme@uit.no)

**Abstract.** Subglacial upwelling of nutrient rich bottom water is known to support high summer primary production in Arctic fjord systems. However, during the winter/spring season, the importance of subglacial upwelling has not been considered yet. We hypothesized that subglacial upwelling under sea ice is present in winter/spring and sufficient to increase phytoplankton primary productivity. We evaluated the effects of the subglacial upwelling on primary production in a seasonally fast ice covered Svalbard fjord (Billefjorden) influenced by a tidewater outlet glacier in April/May 2019. We found clear evidence for subglacial upwelling. Although the estimated entrainment factor (1.6) and total fluxes were lower than in summer studies, we observed substantial impact on the fjord ecosystem and primary production. The subglacial meltwater leads to a salinity stratified surface layer and sea ice formation with low bulk salinity and permeability. The combination of the stratified surface layer, a two-fold higher under-ice irradiance, and higher N and Si concentrations at the glacier front supported two orders of magnitude higher primary production (42.6 mg C m$^{-2}$ d$^{-1}$) compared to a marine reference site at the fast ice edge. The nutrient supply increased primary production by approximately 30 %. The brackish water sea ice at the glacier front with its low bulk salinity contained a reduced brine volume, limiting the inhabitable place and nutrient exchange with the underlying seawater compared to full marine sea ice. Microbial and algal communities were substantially different in subglacial influenced water and sea ice compared to the marine reference site, sharing taxa with the subglacial outflow water. We suggest that with climate change, the retreat of tidewater glaciers could lead to decreased under-ice phytoplankton primary production, while sea ice algae production and biomass may become increasingly important.



## 1 Introduction

Tidewater glacier fronts have recently been recognized as hotspots for marine production including top trophic levels, such as marine mammals, birds and piscivorous fish (Lydersen et al., 2014, Meire et al., 2016), but also primary producers (Meire et al., 2016; Hopwood et al., 2020). During summer, large amounts of freshwater are released below the glacier and entrap nutrient rich bottom water, sediments and zooplankton during the rise to the surface (Meire et al., 2016, Moon et al., 2018). Together with katabatic winds pushing the surface water out of the fjords, it creates a strong upwelling effect (Meire et al., 2016). The biological response to this upwelling will depend on the characteristics of the upwelling water. The high sediment loads of the plumes absorb light and thereby inhibit primary production close to the glacier front. The light absorbing effect of the plumes is highly dependent on the glacial bedrock (Halbach et al., 2019). However, the high nutrient concentrations supplied to the surface increase summer primary production at some distance from the initial upwelling event, once the sediments settled out (Meire et al., 2016, Halbach et al., 2019). These tidewater upwelling effects have been described in a variety of different Arctic fjords including deep glacier termini in western Greenland (Meire et al., 2016), eastern Greenland (Cape et al., 2019), and north-western Greenland (Kanna et al., 2018), but also in shallower fjords on Svalbard (Halbach et al., 2019). Studies of the effect of potential subglacial upwelling in winter/spring on sea ice and pelagic primary productivity are lacking due to the perception of the absence of freshwater outflows in winter.

Tidewater glacier related upwelling mechanisms can also be caused by the melting of deep icebergs (Moon et al., 2018), or the melting of the glacier terminus in contact with warm seawater (Moon et al., 2018, Sutherland et al., 2019). A seasonal study within an East Greenland fjord showed high melt rates of icebergs throughout the year (Moon et al., 2018), while subglacial runoff had been detected as early as April, but with substantially higher freshwater inputs in summer. Glacier terminus melt rates are low compared to the subglacial outflow but can be present throughout the year (Chandler et al., 2013, Moon et al., 2018). In fact, Moon et al. (2018) found higher terminus melt rates below 200 m in winter than in summer, which may allow winter upwelling. Svalbard glaciers are typically shallower and deep terminus melt (below 200 m) and iceberg induced upwelling are less important (Dowdeswell, 1989). However, subglacial outflows can persist throughout winter and specifically in early spring through the release of subglacial meltwater stored from the previous melt season (Hodgkins, 1997), or through constant supply from groundwater, temperate parts of the glacier, geothermal heat, or frictional dissipation (Schoof et al., 2014). While studies on upwelling in winter and spring are limited to oceanographic observations, the biological effects on e.g. primary production have been neglected (Chandler et al., 2013, Moon et al., 2018). Even low rates of subglacial outflows can be sufficient to supply nutrients to the surface, while at the same time entrapping considerably less light absorbing sediments compared to the summer situation. We suggest, that in the absence of wind induced mixing due to the seasonal presence of a fast ice cover, this spring upwelling mechanism could be the primary mechanism to significantly increase primary production, especially towards the end of the ice algal/phytoplankton spring bloom when nutrients become limiting (Leu et al. 2015). With climate change, these dynamics are expected to change substantially (e.g. Błaszczyk et al., 2009, Holmes et al.,





2019). Higher glacial melt rates and earlier runoffs may initially increase tidewater glacier induced upwelling. However, their retreat and transformation into first shallower tidewater glacier termini with less pronounced upwelling and finally into land terminated glaciers will eliminate the upwelling process – thus reducing the primary production.

Due to high inputs of freshwater in the autumn preceding the onset of sea ice formation, tidewater glacier influenced fjords are
often sea ice covered in spring, mainly by coastal fast ice. Within the sea ice, ice algae start growing, once sufficient light is penetrating the snow and ice layers with the onset varying between March and April, depending on latitude and local ice conditions (Leu et al., 2015). While the beginning of the ice algal blooms is typically related to light, the magnitude depends on the initial nutrient concentration and nutrient additions from the water column into the brine channel network (Gradinger, 2009). Thus, subglacial upwelling has the strong potential to extend the duration and increase the magnitude of the ice algal
blooms. Similar control mechanisms apply to phytoplankton bloom formation and duration. Under-ice phytoplankton blooms are thought to be light limited if the ice is snow covered and substantial blooms have been described in areas with lack of snow cover (e.g. melt ponds, after rain events, Fortier et al., 2002, Arrigo et al., 2014) or at the ice edge related to ice edge induced upwelling (Mundy et al., 2009). On Svalbard, low precipitation rates and strong katabatic winds (Esau & Repina, 2012) often limit snow coverage also on the fast ice near glacier fronts (Braaten, 1997), potentially allowing enough light for under-ice
phytoplankton blooms to occur. Once sufficient light reaches the water column, typically a diatom dominated bloom starts along the receding ice edge or even below the sea ice (e.g. Hodal et al., 2012; Lowry et al., 2017). Once silicate becomes limiting for diatom growth, other taxa like *Phaeocystis pouchetii* dominate the next stage of the seasonal succession (von Quillfeldt, 2000). This succession pattern can be significantly influenced by tidewater glacier related spring upwelling. Sea ice formed from brackish water has relatively low bulk salinity, low brine volume and low total ice algal biomass as observed e.g.
in the Baltic Sea (Haecky & Andersson, 1999). Brackish ice conditions with low algal biomass will reduce light absorption allowing more light to reach the water column to potentially fuel under-ice phytoplankton blooms. We suggest that higher nutrient levels supplied via slow subglacial upwelling in the absence of wind mixing may enhance algal growth and cause different succession patterns for phytoplankton and sea ice algae.

We used the natural conditions in a Svalbard fjord as a model system contrasting the biological response at two glacier fronts with different freshwater inputs during the winter/spring transition period while a fast ice cover was present. The aim of the study was to investigate the effect of the glacier terminus, and subglacial outflow related upwelling on winter/spring primary productivity and algae community structures both in and under the sea ice. We hypothesized that; 1) subglacial upwelling throughout winter and spring supplies nutrient rich meltwater and bottom water to the surface, 2) subglacial upwelling
increases primary production near the glacier front, 3) biomass of sea ice algae is lower at glacier fronts as a result of low permeability sea ice.



## 2 Methods

### 2.1 Field work and physical properties

Fieldwork was conducted on Svalbard in Billefjorden (Fig. 1) between 22nd of April and 5th of May 2019, when most samples were collected. For comparison, some samples had been already taken in April 2018 (subglacial outflow water for DNA analyses) and July 2018 (glacier ice and supraglacial runoff). Billefjorden is fed by a few streams, rivers and the tidewater glacier Nordenskiöldbreen and partly fast ice covered from January to June. Tidal currents are very slow with under 0.1 cm s$^{-1}$, which translates to advection below 22 m per tidal cycle (Kowalik et al., 2015). Katabatic winds can be strong due to several

glaciers and valleys leading into the fjord system (Láska et al., 2012). Together with low precipitation, this leads to a thin snow depth on the sea ice. Bare sea ice spots are often present in the sea ice season (personal observations). The fjord is separated from Isfjorden, a larger fjord connected to the West Spitsbergen current, by a shallow sill making Billefjorden an Arctic fjord with limited impacts of Atlantic water inflows. This character is shown in water masses, circulation patterns and animal communities including the presence of polar cod (Maes, 2017, Skogseth et al., 2020).

Samples were taken at three stations 1) at the fast ice edge (IE) – a full marine reference station (78°39'09N, 16°34'01E); 2) at the southern site of the ocean terminated glacier terminus (SG) (approx. 20 m water depth) with freshwater outflow observed during the sampling period (78°39'03N, 16°56'44E) and; 3) at the northern site of the glacier terminus (NG) with no clear freshwater outflow observed and a mostly land-terminating glacier front (78°39'40N, 16°56'19E).

Snow depth and sea ice thickness around the sampling area were measured with a ruler. Sea ice and glacier ice samples were

taken with a Mark II ice corer with an inner diameter of 9 cm (Kovacs Enterprise, Roseburg, OR, USA). Temperature of each ice core was measured immediately by inserting a temperature probe (TD20, VWR, Radnor, PA, USA) into 3 mm thick pre-drilled holes. For further measurements the ice cores were sectioned into the following sections: 0–3 cm, 3–10 cm and thereafter in 20 cm long pieces from the bottom to the top, packed in sterile bags (Whirl-Pak™, Madison, WI, USA) and left to melt at about 4–15 ˚C for about 24–48 h in the dark. Sections for chlorophyll *a* (Chl) measurements, DNA extractions, and

algae and bacteria counts were melted in 50 % vol/vol sterile filtered (0.2 µm Sterivex filter, Sigma-Aldrich, St. Louis, MO, USA) seawater, while no seawater was added to the sections for salinity and nutrient measurements. Salinity was measured immediately after melting using a conductivity sensor (YSI Pro 30, YSI, USA). Brine salinity and brine volume fractions were calculated after Cox et al. (1983) for sea ice temperatures below -2 °C and after Leppäranta and Manninen (1988) for sea ice temperatures above.

Samples of under-ice water were taken using a pooter (Southwood and Henderson, 2000) connected to a hand-held vacuum pump (PFL050010, Scientific & Chemical Supplies Ltd., UK). Deeper water at 1 m, 15 m, 25 m depths and bottom water at IE station were taken with a water sampler (Ruttner sampler, 2 L capacity, Hydro-Bios, Germany). Glacial outflow water was sampled in April 2018 close to SG station using sterile Whirl-Pak™ bags. No outflow water was found around NG station. Cryoconite hole water (avoiding any sediment) was sampled in July 2018 with a pooter on sites known to differ in their

biogeochemical settings (Nordenskiöldbreen main cryoconite site (NC), and Nordenskiöldbreen near Retrettøya (NR) sites





characterized by Vonnahme et al., 2016). One m thick glacier surface ice samples were taken with the Mark II ice corer at the southern side of the glacier on the NC site.

CTD profiles were taken at each station by a CastAway™ (SonTek/-Xylem, San Diego, CA, USA). At the SG station an additional CTD profile was taken with a SAIV CTD SD208 (SAIV, Lakselv, Norway) including turbidity and fluorescence
sensors. Unfortunately, readings at the other stations failed due to sensor freezing at low air temperatures. Surface light data were obtained from the photosynthetic active radiation (PAR) sensor of the ASW 1 weather station in Petuniabukta (23 m a.s.l), operated by the University of South Bohemia (Láska et al., 2012; Ambrožová and Láska, 2017).

During the sampling days, Billefjorden and Adventdalen were overcast. The light regime under the ice was calculated after Masicotte et al. (2018) with a snow albedo of 0.78, a snow attenuation coefficient of 15 m$^{-1}$ (Mundy et al., 2005), ice attenuation
coefficients of 5.6 m$^{-1}$ for the upper 15 cm and 0.6 m$^{-1}$ below (Perovich et al., 1998). For sea ice algae, an absorption coefficient of 0.0025 m$^2$ mg$^{-1}$ Chl was used. The fraction of fjord water vs subglacial meltwater for the water samples was calculated assuming linear mixing of the two salinities (Meltwater = 0 PSU, seawater=34.6 PSU), since no other water masses in regard to temperature or salinity signature were present (Table 1).

## 2.2 Chemical properties

Nutrient samples of water and melted sea ice and glacier ice were sterile filtered as described above, stored in acid washed (rinsed in 5 % vol/vol HCl) and MQ rinsed 50 ml falcon tubes and kept at -20 °C until processing. Total alkalinity (TA), Dissolved inorganic carbon (DIC), and pH samples were sampled in 500 ml borosilicate glass bottles avoiding air contamination and fixed within 24 h with 2 % (fin. con.) HgCl$_2$ and stored at 4 °C until processing.

Nutrients were measured in triplicates using standard colorimetric methods with a nutrient autoanalyser (QuAAtro 39, SEAL
Analytical, Germany) using the instrument protocols: Q-068-05 Rev. 12 for nitrate (detection limit = 0.02 µmol L$^{-1}$), Q-068-05 Rev. 12 for nitrite (detection limit = 0.02 µmol L$^{-1}$), Q-066-05 Rev. 5 for silicate (detection limit = 0.07 µmol L$^{-1}$), and Q-064-05 Rev. 8 for phosphate (detection limit = 0.01 µmol L$^{-1}$). The data were analysed using the software AACE v5.48.3 (SEAL Analytical, Germany). Reference seawater (Ocean Scientific International Ltd., United Kingdom) was used as blanks for calibrating the nutrient analyser. The maximum differences between the measured triplicates were 0.1 µmol L$^{-1}$ for silicate
and nitrate and 0.05 µmol L$^{-1}$ for nitrite and phosphate. Concentrations of nitrate and nitrite (NO$_X$) were used to estimate the fraction of bottom water reaching the surface at SG assuming linear mixing of bottom water (at station IE) and surface water concentration using the NO$_X$ concentration measured at IE (Table 1).

DIC and TA were analyzed within 6 months after sampling as described by Jones et al. (2019) and Dickson et al. (2007). DIC was measured on a Versatile Instrument for the Determination of Titration carbonate (VINDTA 3C, Marianda, Germany),
following acidification, gas extraction, coulometric titration, and photometry. TA was measured with potentiometric titration in a closed cell on VINDTA Versatile INstrument for the Determination of Titration Alkalinity, VINDTA 3S, Marianda, Germany). Precision and accuracy was ensured via measurements of Certified Reference Materials (CRM, obtained from





Dickson, Scripps Institution of Oceanography, USA). Triplicate analyses on CRM samples showed mean standard deviations below ±1 µmol kg$^{-1}$ for DIC and AT.

## 2.3 Biomass and communities

For determination of algal pigment concentrations about 500 ml sea water or melted sea ice were filtered onto GF/F filter (Whatman plc, Maidstone, UK) in triplicates using a vacuum pump (max 200 mbar vacuum) before storing the filter in the dark at -20 °C. Water and melted sea ice for DNA samples were filtered onto Sterivex filter (0.2 µm pore size) using a peristaltic pump and stored at -20 °C until extraction. Algae were sampled in two ways; 1) a phytoplankton net (10 µm mesh size) was pulled up from 25 m and the samples fixed in 2 % (final conc.) neutral Lugol and stored at 4 °C in brown borosilicate glass bottles before processing; and 2) water or melted sea ice was fixed and stored directly as described above. For later bacteria abundance estimation, 25 ml of water was fixed with 2 % (final con.) formaldehyde for 24–48 h at 4 °C before filtering onto 0.2 µm polycarbonate filters (Isopore™, Merck, US) and washing with filtered seawater and 100 % ethanol before freezing at -20 °C.

Algal pigments (Chl, phaeophytin) were extracted in 5 ml 96 % ethanol at 4 °C for 24 h in the dark. The extracts were measured on a Turner Trilogy AU-10 fluorometer (Turner Designs, 2019) before and after acidification with a drop of 5 % HCl. 96 % ethanol was used as a blank and the fluorometer was calibrated using a chlorophyll standard (Sigma S6144). For estimations of algae derived carbon a conversion factor of 30 g C (g Chl)$^{-1}$ was applied (Cloern et al., 1995). The maximum differences (max-min) between the measured triplicates were under 0.05 µg Chl L$^{-1}$ unless stated otherwise.

DNA was isolated from the Sterivex filter cut out of the cartridge using sterile pliers and scalpels, using the DNeasy® PowerSoil® Kit following the kit instructions with a few modifications. Solution C1 was replaced with 600 µL Phenol:Chloroform:Isoamyl Alcohol 25:24:1 and washing with C2 and C3 was replaced with two washing steps using 850 µL chloroform. Before the last centrifugation step, the column was incubated at 55 °C for 5 min to increase the yield. For microbial community composition analysis, we amplified the V4 region of a ca. 292 bp fragment of the 16S rRNA gene using the primers (515F, GTGCCAGCMGCCGCGGTAA and 806R, GGACTACHVGGGTWTCTAAT, assessed by Parada et al., 2016). For eukaryotic community composition analyses, we amplified the V7 region of ca 100-110 bp fragments of the 18S rRNA gene using the primers (Forward 5'-TTTGTCTGSTTAATTSCG-3' and Reverse 5'-GCAATAACAGGTCTGTG-3', assessed by Guardiola et al., 2015). The Illumina MiSeq PE library was prepared after Wangensteen et al. (2018).

For qualitative counting of algal communities, phytoplankton net hauls and bottom sea-ice samples were counted under an inverted microscope (Zeiss Primovert, Carl Zeiss AG, Germany) with 10x40 magnification. For quantitative counts, 10-50 ml of the fixed water samples were settled in an Utermöhl chamber (Utermöhl, 1958) and counted. Algae were identified using identification literature by Tomas (1997), and Throndsen et al. (2007). For bacteria abundance estimates, bacteria on polycarbonate filter samples were stained with DAPI (4,6-diamidino-2-phenylindole) as described by Porter and Feig (1980), incubating the filter in 30 µl DAPI (1 µg ml$^{-1}$) for 5 min in the dark before washing with MQ and ethanol and embedding in Citifluor:Vectashield (4:1) onto a microscopic slide. The stained bacteria were counted using an epifluorescence microscope





(Leica DM LB2, Leica Microsystems, Germany) under UV light at 10x100 magnification. At least 10 grids or 200 cells were counted. The community structure of the phytoplankton net haul was used for estimating the contribution of sea ice algae to the settling community based on typical Arctic phytoplankton (Von Quillfeldt, 2000) and sea ice algal species (von Quillfeldt et al., 2003) described in literature.

**2.4 In situ measurements and incubations**

Vertical algal pigment fluxes were measured using custom made (Faculty of Science, Charles University, Prague, Czech Republic) short-term sediment traps (6.2 cm inner diameter, 44.5 cm height) at 1 m, 15 m, and 25 m under the sea ice anchored to the ice at SG and IE, as described by Wiedmann et al. (2016). Sediment traps were left for 24 h at the SG station and 37 h at the IE station. After recovery, samples for algal pigments were taken, fixed and analysed as described above.

Primary production (PP) was measured based on $^{14}$C-DIC incorporation. Samples were incubated *in situ* in 100 ml polyethylene bottles attached to the rig of the sediment trap giving identical incubation times. Seawater or bottom sea ice melted in filtered seawater (ca 20 °C initial temperature to ensure fast ice melt) on site were incubated with $^{14}$C sodium bicarbonate at final concentration of 1 µCi ml$^{-1}$ (PerkinElmer Inc., Waltham, USA). PP samples were incubated in triplicates for each treatment with two dark controls for the same times as the sediment traps. Samples were filtered onto precombusted Whatman GF/F filters (max 200 mbar vacuum) and acidified with a drop of 37 % fuming HCl for 24 h for removing remaining inorganic carbon. The samples were measured in Ultima Gold™ Scintillation cocktail on a liquid scintillation counter (PerkinElmer Inc., Waltham, USA, Tri-Carb 2900TR) and PP was calculated after Parsons et al., (1984). Dark carbon fixation (DCF) rates were used to estimate bacterial biomass production using a conversion factor of 190 mol POC (mol $CO_2$)$^{-1}$ fixed (Molari et al., 2013).

A reciprocal transplant experiment was conducted in water from 1 m and 15 m depth under the sea ice to test for fertilizing effects of glacial front water at stations SG and IE. At each site, incubations were made where half of the initial water volume was replaced with sterile filtered (0.2 µm) seawater of either the same station or the other station, excluding physical effects (light, temperature, sediment load). These samples were incubated and processed together with the other PP incubations at the adequate depths as described above.

**2.5 Statistics and bioinformatics**

Silicate, phosphate and $NO_X$ concentrations were plotted against salinities and correlation were tested via linear regression analysis using the lm function in R (R Core Team, Vienna, Austria). P values were corrected for multiple testing using the false discovery rate. Since the primary production estimates of the reciprocal transplant experiments were not normally distributed, came from a nested design, and had heterogeneous variance, a robust nested Analysis of variance (ANOVA) was performed to test for significant treatment effects of incubation water with water depth as nested variable. The map was created in R using the PlotSvalbard v0.9.2 package (Vihtakari, 2020). The Svalbard basemap was retrieved from the Norwegian Polar





institute (2020, CC BY 4.0 license), the pan-Arctic map was retrieved from Natural Earth (2020, CC Public domain license), and the bathymetric map was retrieved from the Norwegian mapping authority (Kartverket, 2020, CC BY 4.0 license).

16S sequences were analysed using a pipeline modified after Atienza et al. (2020) based on OBITools v1.01.22 (Boyer et al.,
2014). The raw reads were demultiplexed and trimmed to a median phred quality score minimum of 40 and sequence lengths between 215bp and 299bp (16S rRNA) or between 90 and 150bp (18S rRNA) and merged. Chimaeras were removed using uchime with a minimum score of 0.9. The remaining merged sequences were clustered using swarm (Mahe et al., 2014). 16S swarms were classified using the RDP classifier (Wang et al., 2007) and 18S swarms using the sina aligner (Pruesse et al., 2012) with the silva SSU 138.1 database (Quast et al., 2012). Further multivariate analyses were done in R using the vegan
package. The non-metric multidimensional scaling (NMDS) plots are based on Bray-Curtis dissimilarities and were used to visualize differences between groups (brackish water at SG – Fjord water, sea ice – seawater). Analysis of Similarities (ANOSIM) were done to test for differences of the communities between the groups (999 permutations, Bray-Curtis dissimilarities).

## 3 Results

### 3.1 Physical parameters

The physical conditions of sea ice (temperature T/bulk salinity S) and surface water (uppermost 4 m under the sea ice, T and S) at the freshwater inflow impacted site SG differed substantially from NG and IE. The sea ice and the upper 4 m under the sea ice were having consistently lower salinities (<8 PSU) and higher temperatures (-0.4 ˚C to -0.2 ˚C) at SG compared to NG and IE and also compared to the deeper water masses at SG (salinity > 34.6 PSU, temperature < -1.4 ˚C)(Fig. 2c,d). Sea ice
melt was unlikely because the measured water temperatures and sea ice temperatures were below freezing point considering the sea ice bulk salinity. The water column at SG was highly stratified with a low salinity 4 m thick layer under the sea ice, separated by a sharp ca 1 m thick pycnocline (Fig. 2c,d). In contrast, the water column at IE was fully mixed and at NG only a minor salinity drop from 34.6 to 33.6 PSU occurred within the the upper 50 cm under the sea ice (Fig. 2c,d). Sea ice temperature and salinity showed similar variations between the three sites with SG ice having lower salinities and higher
temperatures relative to sea ice at the other stations (Fig. 2a,b). At SG, bulk salinities were mostly below 0.7 PSU and calculated brine salinities below 14 PSU, except for the uppermost 40 cm where bulk salinities reached around 1.5 PSU and a brine salinity of 32 PSU (Fig. 2). This resulted in very low brine volume fractions below 5 %, except for the lowermost 10 cm with brine volume fractions up to 9 % (Supplementary table S1). At IE and NG, bulk salinities are mostly above 5 PSU (>40 PSU brine salinity) and temperatures were below -0.4 ˚C, which led to brine volume fractions above 6 % in all samples and above
10 % in the bottom 30 cm.

The homogenous temperature and salinity water column profiles at IE and NG stations indicate the presence of only one water mass (Local Arctic water, Skogseth et al., 2020). The only additional water mass was subglacial meltwater (salinity of 0 PSU) mixed into the surface layer of SG. Applying a simple mixing model based on the two salinities (IE= 34.6 PSU, Glacier= 0



PSU) provide an estimation of the fraction of glacially derived water in the surface layer of ca. 85 % in the uppermost 2 m
under the sea ice, before decreasing to 0 % at 4 m under the sea ice below the strong halocline. The water sample taken 1 m
under the sea ice had a fraction of 32 % glacial meltwater (Table 1). For NG, glacial derived water contributed only 3 % in the
first 50 cm under the sea ice.

The SG station was 33 m deep and about 180 m away from the glacier front. The sea ice was 1.33 m thick and covered by 3
cm of snow. The ice appeared clear with some minor sediment and air bubble inclusions and missed a skeletal bottom layer.
In the water column, a higher potential sediment load was observed as a turbidity peak at the halocline (Fig. 3). Direct evidence
of subglacial outflow had been observed at the southern site of the glacier in form of icing and liquid water flowing onto the
sea ice in April 2018, April 2019 and October 2019, but this form of subglacial outflow froze before reaching the fjord, which
was additionally blocked by sea ice. The sea ice temperature was between -0.4 ˚C at the bottom and -1.7 ˚C at the top (Fig.
2b).
NG was 27 m deep and about 360 m away from the glacier front. The sea ice was thinner (0.92 m) and the snow cover thicker
(6 cm) compared to SG. The ice had a well developed skeletal layer at the bottom with brown coloration due to algal biomass.
The ice temperature ranged between -2 ˚C at the bottom to -2.7 ˚C at the top (Fig. 2b). The IE station was about 75 m deep
and 50 m away from the ice edge. The sea ice was thinnest (0.79 m) and the snow cover thickest (10 cm). Sea ice temperatures
were coldest ranging from -2.2 ˚C at the bottom to -3.1 ˚C on the top (Fig. 2b). Loosely floating ice algae aggregates were
present in the water directly under the ice. The recorded surface PAR irradiance were similar during the primary production
incubation times at SG and IE (SG: average=305 $\mu$E m$^{-2}$ s$^{-1}$, min=13 $\mu$E m$^{-2}$ s$^{-1}$, max=789 $\mu$E m$^{-2}$ s$^{-1}$; IE: average=341 $\mu$E m$^{-2}$
s$^{-1}$, min=37 $\mu$E m$^{-2}$ s$^{-1}$, max=909 $\mu$E m$^{-2}$ s$^{-1}$). Using published attenuation coefficients irradiance directly under the ice was 5
$\mu$E m$^{-2}$ s$^{-1}$ at IE and higher at SG with 9 $\mu$E m$^{-2}$ s$^{-1}$ due to the thinner snow cover.

**3.2 Nutrient variability in sea ice and water**

Overall, nutrient concentrations were highest in the bottom water (4.0- 4.5 $\mu$mol L$^{-1}$ Si(OH)$_4$, 9.1- 9.6 $\mu$mol L$^{-1}$ NO$_X$, 0.7-0.8
$\mu$mol L$^{-1}$ PO$_4$) and depleted at the surface and in the sea ice, with the exception of the under-ice water (UIW, 0- 1 cm under
the sea ice) of SG, where NO$_X$ (10 $\mu$mol L$^{-1}$) and silicate (19 $\mu$mol L$^{-1}$) levels were exceptionally high (Fig. 4). SG had overall
higher levels of silicate and NO$_X$ compared to the IE at both 1 m below the sea ice (factors of 3 for Si(OH)$_4$ and 2 for NO$_X$)
and bottom ice (factor of 18 for Si(OH)$_4$ and 3 for NO$_X$ compared to IE bottom ice) (Fig. 4). Silicate concentrations deeper in
the water column were similar at all the stations with values of ca 4 $\mu$mol L$^{-1}$ . Close to the surface silicate was reduced to 1.6
$\mu$mol L$^{-1}$ at 1 m at the IE, while it stayed at 4.3 $\mu$mol L$^{-1}$ at SG (Fig. 4a). In the water column, NO$_X$ and phosphate gradients
were similar between the sites. However in sea ice, NO$_X$ concentrations were more than two times higher at SG than at the IE.
In the bottom 30 cm of sea ice all nutrients had higher concentrations at SG, except for phosphate, which was depleted in the
bottom 3 cm of SG, but not in the bottom of IE sea ice. In the ice interior in 50- 70 cm distance from the ice bottom, also the
other nutrients were depleted at SG, before rising slightly towards the surface of the ice. N:P ratios were generally highest at



SG with values above 40, exceeding Redfield ratios in the surface water and sea ice. N:P ratios at the IE were below Redfield in the entire water column and bottom sea ice with values ranging from 10 to 13. A slight increase in $NO_X$ was observed at the sea ice-atmosphere interface at NG and SG. Subglacial outflow water and glacial ice had relatively low nutrient levels (in
glacial ice: $Si(OH)_4 < 0.3$ µmol $L^{-1}$, $NO_X < 0.9$ µmol $L^{-1}$, $PO_4 < 0.75$ µmol $L^{-1}$, in outflow: $Si(OH)_4 < 1.5$- 2.0, $NO_X$ 1.8- 2.3 µmol $L^{-1}$, $PO_4 < 0.1$ µmol $L^{-1}$), but the nutrient concentrations in subglacial outflow water were higher than in most sea ice samples and the depleted surface water (1 m under the sea ice) at the IE.

Nutrient versus salinity profiles give indications of the endmembers (sources) of the nutrients (Fig. 5). A positive correlation
for example would indicate conservative mixing (assuming high salinity Atlantic water endmember had higher concentrations than melt water). Biological uptake and remineralisation as well as physical processes, such as external inputs by meltwater could inverse or eliminate the correlation. In the water column at NG and IE silicate ($R^2$=0.66, p=0.008), $NO_X$ ($R^2$=0.62, p=0.01) and phosphate ($R^2$=0.69, p=0.005) showed conservative positive mixing patterns (Fig. 5a-c). SG showed a negative correlation for silicate ($R^2$=0.86, p<0.0001) but not positive relations for $NO_X$ and $PO_4$ (Fig. 5d-f). At SG, silicate
concentrations were higher with lower salinities. The same pattern was observed in sea ice, scaled to brine salinities, with higher silicate and $NO_X$ concentrations in the fresher SG ice, compared to NG and IE (Fig. 5g-i). However, the $R^2$ value were lower in particular for $Si(OH)_4$ ($NO_X$: $R^2$=0.18, p=0.059; $Si(OH)_4$: $R^2$=0.41, p=0.002).

The contribution of nutrients by upwelling as well as freshwater inflow from glacial meltwater was estimated by linear mixing
calculations. At 1 m below the sea ice, about 32 % of the water was derived from glacial meltwater based on salinity-based mixing of glacial meltwater and local Arctic water (Table 1). The remaining 68 % came from either bottom water upwelling (25 m at SG as reference) or entrained surface water (IE values at 1 m under the sea ice as reference). Based on a similar estimation for inorganic nutrients, 58 % of $NO_X$ and 48 % of $PO_4$ was provided by subglacial upwelling (Table 1). For silicate, higher concentrations were required in the bottom water of subglacial meltwater at the glacier front to explain the very high
surface concentrations measured. Considering the estimated $NO_X$ and $PO_4$ fractions, the overall fraction of nutrients derived from upwelling was about 53 %. The overall budget 1 m under the sea ice is was 32 % glacial meltwater, 53 % subglacial upwelling (deep water), and 15 % horizontal transport (surface water).

### 3.3 Carbon cycle

Net primary productivity (NPP) was overall one order of magnitude higher at SG than at IE, with the highest production value
occurring within the brackish layer under the ice at SG (5.27 mg $m^{-3}$ $d^{-1}$, Fig. 6, 7). Within this layer, also Chl values were about two times higher compared to IE (21 mg $m^{-3}$ at SG, 9.1 mg $m^{-3}$ at IE), and also the Chl-specific productivity in this layer exceeded values at the other stations (Table 2). Within sea ice, a slightly different pattern emerged. While the primary productivity in the bottom sea ice (0–3 cm) was two times higher at SG compared to IE, Chl values were two order of magnitudes lower (Fig. 6). This indicates high Chl-specific production at SG (5.6 mg C mg Chl $d^{-1}$ in the sea ice and 11.4 mg



C mg Chl d$^{-1}$ integrated over 25 m depth). At the IE, the contribution of released ice algae to algal biomass in the water column was higher and the overall vertical Chl flux was about 1.5 times higher than at SG at 25 m depth. Bacterial biomass was comparable at both stations with higher biomass concentrations within the ice than in the water column. Bacterial activity (based on DCF) was comparable in the bottom sea ice at the two sites; however, it was 63x higher in the brackish surface water of SG leading to very high growth rate estimates (Table 2) of 6 mg C m$^{-3}$ d$^{-1}$.


Integrated Chl values over the uppermost 25 m of the water column were nearly identical for SG and IE with values of about 3.75 mg Chl m$^{-2}$ (Table 2). The fraction of Chl was highest at IE (85 %) and lowest at the SG (30 %) (Table 2). The integrated NPP was considerably higher at SG (42.6 mg C m$^{-2}$ d$^{-1}$ at SG, 0.2 mg C m$^{-2}$ d$^{-1}$ at IE), while the vertical export of Chl was about three times higher at IE than SG. This leads to more (14 times) vertical export than production at IE and considerably

lower (5 %) export than production at SG (Table 2). Relative to the standing stock biomass of Chl at IE, 0.2 % of the Chl was renewed daily by NPP at IE and 3 % was vertically exported daily at IE, which would relate - assuming absence of grazing and advection – a daily loss of 3 % of the standing stock Chl. At SG, 38 % was renewed per day, while 2 % were exported. This leads to an accumulation of biomass of 38 % per day, and a doubling time of about 2.6 days. Bacterial growth doubling times were estimated to be between minutes (SG water) and days (IE water), but within hours in sea ice (Table 2).


Considering the N demand based on the carbon based PP measurement (16 mol C mol N$^{-1}$ after Redfield, 1934), about 2 µmol N L$^{-1}$ month$^{-1}$ (equivalent to 32 % of 1 m value for NO$_X$) was needed to sustain the PP measured at SG. Assuming constant PP and steady state nutrient conditions, 32 % of the surface water had to be replaced by subglacial upwelling per month to supply this N demand via upwelling. Since only 62 % of the upwelling water was entrained bottom water the actual vertical water

replenishment rate would be 52 % per month. Assuming a 2 m freshwater layer under the ice, this translates to flux of about 1.1 m$^3$ m$^{-2}$ month$^{-1}$. Considering the distance of 250 m to the glacier front and a width of 1.6 km of the SG bay, this translates to a minimum of about 422,000 m$^3$ month$^{-1}$.

The results from the reciprocal transplant experiment (Fig. 7) showed clearly that the higher NPP at SG, compared to NG was related to the nutrient concentrations (nested ANOVA, p=0.0038, F=10.88). In any combination, sterile filtered water from the

SG had a fertilising effect, increasing PP of IE communities by approx. 30 %. SG communities of the most active fresh surface layer fixed twice as much CO$_2$ when incubated in the same water, compared to incubations in the IE water.

### 3.4 Bacterial, archaeal and eukaryotic communities

After bioinformatic processing 13,043 bacterial and archaeal (16S rRNA) OTUs, belonging to 1,208 genera with between 9,708 and 331,809 reads were retained. Differences between the bacterial 16S sequences of the various sample types indicated

that they can be used as potential markers for the origin of the water (Fig. 8). The first non-metric multidimensional scaling (NMDS1) axis separated sea ice from water communities (ANOSIM, p=0.004, R=0.35) with no overlapping samples (Fig. 8a). Generally IE and NG communities were very similar, while sea ice and under-ice water communities at SG were





significantly different (ANOSIM, p=0.001, E=0.593) from the other fjord samples. The second NMDS2 axis separated communities along a gradient from subglacial communities towards fjord communities, with SG communities being in between

fjord and subglacial communities (Fig. 8a). Bacterial communities at SG in the bottom layer of the sea ice and the brackish water layer were more similar to subglacial outflow communities than the other samples in both 2018 and 2019. Six OTUs were unique to the glacial outflow and SG surface (*Fluviimonas*, *Corynebacterineae*, *Micrococcinae*, *Hymenobacter*, *Dolosigranuum*), which are 6.6 % of their OTUs. The community structure of supraglacial ice samples was very different from any other sample. Also in the most abundant genera clear differences can be detected (Fig. S1). *Flavobacterium* sp. was most

abundant in sea ice and UIW samples in both 2018 and 2019 at SG, but rare or absent in the other samples. *Aliiglaciecola* sp. was characteristic for NG sea ice and UIW samples. *Paraglaciecola* sp. was abundant in NG and IE sea ice and UIW samples, and *Colwellia* sp. was abundant in all sea ice and UIW samples. In sea water samples the genus *Amphritea* sp. was more abundant. *Pelagibacter* sp. was abundant in all samples. Glacial outflow water was dominated by *Sphingomonas* sp. and glacier ice by *Halomonas* sp., which were rare or absent in the other samples.

The eukaryotic community (18S rRNA) consisted of 4,711 OTUs, belonging to 535 genera, with between 2,204 and 15,862 reads. Overall, the same NMDS clustering has been found as for the 16S rRNA sequencing. We found distinctive communities in the sea ice and 1 m layer under the sea ice at SG being significantly different (ANOSIM, p=0.001, R=0.456) to the other samples (Fig. 8c). In fact, the SG surface communities were more similar to the outflow community (Fig. 8c). The clear differentiation between all sea ice and water column communities was also visible in the 18S rRNA samples (ANOSIM,

p=0.005, R=0.192). As for the 16S communities, also the abundant genera differed between the groups (Fig. S2). The cryptophytes *Hemiselmis* sp. and Geminigeraceae were abundant at SG, but rare at the other sites. Dinophyceae, Imbricatea (*Thaumatomastix* sp.) and Bacillariophyceae were abundant in all samples with diatoms being mostly more abundant in sea ice or UIW. The Chytridiomycota family of Lobulomycetaceae were abundant in water samples from 2018, but not 2019. Subglacial outflow water was dominated by unclassified Cercozoa and *Bodomorpha* sp..

In total 22 different taxa were detected by microscopy. The communities composition was clearly separated between sea ice and water samples. Furthermore species composition at SG station differed from NG and IE (Fig. 8b). SG sea ice was completely dominated by unidentified flagellates (potentially *Hemiselmis* sp., Geminigeraceae, and *Thaumatomastix* sp. based on 18S sequences), with the exception of the 70–90 cm layer with high abundances of *Leptocylindrus minimus*. Sea ice samples at NG and IE were dominated by *Navicula* sp. and *Nitzschia frigida*. Water samples were more diverse with abundance of

*Fragillariopsis* sp., *Coscinodiscus* sp., and *Chaetoceros* sp.. Overall, diatoms dominated most samples at NG and IE in sea ice and water samples.

## 4 Discussion

The hydrography, sea ice properties, water chemistry and bacterial communities at SG provide clear evidence for subglacial upwelling at a shallow tidewater outlet glacier under sea ice, a system previously not considered for subglacial upwelling





processes. Briefly, our first hypothesis that subglacial upwelling persists also in winter/spring, supplying nutrient-rich glacial meltwater and upwelling of bottom fjord water to the surface has been confirmed as discussed in detail below.

## 4.1 Indications for subglacial upwelling

The physical properties at SG were distinctly different to stations NG and IE. In contrast to NG and IE, the marine terminating SG site had a brackish surface water layer of 4 m thickness under the sea ice and low sea ice bulk salinities below 1.5 PSU
comparable to sea ice in the nearby tidewater glacier influenced Tempelfjorden (Fransson et al., 2020) and in brackish Baltic sea ice (Granskog et al., 2003). We excluded surface melt or river run off as freshwater sources for the following reasons. With air temperatures below freezing point during the sampling periods, surface runoff based on snowmelt was not possible and no melting was observed during field work. In addition, no major river flow into the main bay studied (Adolfbukta), as indicated by small catchment areas (Norsk Polarinstitutt, 2020). We did observe some subglacial runoff at the southern site of the glacier
(close to SG), however this outflow water froze before it reached the fjord, which was additionally blocked by a 1.33 m thick sea ice cover. The sea ice cover would also block any inputs by atmospheric precipitation, considering the impermeable sea ice conditions especially at SG with brine volume fractions below 5 % (Golden et al., 1998; Fransson et al., 2020). Additional potential freshwater sources could be related to basal glacial ice melt of glacier fronts (Holmes et al., 2019; Sutherland et al., 2019) or icebergs (Moon et al., 2018). However, in the absence of Atlantic water inflow, which is blocked in Billefjorden by
a shallow sill depth at the entrance of Billefjorden (Skogseth et al., 2020), water temperatures were consistently below freezing point (max -0.2 ˚C) and no Atlantic inflow water was detected at any station, which does not allow basal glacial ice to melt. Subglacial meltwater itself is unlikely to lead to basal ice melting due to its low salinity. However, basal ice melt is likely more important in systems with Atlantic water inflows, such as Greenland or Svalbard fjords without a shallow sill (e.g. Kongsfjorden and Tunabreen, Holmes et al., 2019). Sea ice may melt at lower temperatures compared to glacial ice, but the
absence of typical sea ice algae in the water column at SG and the low salinity of the sea ice indicated that this was not the case. In fact, sea ice with a salinity of 1.5 PSU (measured at SG) would melt at -0.08 ˚C (Fofonoff et al., 1983), but the water and ice temperatures did not exceed -0.2 ˚C. Consistent with our study Fransson et al. (2020) also found substantial amount of freshwater in the sea ice in Tempelfjorden (approx. 50 % meteoric water fraction) in a year with large glacier meltwater contribution further supporting the presence of subglacial upwelling under sea ice. Fransson et al. (2020) suggested the
combination of low salinities with high silicate concentrations as indicator for glacial meltwater, which was also the case in our study. In addition, the overall low sea ice salinity and sediment inclusions at SG cannot be explained by sea ice melt, but must originate from another source.

## 4.2 Potential magnitude of subglacial upwelling

Considering the slow tidal currents in our study area (<22 m per 6 h tidal period, Kowalik et al., 2015) and wind mixing
blocked by sea ice, a potential source of the freshwater within Billefjorden may be remains from the previous melting season. Hence, the question of how much subglacial meltwater reaches the surface at SG is important. We estimated that the fresh



surface water was most likely exchanged on time scales of days to weeks. Even slow vertical mixing would be capable to erode the halocline in over six months since the last melting season. The turbidity peak we observed at the halocline would also settle out in a short time (weeks), if not replenished by fresh inputs (Meslard et al., 2018). Vertical export flux was

determined to account for approximately 4% of the Chl standing stock at 25 m. Considering that glacial sediment settles typically substantially faster than phytoplankton due its higher density this suggests that the turbidity peak would erode within days to weeks without fresh sediment input via upwelling (Meslard et al., 2018). Furthermore, the inorganic nitrogen demand for the measured primary productions would consume the present nutrients in a few (approx. 2) months. Assuming steady state, the nutrient uptake by phytoplankton primary production would require an upwelling driven water flux of at least 1.1 m$^3$

m$^{-2}$ month$^{-1}$.

Microbial communities (16S rRNA and 18S rRNA) in SG UIW and sea ice were similar to the subglacial outflow water. Bacterial communities (16S rRNA) at SG shared 6.6 % of their OTUs with subglacial outflow communities, which is twice as much as NG and IE (3.6 %) shared with the outflow communities. Considering the estimated bacterial growth rates and biomass at SG the doubling time of the bacteria would be between 0.5 h and 7 h. However, the use of a conversion factor for biomass

production based on sediment bacterial data is adding uncertainty to the estimation of the bacterial doubling time. Estimates reported from Kongsfjorden in April are indeed longer (3-10 days, Iversen & Seuthe, 2010), as are other Arctic bacterioplankton doubling time estimates ranging between 1.2 days (Rich et al., 1997), 2.8 days (de Kluijver et al., 2013) and weeks (2 weeks, Rich et al., 1997; 1 week, Kirchman et al., 2005).

Based on the growth in the range of hours to days, the distinctive community at SG would have changed to a more marine

community on time scales of weeks, assuming only growth of marine OTUs at SG and settling out or grazing of inactive glacial bacteria taxa. Consequently, the presence of shared OTUs between SG and the glacial outflow indicates a constant supply of fresh inoculum to sustain these taxa. The clearest evidence for outflow comes from the visual observations of subglacial outflow exiting the southern part of the glacier in October 2019, April 2018 and April 2019 which we assume also occurred under the marine terminating front. In fact, subglacial outflows in spring have been observed at various other Svalbard glaciers

with runoff originating from meltwater stored under the glacier from the last melt season and released by changes in hydrostatic pressure or glacier movements (Wadham et al., 2001). Active subglacial drainage systems in winter have also been described elsewhere, and can be sustained by geothermal heat or frictional dissipation, groundwater inputs, or temperate ice in the upper glacier (Wilson 2012; Schoof et al., 2014). This meltwater has also been found to be rich in silicate due to the long contact with the subglacial bedrock during its storage over winter (Wadham et al., 2001; Fransson et al., 2020). We therefore suggest

that winter and spring subglacial upwelling is not unique to Billefjorden, but likely occurs at all polythermal or warm based marine-terminating glaciers.

The amount of upwelling was estimated using hydrographic data. In our study, three water masses were distinguished; i) subglacial outflow (SGO) with low salinity (0 PSU) relatively high temperatures (>0 ˚C) and high silicate concentrations (Cape et al.,2019), (ii) deep local Arctic water (DLAW) with low temperatures (-1.7 ˚C) high salinities (34.6 PSU) and high nutrient

concentrations (Skogseth et al., 2020), and iii) surface local Arctic water (SLAW) with the same temperature and salinity





signature as the DLAW, but depleted in nutrients (Skogseth et al., 2020). Our mixing calculations estimate that 32 % of the SG water 1 m under the sea ice was derived by SGO, which pulled 1.6 times more (53 %) DLAW with it during upwelling. Fransson et al. (2020) found that 30-60 % of glacier derived meltwater was incorporated in the bottom sea ice at the glacier front of Tempelfjorden, again indicating that this is a widespread process at marine terminating glacier fronts.

### 4.3 Importance of subglacial upwelling under sea ice

Compared to the massive subglacial plumes of summer systems (250-500 $m^{-3}$ $s^{-1}$, Hopwood et al., 2020), subglacial upwelling in spring is a small volume transport with only about >1.1 $m^3$ $m^{-2}$ month$^{-1}$ upwelling needed to sustain measured surface primary production. This careful estimate translates to a freshwater input for Billefjorden of at least 1.76 x $10^5$ $m^3$ day$^{-1}$, which is one order of magnitude lower than summer values at Kronebreen (2.7 × $10^6$ $m^3$ day$^{-1}$, Halbach et al., 2019), a Svalbard 465 tidewater glacier of similar size. In addition, less bottom water was entrained with subglacial outflow water (lower entrainment factor) compared to other subglacial upwelling studies (e.g. Hopwood et al.,2020). In our study, each volume of SGO water pulled about the same volume of DLAW with it to the surface (Entrainment factor of 1.6 – see above). This value is low compared to other entrainment factor estimates ranging mostly between 6 and 10 (Hopwood et al., 2020). The entrainment factor is mostly dependent on the depth of the glacier front (Hopwood et al., 2020), which can explain the low rate at 470 Nordenskiöldbreen in Billefjorden, with an estimated depth of 20 m at the terminus (based on CTD cast at terminus in April 2018, data not shown). Kronebreen with a glacier terminus depth of about 70 m and an entrainment factor of 3 is the most comparable tidewater glacier to Nordenskiöldbreen, where these fluxes were estimated. Although entrainment rate was low, it substantially increased summer primary production in Kongsfjorden (Halbach et al., 2019). In spite of the low discharge and entrainment rate of our study, subglacial upwelling appears to be the main mechanism to replenish bottom water with high 475 nutrient concentrations to the surface and can substantially increase spring primary production due to; i) the absence of any other terrestrials inputs, ii) Atlantic water blocked by a shallow sill (Skogseth et al., 2020), iii) very weak tidal currents (Kowalik et al., 2015), and iv) wind mixing blocked by sea ice in Billefjorden.

### 4.4 Importance for under-ice phytoplankton

Our main finding was that i) higher irradiance, ii) a stratified surface layer, and iii) increased nutrient supply via subglacial 480 upwelling allowed increased phytoplankton primary production at SG. Surprisingly, the ice edge station (IE) was light and nutrient limited and supported a lower phytoplankton primary production.

### 4.4.1 Increased light

Despite the substantial subglacial upwelling, the negative effect of light limitation with the massive sediment plumes in summer (Pavlov et al., 2019) were not observed in spring. We did measure a small turbidity peak under the SG sea ice, but the 485 values were comparable to open fjord systems in summer (Meslard et al., 2018, Pavlov et al., 2019), where light is not considered limiting. Under-ice phytoplankton blooms are typically limited by light, which is attenuated and reflected by the





snow and sea ice cover (Fortier et al., 2002, Mundy et al., 2009, Ardyna et al., 2020). Some blooms have been observed, mostly under snow-free sea ice, such as after snow melt (Fortier et al., 2002), under melt ponds (Arrigo et al., 2012, Arrigo et al., 2014), after rain events (Fortier et al., 2002), or at the ice edge related to ice edge driven upwelling (Mundy et al., 2009). In
our study however, light levels available for phytoplankton growth were low compared to other under-ice phytoplankton bloom studies (Mundy et al., 2009, Arrigo et al., 2012), but higher at SG than at IE. This can be explained through the combined effects of sea ice and snow properties at SG. Light attenuation in low salinity sea ice is typically lower due to a lower brine volume (Arst and Sipelgas, 2004). Also, lower sea ice algae biomass and thinner snow cover due to snow removal with katabatic winds (e.g. Braaten 1997; Laska et al., 2012) leads to less light attenuation and lower albedo. Our estimates showed
that about twice as much light reached the water at SG compared to the IE, in spite of the thicker sea ice cover and the estimated light levels of 5 and 9 µE m$^{-2}$ s$^{-1}$ were above the minimum irradiance (1 µE m$^{-2}$ s$^{-1}$) required for primary production (Mock & Gradinger, 1999). Hence, the increased light under the brackish sea ice at SG could be one factor explaining the under-ice phytoplankton bloom observed.

### 4.4.2 Stratified surface layer

The strong stratification at SG is another factor; allowing phytoplankton to stay close to the surface, where light is available, allowing a bloom to form. In fact, Lowry et al. (2017) found that convective mixing by brine expulsion in refreezing leads can inhibit phytoplankton blooms even in areas with sufficient under-ice light and nutrients. At the same time, they found moderate phytoplankton blooms under snow covered sea ice (1–3 mg Chl m$^{-3}$), which was, however, still an order of magnitude lower than the SG values. Our finding of a higher vertical flux at IE compared to SG shows that stronger stratification may indeed
be a contributing factor for the higher phytoplankton biomass at SG due to lower loss rate. However, our reciprocal transplant experiment clearly showed, that location alone (light, stratification) could not explain the increased primary production, but that the water properties at SG had a fertilising effect on algal growth, most probable because of higher nutrient levels, which were limiting at IE.

### 4.4.3 Upwelling of nutrients

Algal growth at IE was co-limited by lower irradiance as well as nutrient concentrations. Dissolved inorganic nitrogen (DIN) to phosphate rations (N:P) at the IE were mostly below Redfield ratios (16:1), especially in sea ice with DIN concentrations below 1 µmol L$^{-1}$, indicating potential nitrogen limitations (Ptacnik et al., 2010), while the N:P ratio at SG was balanced and close to Redfield. Silicate concentrations below 2 µmol L$^{-1}$ are typically considered limiting for diatom growth (Egge & Aksnes, 1992) and this threshold had been reached at UIW and sea ice (concentration estimate in brine volume) at IE, but not
at SG. This indicates that nitrate supplied by deep water upwelling and silicate by combined upwelling and additions from the glacial run off had a fertilising effect on the SG water.. High silicate values have been observed at glacier fronts in other areas such as the Greenland fjords (Azetsu-Scott and Syvitski, 1997) and Tempelfjorden (Fransson et al., 2015:2020). Iron has not been measured, but is an essential micronutrient, often enriched in subglacial meltwater (Hopwood et al., 2020). However,



iron limitation is unlikely in these systems (Hopwood et al., 2020). Besides the subglacial upwelling, nutrient concentrations
may simply be higher due to the shallower depth at SG compared to IE. However, NG was slightly shallower than SG and
algal growth was still limited by nutrients. Besides, neither silicate nor nitrate followed a conservative mixing pattern with
salinity, when including SG samples. In fact, these nutrients only mixed conservatively at IE and NG, but showed non-
conservative mixing at SG, which is a clear evidence for subglacial upwelling and/or meltwater input (Hopwood et al., 2020).
Biological nutrient uptake did not play a significant role, due to relatively low bacterial and primary production. The subglacial
outflow water itself was poor in nitrate, but high in silicate due to the interaction with the bedrock and long residence time
below the glacier (Wadham et al., 2001), which was also found in the Tempelfjorden (Fransson et al., 2015; 2020).
Nordenskiöldbreen has a mix of metamorphic bedrock including silicon rich gneiss, amphibolite, and quartzite, but also
carbonate rich marble (Strzelecki, 2011), which can partly contribute to the high silicate levels observed. The role of bedrock
derived minerals and particles for composition of sea ice chemistry have been described in detail by Fransson et al. (2020).
The values in subglacial outflow water were lower compared to estimates in Greenland (Hawkins et al., 2017, Hatton et al.,
2019), indicating that direct fertilisation in spring may be even more important in other tidewater glacier influenced fjords.
Another potential source may be higher silicate concentrations in the sediments at SG (Hawkins et al., 2017). However. bottom
water values were similar between SG and IE, showing a limited role of higher silicate inputs from sediment. Besides, iron
may be supplied via subglacial outflow (Bhatia et al., 2013, Hopwood et al., 2020), but it is most likely not limiting in coastal
systems (Hopwood et al., 2020).

Another nitrogen source may be ammonium, which has been related to subglacial upwelling in Kongsfjorden (Halbach et al.,
2019). Ammonium regeneration and subsequent nitrification (Christman et al., 2011), may explain the exceptionally high
nitrate concentration of the UIW at SG. In fact, bacterial activity was higher at SG potentially allowing higher ammonium
recycling. Atmospheric inputs of N have been shown in the Baltic Sea, but thinner sea ice and warm periods with increased
sea ice permeability were needed for the N to reach the brine pockets or water column (Granskog et al., 2003). Our $NO_X$
profiles show some evidence of atmospheric N deposition, but only at NG and SG, which may be related to precipitation or
surface flooding. For under-ice phytoplankton, these atmospheric N inputs play no role, but may have benefitted the high algae
biomass layer in the upper ice parts of SG. Overall, the clearest evidence of nutrient limitations and fertilisation by subglacial
upwelling was demonstrated with the reciprocal transplant experiment, which showed an approx. 30 % increase in primary
production related to SG water. Overall, primary production at SG was an order of magnitude higher than at IE. This indicates
that both fertilisation by subglacial upwelling and increased light play a role in increasing phytoplankton primary production.

### 4.4.4 Increased phytoplankton primary production

The integrated primary production to 25 m at SG was 42.6 mg C m$^{-2}$ d$^{-1}$ which is low compared to other marine terminating
glacier influenced fjord systems in summer with integrated NPP of 480 mg C m$^{-2}$ d$^{-1}$ (Hopwood et al., 2020). Also studies in
the same month (April) observed higher primary production rates in a marine-terminating glacier influenced fjord system, such
as Kongsfjorden (405 mg C m$^{-2}$ d$^{-1}$, Hopwood et al., 2020). However, none of these systems was sea ice covered during the





studies and therefore not limited by light compared to our study. Under sea ice, phytoplankton communities have typically much lower NPP rates of 20–310 mg C m$^{-2}$ d$^{-1}$ with only about 10 % or less light transmission reaching the water column (Mundy et al., 2009). These values are more comparable to the SG values, despite the lower light transmission (3 %). In the central Arctic, higher under-ice NPP has been observed, but always related to high light transmission due to the absence of ice, or under melt ponds with light transmissions up to 59 % (Arrigo et al., 2012). However, in the sea ice area north of Svalbard, Assmy et al. (2017) found substantial spring PP below relatively thick sea ice caused by leads. This was also confirmed by large $CO_2$ decrease due to primary production under the sea ice (Fransson et al., 2017). Phytoplankton production under snow covered Arctic sea ice is often considered negligible compared to sea ice algae or summer production. This can be shown in low biomass, mostly consisting of settling sea ice algae (Leu et al., 2015), or very low NPP rates (e.g. Pabi et al., 2008). The same has been observed under Baltic sea ice with similar low light levels and primary production between 0.1–5 mg C m$^{-2}$ d$^{-1}$ under snow covered ice and about 30 mg C m$^{-2}$ d$^{-1}$ under snow-free sea ice (Haecky & Andersson, 1999). These values are comparable to the IE without subglacial meltwater influence, but an order of magnitude lower than the SG production. Moderate blooms of 1–3 mg Chl m$^{-3}$ have been described under snow covered sea ice with equal (3 %) light transmission (Lowry et al., 2017). Lowry et al. (2017) argues that a stratified water column and sufficient nutrients allow moderate blooms even under these low light conditions. Our study found Chl values up to an order of magnitude higher than Lowry et al. (2017), showing that under-ice phytoplankton blooms are indeed important under snow covered sea ice and can be facilitated by subglacial upwelling.

Our study is the first to show that the combination of several factors (stratified water column, increased light and supply of fresh nutrients via tidewater glacier driven processes) can support a rather productive under-ice phytoplankton community, exceeding biomass and production of under-ice phytoplankton in systems with comparable light levels. Besides the increased and extended primary production fuelled by tidewater glacier, the active and abundant phytoplankton taxa in surface water with consistently replenished nutrients, are a viable seed community for summer phytoplankton blooms, once the sea ice disappears and light levels increase (Hegseth et al., 2019). The significantly different community at SG may also contribute to a more diverse seed community available to the entire fjord, compared to fjords without spring subglacial upwelling.

### 4.5 Impact on sea ice algae

### 4.5.1 Impact on biomass and primary production

While phytoplankton biomass and production were clearly enhanced at SG, exceeding levels of other snow covered under-ice systems, sea ice algal biomass and activity had been differently affected. Our third hypothesis suggested lower sea ice algae biomass and production at SG due to the lower brine volume fractions. In agreement with our hypothesis, algal biomass was indeed an order of magnitude lower compared to the IE and NG. However, their production was two times higher, showing more efficient photosynthesis.



Compared to most other sea ice studies conducted at the same period of the year, typically representing the mid-bloom phase with 10–20 mg Chl m$^{-2}$ (Leu et al., 2015), Chl biomass was very low at all stations of our study (<0.32 mg Chl m$^{-2}$). Only Greenland fjords (0.5 mg Chl m$^{-2}$) or pre- and post-bloom systems had comparably low biomass (Leu et al., 2015). The significantly different communities with high number of cryptophyte flagellates, a high proportion of phaeophytin (14–68 % in the bottom 3 cm), and the high contribution of sea ice algae in the water column indicate that we sampled indeed a post-bloom situation. Considering the low air, sea ice and water temperatures and the absence of a fresh UIW layer at the IE, the bloom was most likely not terminated by bottom ice erosion but limited by nutrients. In fact, SG bottom ice was limited by phosphate (0.27 µmol (L brine)$^{-1}$), while the IE was limited by silicate (1 µmol (L brine)$^{-1}$) and nitrogen (N:P = 1 mol N mol P$^{-1}$). This finding fits to earlier studies where phosphate limitations had been described as limiting for brackish sea ice algae at concentrations below 0.27 µmol L$^{-1}$ (Haecky and Andersson, 1999), while N and Si limitations are typical for Arctic sea ice algae (Gradinger, 2009). The low concentrations of phosphate in the subglacial meltwater would partly explain the low concentration in SG sea ice. In addition, most studies summarized by Leu et al. (2015) were done 10 years or more prior our measurements. Hence, an earlier sea ice algae bloom and the earlier termination observed in our study may be related to thinner sea ice due to the warmer climate. In fact, the Greenland study by Mikkelsen et al. (2008) with comparable sea ice algae biomass had the thinnest sea ice cover of 0.5 m sampled in the warmest year (2006). During our study, the weather station in Longyearbyen measured a mean temperature of –3.9 °C in April 2019, which was –8.3 °C above average and the second warmest average April temperature recorded after April 2006 (0.1 °C), indicating that a warmer climate may explain the earlier bloom termination (yr.no).

Similar to algal biomass, primary production (approx. 0.01 mg C m$^{-2}$ d$^{-1}$ at SG and 0.005 mg C m$^{-2}$ d$^{-1}$ at IE, assuming 10 cm productive bottom layer) was considerably lower than in most studies of Arctic sea ice (0.8–55 mg C m$^{-2}$ d$^{-1}$ in the Barents Sea) mentioned by Leu et al.(2015). Only algal aggregates (Assmy et al., 2013) and Baltic sea ice (Haecky & Andersson, 1999) measured similarly low production rates indicating that the senescence of the bloom (aggregates) and brine volume fraction (Baltic Sea) were factors contributing to low primary production in sea ice.

### 4.5.2 Stressors in brackish sea ice

In addition to the post bloom status of the bloom, the lower biomass at SG can be partly explained by the lower brine salinity. Permeability of sea ice is typically related to salinity and temperature, which determine the brine volume. With a brine volume fraction below 5 %, or temperature below -5 °C and salinity below 5 PSU, sea ice is considered impermeable (Golden et al., 1998). At SG, temperatures were higher, but a brine volume fraction above 5 % was only found in bottom ice sections (7–9 %), indicating that the brine channels are weakly connected and algae had limited inhabitable place and nutrient supply (Granskog et al., 2003), especially in the upper layers of the sea ice. In more saline systems, such as the Chuckchi or Beaufort Sea a high flux of seawater through the ice (0.4–19 m$^{3}$ seawater m$^{-2}$ sea ice) has been discussed as crucial to allow continuous primary production and accumulation of biomass (Gradinger, 2009). In impermeable ice, this flux is eliminated. However, the algal biomass at SG was very low, even compared to other brackish sea ice system, such as the Baltic Sea with similar or lower



brine volume fractions and comparable light levels (Granskog et al., 2003: 3-6 mg Chl m$^{-3}$; Haecky & Andersson, 1999: 1.2 mg Chl m$^{-2}$), indicating that other stressors played a role at SG. Grazing is assumed to be a minor control on algae production and biomass in Arctic sea ice (Gradinger, 2002). However, grazing by heterotrophic flagellates on small primary producers has been described as important in the Baltic Sea, indicating that it might plays a role at SG as well (Haecky & Andersson,

1999). SG sea ice communities were indeed dominated by small flagellate algae (microscopy based) and a high proportion of potential grazers (18S rRNA data). Other stressors, such as phosphate limitation, viral lysis, or osmotic stress related to episodic outbursts of subglacial meltwater are likely additional factors explaining the low biomass.

DIC has also been described as potentially limiting for sea ice primary production, especially towards the end of the bloom (Haecky & Andersson, 1999) and may be supplied with the carbonate rich subglacial outflow (Fransson et al., 2020). Higher

mortality due to factors mentioned above, together with the higher measured bacterial activity, allowing recycling of nutrients may be another factor explaining higher production with lower Chl biomass. Last, nutrients may have been replenished recently via advective processes when the brine volume fraction was higher.

At SG another layer of potentially high activity has been found in the upper sea ice. In this layer, depleted nutrient concentrations correspond with high *Leptocylindrus minimus* abundances indicating that these algae were actively taking up

the nutrients, despite the impermeable sea ice. NO$_X$ concentrations increased towards the surface and bottom indicating inputs from surface flooding above (Granskog et al., 2003) and seawater below. Silicate and phosphate were only supplied from the seawater below. The observed brine volume fractions below 5 % would not allow inputs of these nutrients, but episodes with higher temperatures and thereby higher brine volume fractions may be sufficient to supply the needed nutrients to this distinctive layer.

Overall, sea ice influenced by subglacial outflow was very similar to other brackish sea ice such as in the Baltic Sea in regard to structure, biomass and production (Haecky & Andersson, 1999, Granskog et al., 2005). Compared to Arctic sea ice the effect was negative on sea ice algae biomass due to low brine volume fractions, phosphate limitation and potentially higher mortality via grazing and possibly higher osmotic stress.

## 5 Outlook

Our study showed that even a shallow marine-terminating glacier can lead to increased under-ice phytoplankton production by locally enhanced light levels, stronger stratification and nutrient supply by subglacial upwelling, which are all factors expected to change due to climate change. While most of our evidence is circumstantial, the number of different evidence leading to the same conclusion makes our findings rather robust. We propose that our findings are applicable to other tidewater glaciers with a polythermal or warm base, as is common on Svalbard, but also on Greenland (Hagen et al., 1993; Irvine-Fynn

et al., 2011). With a changing climate, tidewater glaciers will retreat and transform towards land terminating glaciers (Błaszczyk et al., 2009). In winter and spring, this would result in the lack of subglacial upwelling and systems more similar to the IE with less nutrients and light available for phytoplankton. The local effect would reduce primary production, biomass and bacterial production in the water column, but higher biomass of sea ice algae with the known Arctic taxa of pennate



diatoms. The pelagic/sympagic benthic coupling would be stronger supporting the benthic food web. In addition, the lack of
upwelling will have consequences for summer production, which is dependent on algae seeds for the development of
phytoplankton blooms (Hegseth et al., 2019). In absence of subglacial upwelling and wind mixing blocked by sea ice or slow
tidal currents, the seed material from the deeper sediments would not reach the water column, leading to a reduced and delayed
phytoplankton summer bloom. Winter and spring subglacial upwelling is most likely present at all polythermal or warm-based
marine-terminating glaciers, which includes glacier termina with much deeper fronts, and much higher entrainment rates of
bottom water. Thus, the effect of spring subglacial upwelling is likely more pronounced in other fjords. Additional effects of
climate change include increased precipitation in the Arctic, which would reduce light levels below the sea ice. However, also
land-terminating glaciers would allow snow removal by katabatic wind as discussed for Nordenskiöldbreen. Another impact
of climate change will be the reduction of sea ice and Atlantification of fjords, leading to increased light, and wind mixing. In
the ice free Kongsfjorden, higher primary production rates have been measured in the same month, indicating that the lack of
sea ice may lead to increased overall primary production (Iversen & Seuthe, 2010). However, Kongsfjorden is still influenced
by subglacial upwelling, supplying nutrients for the bloom (Halbach et al., 2017). In systems not affected by subglacial
upwelling the additional light will most likely not lead to substantially higher primary production as indicated by lower
measured rates in these type of fjords (Hopwood et al., 2020).

## 6 Acknowledgements

The field was funded by the individual Arctic field grants of the Svalbard Science forum for TV, UD, CD, and EH (project
numbers: 282622 (TV, UD, CD), 282600 (TV), 296538 (EH), 281806 (UD)). Additional, funding for lab work and analyses
was obtained by the ArcticSIZE - A research group on the productive Marginal Ice Zone at UiT (grant no. 01vm/h15). JE was
also supported by the the Ministry of Education, Youth and Sports of the Czech Republic ECOPOLARIS, project No.
CZ.02.1.01/0.0/0.0/16_013/0001708 and the Institute of Botany CAS (grant no. RVO 67985939). The publication charges for
this article have been partly funded by a grant from the publication fund of UiT The Arctic University of Norway.
We also wish to thank Jan Pechar, Jiří Štojdl, and Marie Šabacká for field assistance; and Janne Søreide, Maja Hatlebekk,
Christian Zoelly, Marek Brož, Stuart Thomson, and Tore Haukås for field work preparation help. We are also acknowledged
to Melissa Brandner, Paul Dubourg, and Claire Mourgues for the help in the lab and Owen Wangensteen for the help with
bioinformatics analyses. We are thankful for the meteorological data of Petuniabukta supplied by Kamil Laska.

## 7 Authors contributions


TRV designed the experiments, formulated the hypotheses and developed the sampling design with contributions of CD and
UD, and RG. Fieldwork was conducted by TRV, UD, CD, EH, and JE with support by RG and EP for preparations. Lab





analyses were done by TRV, UD, EP, CD, MC and EH. Computational analyses were performed by TRV. The manuscript has been prepared by TRV with contributions of all co-authors.

## 8 Data availability


Environmental data have been archived at Dataverse under the doi number https://doi.org/10.18710/MTPR9E. 18S and 16S rRNA sequences have been archived at the European Nucleotide archive under the project accession number PRJEB40294. The R and unix code for the statistical and bioinformatics analyses are available from the corresponding author upon request. More detailed reports of the fieldwork are available in the Research in Svalbard database under the RiS-ID 10889.

## 9 Competing interests


The authors declare that they have no conflict of interest.

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





Fig 2. Bulk salinity and temperature profiles in a,b) sea ice cores (0 cm at the bottom) and c,d) the water column down to 10 m, of the three stations.






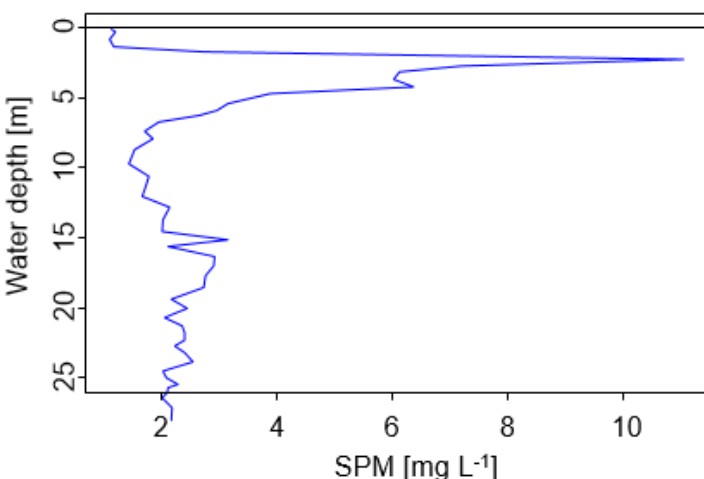

Fig 3. Turbidity profile of the SG station converted to suspended particles








Fig 4. Nutrients in the water column (below grey line) and in sea ice (above the grey line) of a) silicate with a suggested threshold for limitation marked as dashed grey line, b) NO$_X$ as nitrate and nitrite, c) phosphate and d) molar N:P ratios with the Redfield threshold of N:P 16:1 marked as dashed grey line indicating potential N limitation. Dashed lines indicate the position of the ice surface, while solid lines show the measured data.






Fig 5. Salinity-nutrient correlations of NG and IE water samples (a–c), NG, IE, and SG water stations (d–f) and sea ice samples of NG, IE and SG (g–i). Conservative mixing shows as a positive correlation, non-conservative mixing as a negative correlation. Significant correlations (p<0.05) are asterisk marked behind the $R^2$ value.

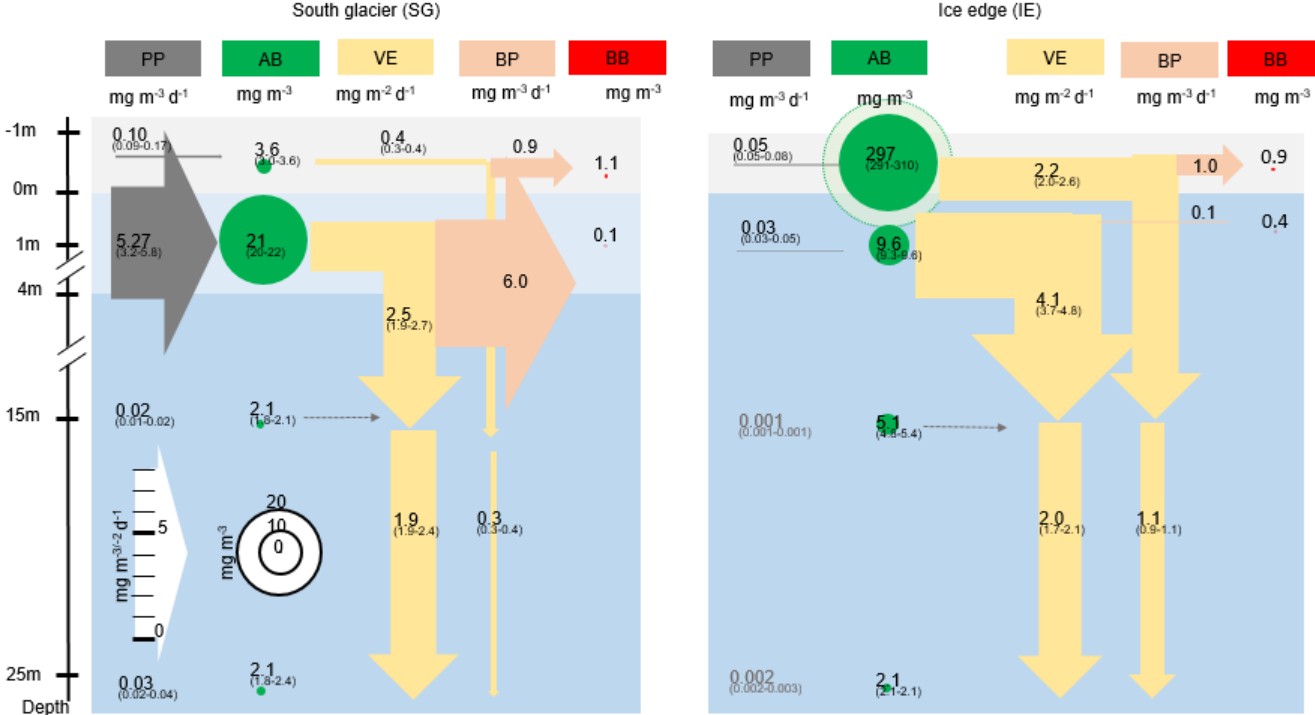

Fig 6. Schematic representation of the C cycle at SG and IE stations. All units are in mg C with the median given in the circles and arrows and the minimum and maximum in brackets below. 0 m depth is at the sea ice water interface. Grey arrows indicate

net primary production (PP) with its height scaled to the uptake rates. Green circles show standing stock algae biomass (AB) converted from Chl to C (conversion factor = 30 gC gChl$^{-1}$, Cloern et al., 1995) with its diameter scaled to the concentrations, except sea ice at IE with the light green circle scaled one order of magnitude higher. Yellow arrows indicate vertical export (VE) of chlorophyll converted to C (conversion factor = 30 gC gChl$^{-1}$, Cloern et al., 1995) with the contribution of sea ice algae and phytoplankton estimated by the fraction of typical sea ice algae in phytoplankton net hauls and the width of the

arrows scaled to the fluxes. Orange arrows indicate bacterial biomass production (BP) based on dark carbon fixation (conversion factor = 129 gC gDIC$^{-1}$, Molari et al., 2013) with the arrows scaled to the values. Red circles to the right are bacteria biomass (BB) assuming 20 fg C cell$^{-1}$ in the bottom sea ice and UIW. The grey area represents sea ice, the light blue area a brackish water layer and the darker blue area deeper saline water layers.





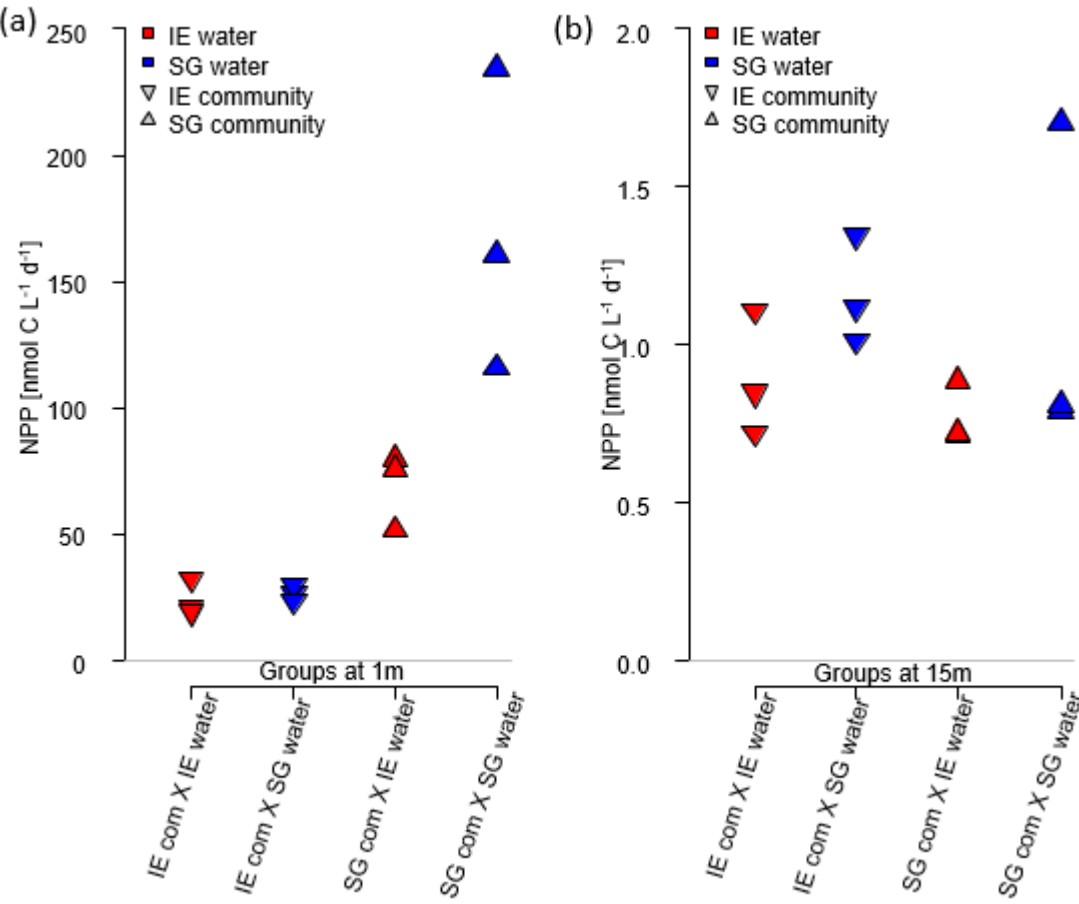


Fig 7. Impact of water source on primary production assessed via a reciprocal transplant experiment. Primary production of IE and SG communities incubated in sterile filtered water originated from either station at a) 1 m and b) 15 m depth. The symbols show the source of the community and the colors indicate the source of the sterile filtered incubation water. The type of incubation water (color) explains the variation in a nested ANOVA with community (symbol) and depth as nested

constrained variables and water source (color) as explanatory variable (p=0.0038, F=10.88).





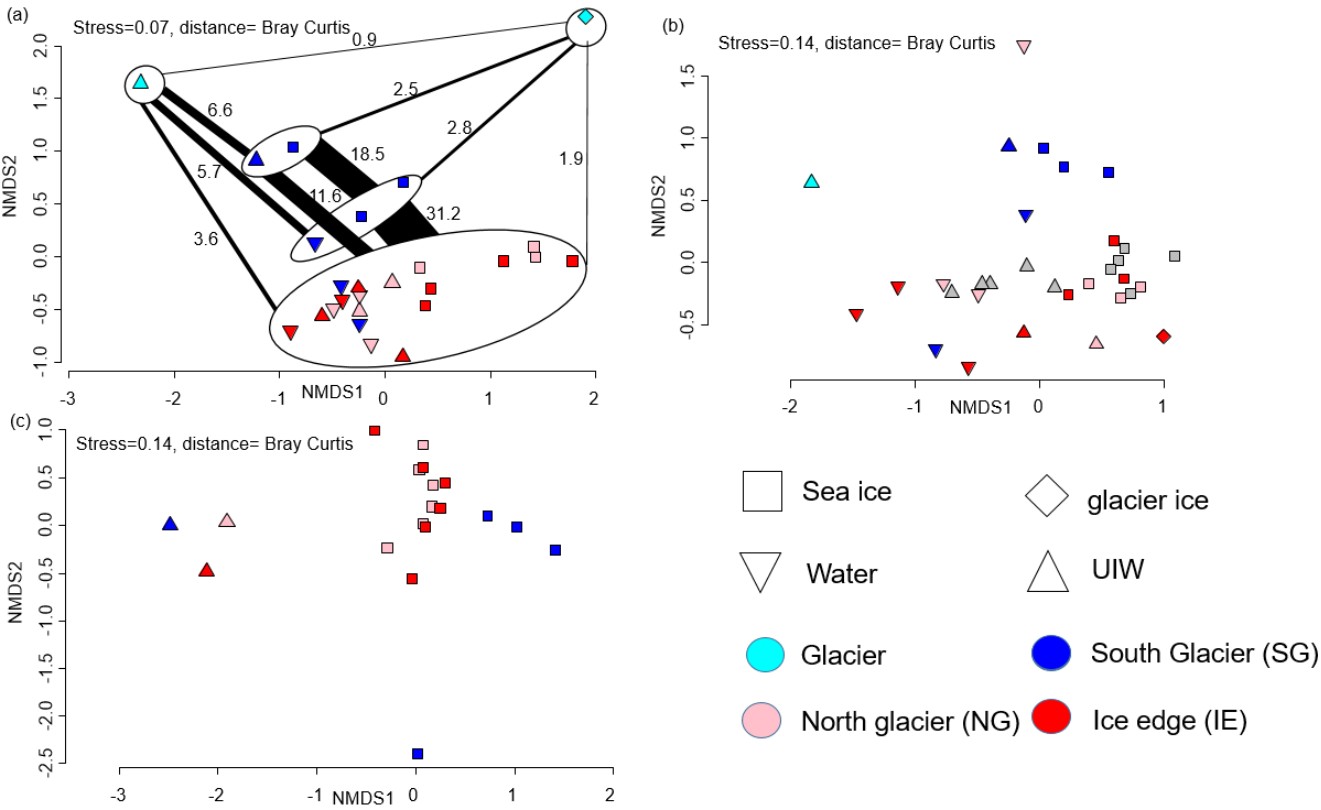

Fig 8. a) NMDS plot of microbial community structure based on 16S data, including samples from April 2018. Groups highlighted in eclipses: glacier ice (top right), glacial outflow (top left), station SG 2019 (top blue), station SG 2018 (bottom blue) and others (bottom). The fraction of shared OTUs (in %) are shown as lines scaled to the fraction [%] of shared OTUs. b) NMDS plot of community structure based on 18S data, including samples from April 2018, c) NMDS plot based on algae abundances in sea ice and UIW based on light microscopic counts.





**Tables**

Table 1. Properties of the three identified water masses and SG surface water and the estimated contributions of the different water masses based on salinity and different nutrients.

| | Surface water | | Bottom water | | Meltwater | | SG 1 m |
|---|---|---|---|---|---|---|---|
| Salinity [PSU] | 34.6 | | 34.6 | | 0 | 32 % | 23.6 |
| Temperature [°C] | -1.4 | | -1.4 | | 0 | | -0.4 |
| Silicate [μmol L$^{-1}$] | 1.59 | 0 % | 4.46 | 84 % | 1.79 | 32 % | 4.30 |
| NO$_X$ [μmol L$^{-1}$] | 3.27 | 10 % | 9.57 | 58 % | 2.06 | 32 % | 6.52 |
| Phosphate [μmol L$^{-1}$] | 0.34 | 20 % | 0.67 | 48 % | 0.09 | 32 % | 0.42 |



Table 2. Integrated standing stock biomass of Chl and fluxes of Chl and C, fractions of the different fluxes and standing stocks, and bacterial production based on dark carbon fixation (DCF).

| Variable | SG | IE | Unit |
|---|---|---|---|
| Chl int. in sea ice | 0.02 | 0.40 | mg m$^{-2}$ |
| NPP in bottom sea ice | 0.10 | 0.05 | mg C m$^{-3}$ d$^{-1}$ |
| Chl int. in 25 m water column | 3.74 | 3.75 | mg m$^{-2}$ |
| Vertical Chl flux to 25 m | 0.07 | 0.11 | mg Chl m$^{-2}$ d$^{-1}$ |
| NPP at 1 m | 5.27 | 0.03 | mg C m$^{-3}$ d$^{-1}$ |
| C based NPP int. over 25 m | 42.6 | 0.2 | mg C m$^{-2}$ d$^{-1}$ |
| Estimated Chl production int. over 25 m | 1.4 | 0.0 | mg C m$^{-2}$ d$^{-1}$ |
| mg C fixed per mg Chl | 11.4 | 0.1 | mg C mg Chl d$^{-1}$ |
| NPP as fraction of Chl standing stock | 38 % | 0.2 % | % Chl renewal d$^{-1}$ |
| Doubling time | 2.63 | 500 | days |
| Vertical Chl flux as % of Chl standing stock | 2 % | 3 % | % export of Chl d$^{-1}$ |
| Vertical Chl flux as % of NPP based Chl prod. | 5 % | 1375 % | % export of NPP d$^{-1}$ |
| Loss of Chl from 15 to 25 m | 12 % | 19 % | Δexp 15m to 25m |
| Average Chl fraction of (Chl + Phaeo) in 0-3 cm ice | 30% | 85% | % Chl |
| Average Chl fraction of (Chl + Phaeo) in water | 47 % | 50 % | % Chl |
| Bacteria DCF ice | 7.0 | 7.6 | µg C m$^{-3}$ d$^{-1}$ |
| Bacteria Biomass prod (DCF based) ice | 0.9 | 1.0 | mg C m$^{-3}$ d$^{-1}$ |
| Doubling time | 1.2 | 0.9 | days |
| Bacteria DCF 1 m | 46.9 | 1.1 | µg DIC m$^{-3}$ d$^{-1}$ |
| Bacteria Biomass prod (DCF based) 1m | 6.0 | 0.1 | mg C m$^{-3}$ d$^{-1}$ |
| Doubling time | 0.02 | 2.9 | days |





**Appendix**

Fig. A1. Community composition of the most abundant genera based on 16S rRNA sequencing data.

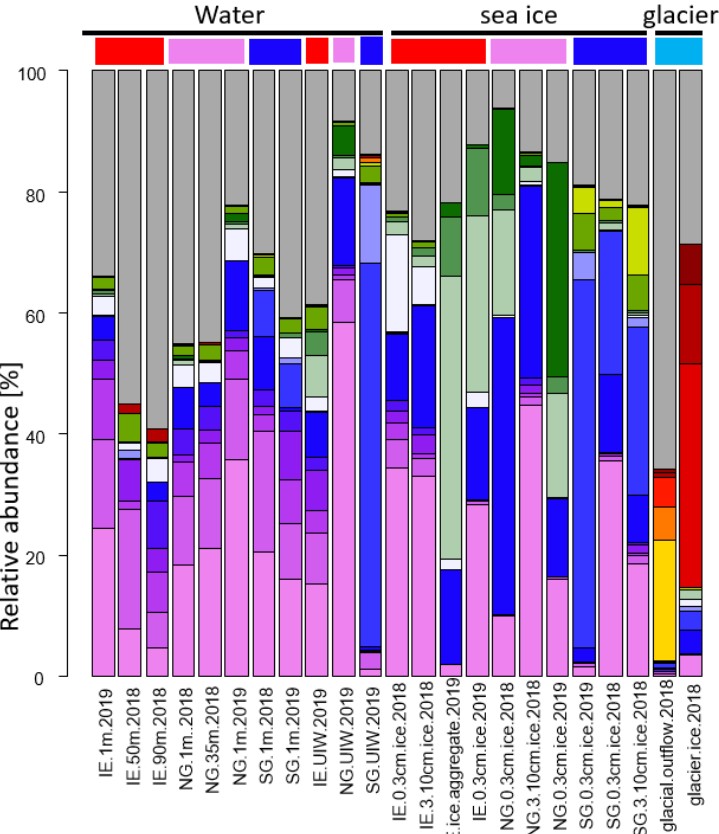

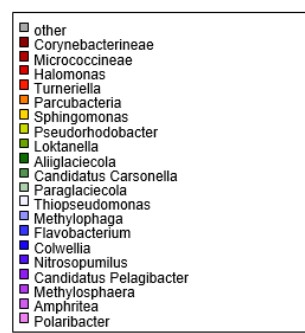







Fig. A2. Community composition based on 18S rRNA sequencing data of the most abundant genera or highest taxonomic

level if no related genus has been found.

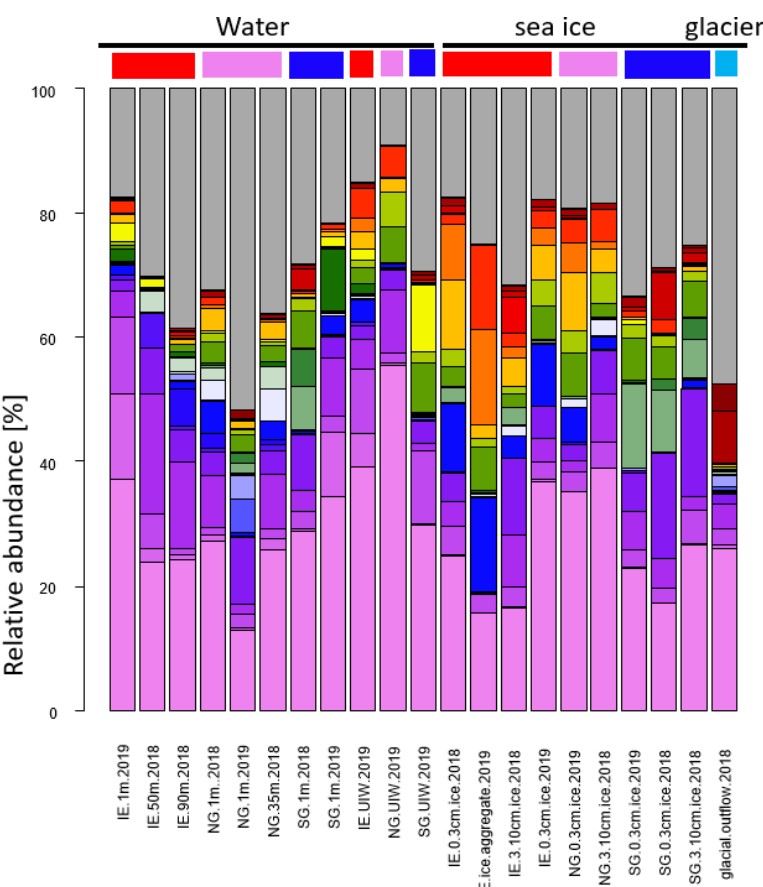

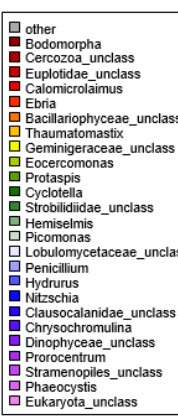







Fig. A3. Sea ice algae and UIW algae community composition of the most abundant taxonomic groups based on light microscopy.

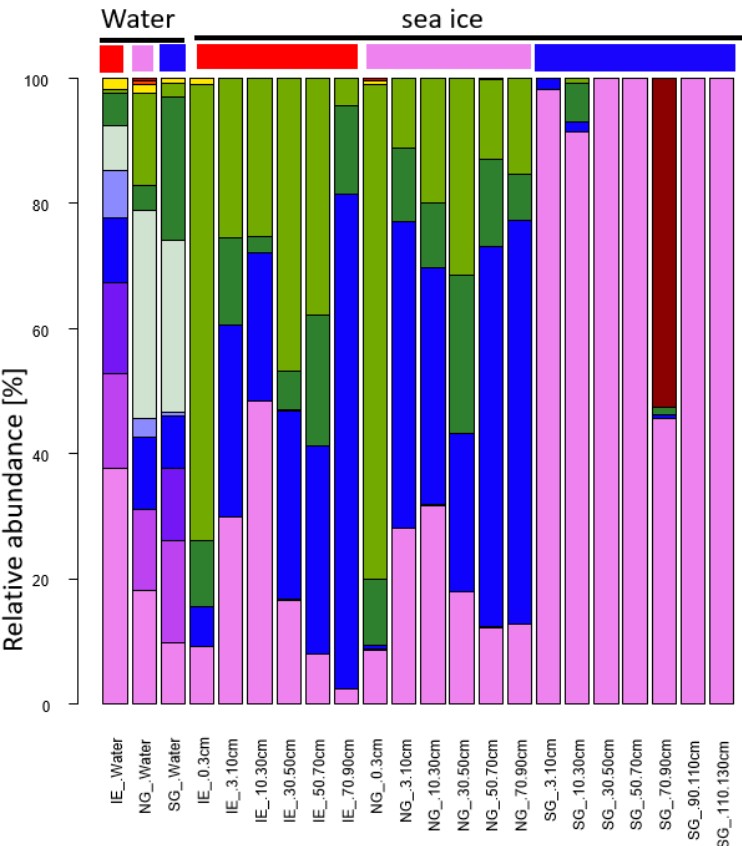

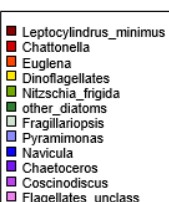







Table B1. Sea ice properties and conversions from bulk salinity and temperature to brine salinity, densities, and brine volume
fractions.

| Station | Ice core section [cm] | Temp [°C] | Brine Sal [PSU] | Ice density [kg m⁻³] | Brine density [kg m⁻³] | Brine volume fraction |
|---------|----------------------|-----------|-----------------|----------------------|------------------------|------------------------|
| SG | 0 to 3 | -0.4 | 7.4 | 917.1 | 1005.9 | 7 % |
|  | 3 to 10 | -0.4 | 7.4 | 917.1 | 1005.9 | 9 % |
|  | 10 to 30 | -0.5 | 8.5 | 917.1 | 1006.8 | 5 % |
|  | 30 to 50 | -0.6 | 10.7 | 917.1 | 1008.6 | 3 % |
|  | 50 to 70 | -0.5 | 10.0 | 917.1 | 1008.0 | 1 % |
|  | 70 to 90 | -0.7 | 13.3 | 917.1 | 1010.6 | 0 % |
|  | 90 to 100 | -1.5 | 27.6 | 917.2 | 1022.0 | 1 % |
|  | 110 to 130 | -1.7 | 31.7 | 917.2 | 1025.4 | 5 % |
| NG | 0 to 3 | -2.0 | 37.6 | 917.3 | 1030.0 | 28 % |
|  | 3 to 10 | -2.1 | 39.4 | 917.3 | 1031.6 | 16 % |
|  | 10 to 30 | -2.3 | 42.8 | 917.3 | 1034.2 | 10 % |
|  | 30 to 50 | -2.5 | 46.1 | 917.3 | 1036.9 | 10 % |
|  | 50 to 70 | -2.8 | 51.8 | 917.4 | 1041.4 | 8 % |
|  | 70 to 92 | -2.7 | 51.1 | 917.4 | 1040.9 | 8 % |
| IE | 0 to 3 | -2.2 | 41.3 | 917.3 | 1033.1 | 16 % |
|  | 3 to 10 | -2.4 | 44.1 | 917.3 | 1035.3 | 11 % |
|  | 10 to 30 | -2.6 | 48.3 | 917.4 | 1038.6 | 11 % |
|  | 30 to 50 | -3.0 | 55.6 | 917.4 | 1044.5 | 9 % |
|  | 50 to 70 | -3.1 | 57.7 | 917.4 | 1046.2 | 8 % |
|  | 70 to 80 | -3.1 | 57.3 | 917.4 | 1045.9 | 6 % |







Table B2. Geographic metadata and nutrient concentrations in µmol L$^{-1}$ related to Billefjorden.

| Depth | Station | Latitude (N) | Longitude (E) | type | Depth [m] | Si(OH)$_4$ [µmol L$^{-1}$] | NO$_x$ [µmol L$^{-1}$] | PO$_4$ [µmol L$^{-1}$] | N:P [mol mol$^{-1}$] |
|---|---|---|---|---|---|---|---|---|---|
| UIW | SG | 78°39'03 | 16°56'44 | water | 0.01 | 19.3 | 10.4 | 0.19 | 55.8 |
| 1 m | SG | 78°39'03 | 16°56'44 | water | 1 | 4.3 | 6.5 | 0.42 | 15.7 |
| 15 m | SG | 78°39'03 | 16°56'44 | water | 15 | 4.4 | 8.7 | 0.68 | 12.9 |
| 25 m | SG | 78°39'03 | 16°56'44 | water | 25 | 4.5 | 9.6 | 0.67 | 14.2 |
| UIW | NG | 78°39'40 | 16°56'19 | water | 0.01 | 1.2 | 1.5 | 0.07 | 21.4 |
| 1 m | NG | 78°39'40 | 16°56'19 | water | 1 | 3.3 | 7.6 | 0.53 | 14.3 |
| 15 m | NG | 78°39'40 | 16°56'19 | water | 15 | 3.8 | 8.7 | 0.62 | 14.0 |
| 25 m | NG | 78°39'40 | 16°56'19 | water | 25 | 4.0 | 9.1 | 0.68 | 13.5 |
| UIW | IE | 78°39'09 | 16°34'01 | water | 0.01 | 2.8 | 6.1 | 0.44 | 13.8 |
| 1 m | IE | 78°39'09 | 16°34'01 | water | 1 | 1.6 | 3.3 | 0.34 | 9.7 |
| 15 m | IE | 78°39'09 | 16°34'01 | water | 15 | 3.6 | 7.8 | 0.62 | 12.6 |
| 25 m | IE | 78°39'09 | 16°34'01 | water | 25 | 4.0 | 9.5 | 0.86 | 11.1 |
| Bot | IE | 78°39'09 | 16°34'01 | water | 57 | 4.0 | 9.1 | 0.70 | 13.0 |
| 0-3 cm | IE | 78°39'09 | 16°34'01 | Sea ice | -1.5 | 0.2 | 0.6 | 0.46 | 1.2 |
| 3-10 cm | IE | 78°39'09 | 16°34'01 | Sea ice | -6.50 | 0.1 | 0.2 | 0.04 | 5.1 |
| 10-30 cm | IE | 78°39'09 | 16°34'01 | Sea ice | -20 | 0.1 | 0.6 | 0.01 | 63.5 |
| 30-50 cm | IE | 78°39'09 | 16°34'01 | Sea ice | -40 | 3.6 | 1.0 | 0.04 | 26.6 |
| 50-70 cm | IE | 78°39'09 | 16°34'01 | Sea ice | -60 | 0.1 | 0.3 | 0.01 | 27.1 |
| 70-80 cm | IE | 78°39'09 | 16°34'01 | Sea ice | -75 | 0.1 | 0.5 | 0.01 | 48.1 |
| 0-3 cm | NG | 78°39'40 | 16°56'19 | Sea ice | -1.5 | 0.3 | 0.8 | 1.29 | 0.6 |
| 3-10 cm | NG | 78°39'40 | 16°56'19 | Sea ice | -6.50 | 0.1 | 0.2 | 0.03 | 7.9 |
| 10-30 cm | NG | 78°39'40 | 16°56'19 | Sea ice | -20 | 0.0 | 0.1 | 0.00 | 104.0 |
| 30-50 cm | NG | 78°39'40 | 16°56'19 | Sea ice | -40 | 0.1 | 0.2 | 0.01 | 20.3 |
| 50-70 cm | NG | 78°39'40 | 16°56'19 | Sea ice | -60 | 0.2 | 0.6 | 0.05 | 10.9 |
| 70-90 cm | NG | 78°39'40 | 16°56'19 | Sea ice | -80 | 0.4 | 1.5 | 0.21 | 6.9 |
| 0-3 cm | SG | 78°39'03 | 16°56'44 | Sea ice | -1.5 | 2.9 | 2.2 | 0.02 | 89.8 |
| 3-10 cm | SG | 78°39'03 | 16°56'44 | Sea ice | -6.50 | 3.2 | 3.3 | 0.03 | 97.1 |
| 10-30 cm | SG | 78°39'03 | 16°56'44 | Sea ice | -20 | 2.0 | 2.9 | 0.04 | 71.2 |
| 30-50 cm | SG | 78°39'03 | 16°56'44 | Sea ice | -40 | 0.6 | 1.3 | 0.02 | 68.1 |
| 50-70 cm | SG | 78°39'03 | 16°56'44 | Sea ice | -60 | 0.4 | 1.2 | 0.02 | 57.6 |
| 70-90 cm | SG | 78°39'03 | 16°56'44 | Sea ice | -80 | 0.9 | 0.4 | 0.01 | 38.9 |
| 90-110 cm | SG | 78°39'03 | 16°56'44 | Sea ice | -100 | 2.4 | 2.3 | 0.04 | 56.3 |
| 110-130 cm | SG | 78°39'03 | 16°56'44 | Sea ice | -120 | 2.6 | 2.4 | 0.04 | 55.4 |






Table B3. Geographic metadata and nutrient concentrations related to Nordenskiöldbreen.

| Date | Stat | Lat (N) | Lon (E) | type | Silicate [µmol L⁻¹] | NOₓ [µmol L⁻¹] | Phosphate [µmol L⁻¹] | Nitrite [µmol L⁻¹] | Nitrate [µmol L⁻¹] |
|---|---|---|---|---|---|---|---|---|---|
| 09.07.2018 | NC | 78°38'3 | 16°59'4 | Cryoconite | 0.18 | 0.741 | 0.597 | 0.133 | 0.608 |
| 09.07.2018 | NC | 78°38'3 | 16°59'4 | Cryoconite | 0.179 | 0.555 | 0.75 | 0.084 | 0.471 |
| 09.07.2018 | NC | 78°38'3 | 16°59'4 | Cryoconite | 0.066 | 0.732 | 0.332 | 0.069 | 0.663 |
| 09.07.2018 | NC | 78°38'3 | 16°59'4 | Cryoconite | 0.157 | 0.674 | 1.281 | 0.067 | 0.607 |
| 09.07.2018 | NC | 78°38'3 | 16°59'4 | Cryoconite | 0.044 | 0.681 | 0.163 | 0.052 | 0.629 |
| 09.07.2018 | NR | 78°39'3 | 16°56'5 | Cryoconite | 0.323 | 0.537 | 0.611 | 0.311 | 0.226 |
| 09.07.2018 | NR | 78°39'3 | 16°56'5 | Cryoconite | 0.073 | 0.671 | 0.201 | 0.07 | 0.601 |
| 09.07.2018 | NR | 78°39'3 | 16°56'5 | Cryoconite | 0.062 | 0.361 | 0.383 | 0.077 | 0.284 |
| 09.07.2018 | NR | 78°39'3 | 16°56'5 | Cryoconite | 0.146 | 0.609 | 0.222 | 0.113 | 0.496 |
| 09.07.2018 | NR | 78°39'3 | 16°56'5 | Cryoconite | 0.049 | 0.53 | 0.26 | 0.065 | 0.465 |
| 25.04.2018 | Out | 78°38'2 | 16°75'2 | outflow | 1.535 | 2.304 | 0.083 | 0.009 | 2.295 |
| 25.04.2018 | Out | 78°38'2 | 16°75'2 | outflow | 2.047 | 1.814 | 0.096 | 0.013 | 1.801 |
| 25.04.2018 | NC | 78°38'3 | 16°59'4 | glacier ice | 0.085 | 0.928 | 0.038 | 0.008 | 0.92 |