# Peer review of "Early spring submarine discharge plumes fuel under-ice primary production at a Svalbard tidewater glacier"

_The Cryosphere, 2020_

## Referee Comment (RC1) · Anonymous Referee #1 · 10 Nov 2020

Vonnahme et al., conduct spring-time measurements in a tidewater glacier fjord and provide evidence that entrainment of subsurface-fjord waters by early season fresh-water discharge is a measurable nutrient source to under-ice phytoplankton blooms which would otherwise be nutrient-limited at this time of year. The hypothesis and general idea is quite novel, supported by some in situ data as well as incubation studies, and I think it is an interesting addition to the field. I have no specific expertise in sea-ice or in the analysis of algae community composition- I defer to a more appropriate reviewer on these aspects. I do have quite a few comments throughout the text, but nevertheless found the discussion paper an interesting read suitable for the journal. Most of my comments are minor, or simply requests for a little more clarity.

in some Arctic fjord systems, many don't have this and some of the few available

case studies don't have high primary production.

You do briefly comment on this once, but considering the timescale required to get significant shifts in the groundling line of these glaciers, could you also think about trends in sea-ice loss as well. Could the timing of sea-ice loss from some of these fjords shift, or could sea-ice even completely disappear, before significant changes in grounding line are evident?

"During summer" Here I think the reference you mean is a similar paper by the same author that deduces the 'upwelling' effect in summer has a measurable impact on silicic acid availability1, the paper cited is from the same area but concerns the spring bloom in the same fjord (which has some sea-ice cover in spring)2. I think you could be a little more precise here.

Is time not also important here? If you didn't have any sediment close to the glacier front, the nature of the upwelling followed by outflow at the surface would surely lead to relatively low primary production at the glacier front anyway, because the freshly upwelled water is being laterally advected away from the front i.e. I assume you would never see the highest cell counts here even in the absence of high turbditity? Looking specifically at the system studied by Meire et al., the bloom also seems to peak a fair way downstream of the large glaciers, even though there isn't much turbidity in the inner fjord.

I don't think there's a perception of no freshwater release from these systems over-winter, there are several papers demonstrating this (for a good recent Arctic example3), although I agree there is a big problem with bias in the distribution of data towards the peak meltwater season, and model discharge curves do I think include a little early/late season discharge comparable to that observed here4. I think a more accurate state-ment would be that whilst it's known that a little discharge occurs early/late season there simply isn't much data to quantify it.

I think you should distinguish 3 sources here, as written icebergs and terminus melt,
but also subglacial discharge at the grounding line in summer. (Reference?)

"while at the same time entrapping considerably less light absorbing sediments" I'd be genuinely interested to know if you can provide data/refs to support this. I'm not sure we know what turbidity looks like in these sub-surface plumes and to what extent it represents particles from the ice melt/basal erosion, or resuspension, and how this changes through the year, although I would agree it would be expected to be low in this environment given the description of the fjord

"this spring upwelling mechanism could be the primary mechanism to significantly increase primary production" does not read well, try "could be a mechanism via which primary production is increased in tidewater fjords compared to similar fjords without these glaciers..." or similar.

I think you need a reference here. Retreat may generally co-occur with shoaling of the grounding line, but not always, and there may also be an increase in discharge which could offset shoaling to some extent as entrainment also depends on freshwater discharge volume. Also, you comment on upwelling being eliminated, but wind-driven upwelling will remain, so maybe be a bit more precise.

'shallow' Do you know the approximate depth?

"were melted in 50 % vol/vol sterile filtered" Is there a reason for this?

One metre or 1 m

Does the exact salinity you use for your inflowing seawater have a large impact on your calculations, can you state what the value of 34.6 refers to? Also, can you clarify to what extent small changes in this would matter (you may want to calculate the saline endmember uncertainty and propagate it?

What does 'net haul' refer to?

"where NOX (10 $\mu$mol L-1) and silicate (19 $\mu$mol L-1) levels were exceptionally high (Fig. 4)." Am I right that this is basically driven by one data point? If so, how do you explain such high NOx concentrations? Is it possible that this value is an anomaly? And what are the implications of this? From your profile, and from the range of the other samples I guess that this is an outlier for both NOx and Si, with the NOx harder to explain from environmental processes. If this is an outlier, I think the calculations throughout need amending or flagging to reflect this, noting that there is a large difference depending on whether or not this data point is included.

"N:P ratios were generally highest. . ." Somewhere it would be interesting to comment on what drives this trend? Is it a source of N, or a sink/dilution of P? If saline water inflow dominated the N and P supply, would you expect such strong shifts? I suspect you need some sort of local process leading to a net accumulation of N or loss of P to get these ratios (you do comment on this for NH4 briefly), and whilst there are no other spring studies I can think of looking at this, I think a few papers have commented on some not particularly well explained P loss in similar environments in summer 5,6.

"Nutrient versus salinity profiles give indications of the endmembers (sources) of the nutrients (Fig. 5). A positive correlation for example would indicate conservative mixing (assuming high salinity Atlantic water endmember had higher concentrations than melt water). Biological uptake and remineralisation as well as physical processes, such as external inputs by meltwater could inverse or eliminate the correlation." This isn't quite right and needs a bit more clarity, you will find a lot of literature on this in marine chemistry or in a good textbook. In simple terms, a linear correlation shows conservative mixing, the absence of a non-linear correlation suggests non-conservative processes (although there are some subtleties to this, some physical factors can also lead to non-linearity). The gradient, not the strength of the correlation, indicates whether fresh, or saline, endmembers have a higher concentration, i.e. an increasing nutrient concentration with salinity (positive gradient) suggests saline inflow has higher nutrient concentrations, whereas a decrease with salinity suggests (negative gradient) a higher freshwater concentration.

"The contribution of nutrients by upwelling as well as freshwater inflow from glacial meltwater was estimated by linear mixing calculations". Can you show these, maybe in the supplement, I am a little confused mainly because of the unclear description above. Similarly for the % nutrient values, please clarify how these were calculated and consider the error on them – especially if it is the case that the single SG value with very high NOx and Si is basically dominating the trend and an outlier.

Can you clarify what you mean by the vertical export of Chl a and how this was calculated please

These values are hard to compare as written because the first (250-500) refers to the vertical plume volume, whereas the second (1.1) refers to the volume transport over the fjord surface, with different units and spatial scales, it would be better to calculate a set of numbers with the same units for comparison.

'careful' implies errors/uncertainties are quantified, at the moment I would say it was a little crude.

465-468 There are 2 sentences here basically saying the same thing

'depth of glacier front' it would be better to cite the physical studies which specifically show this rather than a review

It would be useful to mention the grounding line estimate earlier in the text required for photosynthesis or something more specific (primary production can occur in the dark)

Yes I would agree, but I don't think the review cited explicitly shows this. You can however find a lot of work that suggests most of the Arctic is basically nitrate limited based on observed macronutrient distributions in summer7, and I think this has been explicitly tested closed to Svalbard showing no significant effect of Fe additions8.

I'm not sure what you mean by nutrient concentrations are higher at shallower depths, something due to relatively more evident benthic input as you mentioned for NH4 briefly?

As above, I think this is not quite right. Non-conservative silicic acid behavior (but conservative N/P behavior) would suggest glacier associated input from dissolution of glacier-derived particles either directly into the water column or from sediments into the water column, although conservative silicic acid behavior is also equally often observed downstream of glaciers so this is not really a clear universal meltwater signature[9]. I think the only generalizations you could make would concern concentrations that melt is generally expected to be a low or not significant source of nitrate/phosphate, and a more important source of silicic acid, see Ref1 and the supplement for a summary.

530. I'm not sure these values are low compared to Greenland work if you compare to the dissolved values in freshwater. See the supplement for Ref for a summary[1], I suspect if you calculate mean/median for data available for Greenland or Svalbard your values are likely not atypical (for silicic acid especially, I think the mean and range is high because 1 or 2 catchments have exceptionally high concentrations, but median concentrations are likely a few micromolar.) Note spelling Hawkings (I assume).

This is curious, do you have any idea why? Based on the Ref10 cited, this source would be expected to be quite large (i.e. silicic acid entering solution from glacier derived particles) compared to direct glacier inputs of dissolved silicic acid, but I'm not sure how much evidence there is for this, elsewhere around Svalbard I think the same summary can be made as herein that there doesn't seem to be strong evidence for a significant silicic acid source from glacier sediments[6].

Besides. . . This is repetition, I don't think it's particularly controversial to assume NOx as the limiting nutrient in this environment[11,12], I think a very brief comment about Fe would suffice, there is more than ample evidence for really high Fe inputs in and around Svalbard[13,14] and low nitrate levels through most of the growing season.

Given the lack of relevance for atmospheric inputs to under-ice blooms, I don't think you need to discuss this, unless you are writing about incorporation of such nutrients into sea-ice

I'm not sure which value in the cited review you're referring to here, it would be more useful to cite the specific studies that measured primary production (there are many studied on primary production for Svalbard including values specifically for spring which are presumably the best comparison)12,15,16.

570-575 As written this is fine, but please note I think the 'seed' hypothesis specifically referring to inner-fjord communities seeding outer-fjord/shelf areas is not particularly well supported by literature, especially since in the context of sea-ice covered fjords, I think the bloom generally occurs earlier outside the fjord than it does inside (I'm not sure if that is the case here). Elsewhere on seasonal timescales there is evidence of marine inflow changing the in-fjord bloom and not really of the opposite17,18.

These are averages you're referring to? It may be worth commenting on the variability, I expect there's a huge range when you're writing about all Arctic glacier-fjords

'limited by phosphate' do you actually show this, or do you mean than based on measured concentrations, there was a deficiency of phosphate?

This appears very speculative, because I think you are comparing broad regional averages to a spot measurement?

-8.3 above average doesn't make sense

Here, in this section, I think you need to consider where sea-ice cover occurs and also how that and the timing of its breakup may also change in the future.

"the seed material from the deeper sediments would not reach the water column, leading to a reduced and delayed phytoplankton summer bloom" Whilst I've read this hypothesis in a few places, I'm not sure there's much evidence for this, can you cite studies specifically showing this does affect the summer bloom?
650-660 There are a lot of ideas in these paragraphs which are not extensively developed. I think it would be good to either develop these a bit more, or remove them. For the later comment, Holding et al., is probably the best ref I can think of – you also need to think about stratification19 if you want to write about changes in summertime, but I generally suggest you cut this given the spring focus of your manuscript. The writing concerning spring is much better developed and the comments concerning changes in summer bloom lack discussion of the many factors (changing discharge, stratification, circulation) that change seasonally and are generally beyond the scope of the manuscript.

In your comments about how significant/important this process is, maybe you could think about how it works with respect to the availability of nutrients and timing. If I understood correctly, the entrainment occurs from only 20 m depth, so if it started slightly later in the season it would be presumably much less effective as nitrate would already have been drawdown and meltwater would just be mixing into an already nutrient deficient top 20 m layer? Presumably this means the relative timing of bloom onset, and early discharge is an important feature to think about in determining when/when this is important? (And, also sea ice break up, the dates of which presumably are also changing?)

Data files: These are generally well organized but I could not find the nutrient data in the file which the readme says it is in, did I miss something?

Fig. 3 The blue line doesn't quite display properly in my version

Fig. 4 There are a couple of suspect anomalies here, along the line that represents the ice boundary there are a few nutrient concentrations that appear well above the trend for either ice or water column concentrations, are you sure these are real?

Fig. 5 As in text, the description of 'conservative mixing' isn't quite right. "Conservative mixing shows as a positive correlation, non-conservative mixing as a negative correlation". The strength of the correlation indicates roughly how conservative it is. The sign

[Figure]

of the gradient indicates whether the concentrations are increasing or decreasing with salinity i.e. whether freshwater or saline water has the higher concentration. It would be useful to have the actual p values written somewhere.

Fig. 6 This took a while to read, there are a lot of abbreviations.

(1) Meire, L.; Meire, P.; Struyf, E.; Krawczyk, D. W.; Arendt, K. E.; Yde, J. C.; Juul Pedersen, T.; Hopwood, M. J.; Rysgaard, S.; Meysman, F. J. R. High Export of Dissolved Silica from the Greenland Ice Sheet. Geophys. Res. Lett. 2016, 43 (17), 9173–9182. https://doi.org/10.1002/2016GL070191.

(2) Meire, L.; Mortensen, J.; Rysgaard, S.; Bendtsen, J.; Boone, W.; Meire, P.; Meysman, F. J. R. Spring Bloom Dynamics in a Subarctic Fjord Influenced by Tidewater Outlet Glaciers (Godthåbsfjord, SW Greenland). J. Geophys. Res. Biogeosciences 2016, 121 (6), 1581–1592. https://doi.org/10.1002/2015JG003240.

(3) Schaffer, J.; Kanzow, T.; von Appen, W. J.; von Albedyll, L.; Arndt, J. E.; Roberts, D. H. Bathymetry Constrains Ocean Heat Supply to Greenland's Largest Glacier Tongue. Nat. Geosci. 2020. https://doi.org/10.1038/s41561-019-0529-x.

(4) Carroll, D.; Sutherland, D. A.; Hudson, B.; Moon, T.; Catania, G. A.; Shroyer, E. L.; Nash, J. D.; Bartholomaus, T. C.; Felikson, D.; Stearns, L. A.; Noël, B. P. Y.; van den Broeke, M. R. The Impact of Glacier Geometry on Meltwater Plume Structure and Submarine Melt in Greenland Fjords. Geophys. Res. Lett. 2016, 43 (18), 9739–9748. https://doi.org/10.1002/2016GL070170.

(5) Cape, M. R.; Straneo, F.; Beaird, N.; Bundy, R. M.; Charette, M. A. Nutrient Release to Oceans from Buoyancy-Driven Upwelling at Greenland Tidewater Glaciers. Nat. Geosci. 2018. https://doi.org/10.1038/s41561-018-0268-4.

(6) Cantoni, C.; Hopwood, M. J.; Clarke, J. S.; Chiggiato, J.; Achterberg, E. P.; Cozzi, S. Glacial Drivers of Marine Biogeochemistry Indicate a Future Shift to More Corrosive Conditions in an Arctic Fjord. J. Geophys. Res. Biogeosciences 2020, n/a (n/a), e2020JG005633. https://doi.org/https://doi.org/10.1029/2020JG005633.

(7) Codispoti, L. A.; Kelly, V.; Thessen, A.; Matrai, P.; Suttles, S.; Hill, V.; Steele, M.; Light, B. Synthesis of Primary Production in the Arctic Ocean: III. Nitrate and Phosphate Based Estimates of Net Community Production. Prog. Oceanogr. 2013. https://doi.org/10.1016/j.pocean.2012.11.006.

(8) Krisch, S.; Browning, T. J.; Graeve, M.; Ludwichowski, K.-U.; Lodeiro, P.; Hopwood, M. J.; Roig, S.; Yong, J.-C.; Kanzow, T.; Achterberg, E. P. The Influence of Arctic Fe and Atlantic Fixed N on Summertime Primary Production in Fram Strait, North Greenland Sea. Sci. Rep. 2020, 10 (1), 15230. https://doi.org/10.1038/s41598-020-72100-9.

(9) Brown, M. T.; Lippiatt, S. M.; Bruland, K. W. Dissolved Aluminum, Particulate Aluminum, and Silicic Acid in Northern Gulf of Alaska Coastal Waters: Glacial/Riverine Inputs and Extreme Reactivity. Mar. Chem. 2010, 122 (1–4), 160–175. https://doi.org/10.1016/j.marchem.2010.04.002.

(10) Hawkings, J. R.; Wadham, J. L.; Benning, L. G.; Hendry, K. R.; Tranter, M.; Tedstone, A.; Nienow, P.; Raiswell, R. Ice Sheets as a Missing Source of Silica to the Polar Oceans. Nat. Commun. 2017, 8, 14198.

(11) van De Poll, W. H.; Maat, D. S.; Fischer, P.; Rozema, P. D.; Daly, O. B.; Koppelle, S.; Visser, R. J. W.; Buma, A. G. J. Atlantic Advection Driven Changes in Glacial Meltwater: Effects on Phytoplankton Chlorophyll-a and Taxonomic Composition in Kongsfjorden, Spitsbergen . Frontiers in Marine Science . 2016, p 200.

(12) Van De Poll, W. H.; Kulk, G.; Rozema, P. D.; Brussaard, C. P. D.; Visser, R. J. W.; Buma, A. G. J. Contrasting Glacial Meltwater Effects on Post-Bloom Phytoplankton on Temporal and Spatial Scales in Kongsfjorden, Spitsbergen. Elementa 2018. https://doi.org/10.1525/elementa.307.

(13) Herbert, L. C.; Riedinger, N.; Michaud, A. B.; Laufer, K.; Røy, H.; Jørgensen, B. B.; Heilbrun, C.; Aller, R. C.; Cochran, J. K.; Wehrmann, L. M. Glacial Controls on Redox-

Sensitive Trace Element Cycling in Arctic Fjord Sediments (Spitsbergen, Svalbard). Geochim. Cosmochim. Acta 2020. https://doi.org/10.1016/j.gca.2019.12.005.

(14) Hopwood, M. J.; Cantoni, C.; Clarke, J. S.; Cozzi, S.; Achterberg, E. P. The Heterogeneous Nature of Fe Delivery from Melting Icebergs. Geochemical Perspect. Lett. 2017, 3 (2), 200–209. https://doi.org/10.7185/geochemlet.1723.

(15) Hop, H.; Pearson, T.; Hegseth, E. N.; Kovacs, K. M.; Wiencke, C.; Kwasniewski, S.; Eiane, K.; Mehlum, F.; Gulliksen, B.; Wlodarska-Kowalczuk, M.; Lydersen, C.; Weslawski, J. M.; Cochrane, S.; Gabrielsen, G. W.; Leakey, R. J. G.; Lønne, O. J.; Zajaczkowski, M.; Falk-Petersen, S.; Kendall, M.; Wängberg, S.-Å.; Bischof, K.; Voronkov, A. Y.; Kovaltchouk, N. A.; Wiktor, J.; Poltermann, M.; Prisco, G.; Papucci, C.; Gerland, S. The Marine Ecosystem of Kongsfjorden, Svalbard. Polar Res. 2002, 21 (1), 167–208. https://doi.org/10.1111/j.1751-8369.2002.tb00073.x.

(16) Hodal, H.; Falk-Petersen, S.; Hop, H.; Kristiansen, S.; Reigstad, M. Spring Bloom Dynamics in Kongsfjorden, Svalbard: Nutrients, Phytoplankton, Protozoans and Primary Production. Polar Biol. 2012, 35 (2), 191–203. https://doi.org/10.1007/s00300-011-1053-7.

(17) Krawczyk, D. W.; Witkowski, A.; Juul-Pedersen, T.; Arendt, K. E.; Mortensen, J.; Rysgaard, S. Microplankton Succession in a SW Greenland Tidewater Glacial Fjord Influenced by Coastal Inflows and Run-off from the Greenland Ice Sheet. Polar Biol. 2015, 38 (9), 1515–1533. https://doi.org/10.1007/s00300-015-1715-y.

(18) Hegseth, E. N.; Tverberg, V. Effect of Atlantic Water Inflow on Timing of the Phytoplankton Spring Bloom in a High Arctic Fjord (Kongsfjorden, Svalbard). J. Mar. Syst. 2013, 113–114, 94–105. https://doi.org/https://doi.org/10.1016/j.jmarsys.2013.01.003.

(19) Holding, J. M.; Markager, S.; Juul-Pedersen, T.; Paulsen, M. L.; Møller, E. F.; Meire, L.; Sejr, M. K. Seasonal and Spatial Patterns of Primary Production in a High-Latitude Fjord Affected by Greenland Ice Sheet Run-Off. Biogeosciences 2019.

https://doi.org/10.5194/bg-16-3777-2019.

---

## Author Comment (AC1) · 20 Nov 2020

Vonnahme et al., conduct spring-time measurements in a tidewater glacier fjord and provide evidence that entrainment of subsurface-fjord waters by early season freshwater discharge is a measurable nutrient source to under-ice phytoplankton blooms which would otherwise be nutrient-limited at this time of year. The hypothesis and general idea is quite novel, supported by some in situ data as well as incubation studies, and I think it is an interesting addition to the field. I have no specific expertise in seaice or in the analysis of algae community composition- I defer to a more appropriate reviewer on these aspects. I do have quite a few comments throughout the text, but nevertheless found the discussion paper an interesting read suitable for the journal. Most of my comments are minor, or simply requests for a little more clarity.

We want to thank the reviewer sincerely for the thorough and very helpful review. We addressed all comments as described below and believe that the changes improved the manuscript considerably. We used following font colors and highlighting for the changes:
- Grey text: Reviewers comments
- Black text: Our response
- Red text: Text from the manuscript with the suggested changes
- Yellow highlights: Parts of the manuscript that we suggest to change

12 in some Arctic fjord systems, many don't have this and some of the few available case studies don't have high primary production.

We agree that the current formulation is inadequate, since not all fjords have tidewater glaciers, and the primary production in these tidewater-influenced fjord is typically high compared to other summer systems, but not high compared to spring blooms.

We suggest the following change:

Subglacial upwelling of nutrient rich bottom water is known to sustain elevated summer primary production in tidewater glacier influenced fjord systems.

25 You do briefly comment on this once, but considering the timescale required to get significant shifts in the groundling line of these glaciers, could you also think about trends in sea-ice loss as well. Could the timing of sea-ice loss from some of these fjords shift, or could sea-ice even completely disappear, before significant changes in grounding line are evident?

We agree that sea-ice loss is certainly important and should be discussed together with the glacier retreat.

We suggest following change in the abstract:

We suggest that climate change caused retreat of tidewater glaciers could lead to decreased under-ice phytoplankton primary production, while sea ice algae production and biomass may become increasingly important. However, under a scenario of climate driven earlier loss of the sea ice cover, spring phytoplankton primary production may increase.

We also added more details about sea-ice loss and changed in timing of the spring bloom due to earlier sea-ice break up, increased DOM and sediment inputs (brownification) to the discussion.

35 "During summer" Here I think the reference you mean is a similar paper by the same author that deduces the 'upwelling' effect in summer has a measurable impact on silicic acid availability1, the paper cited is from the same area but concerns the spring bloom in the same fjord (which has some sea-ice cover in spring)2. I think you could be a little more precise here.

We thank the reviewer for the comment and changed the reference accordingly.

39 Is time not also important here? If you didn't have any sediment close to the glacier front, the nature of the upwelling followed by outflow at the surface would surely lead to relatively low primary production at the glacier front anyway, because the freshly upwelled water is being laterally advected away from the front i.e. I assume you would never see the highest cell counts here even in the absence of high turbdity? Looking specifically at the system studied by Meire et al., the bloom also seems to peak a fair way downstream of the large glaciers, even though there isn't much turbidity in the inner fjord.

We agree that time and advection are also important here. While phytoplankton biomass increases towards a bloom, it is already advected away from the glacier front.

We suggest following change:

Primary production is typically low in direct proximity to the glacier front due to high sediment loads of the plumes absorbing light, but potentially also due to lateral advection and the time needed for the phytoplankton bloom formation (Meire et al., 2016a,b Halbach et al., 2019).

46 I don't think there's a perception of no freshwater release from these systems overwinter, there are several papers demonstrating this (for a good recent Arctic example3), although I agree there is a big problem with bias in the distribution of data towards the peak meltwater season, and model discharge curves do I think include a little early/late season discharge comparable to that observed here4. I think a more accurate statement would be that whilst it's known that a little discharge occurs early/late season there simply isn't much data to quantify it.

We agree that the formulation need clarification. There are indeed studies on the subglacial upwelling in winter and spring, but they are rare, compared to summer studies.

We add the suggested references and suggest following change:

Studies on
The few studies conducted during the winter/spring period indicate only a small amount of freshwater discharge (e.g. Fransson et al., 2020, Schaffer et al., 2020) compared to the peak melt season. However, the limited amount of data makes the quantification of subglacial outflow difficult. Currently, studies on the potential impacts on sea ice and pelagic primary production are lacking.

48 I think you should distinguish 3 sources here, as written icebergs and terminus melt, but also subglacial discharge at the grounding line

We agree and suggest following change:

In addition to subglacial discharge at the grounding line, tidewater glacier related upwelling mechanisms can also be caused by the melting of deep icebergs (Moon et al., 2018), or the melting of the glacier terminus in contact with warm seawater (Moon et al., 2018, Sutherland et al., 2019).

51 in summer. (Reference?)
We added the references by Moon et al. (2018) and Sutherland et al. (2014)

Moon, T., Sutherland, D. A., Carroll, D., Felikson, D., Kehrl, L., and Straneo, F.: Subsurface iceberg melt key to Greenland fjord freshwater budget, Nat Geosci, 11(1), 49-54, https://org/10.1038/s41561-017-0018-z, 2018.

Sutherland, D. A., Straneo, F., & Pickart, R. S.: Characteristics and dynamics of two major Greenland glacial fjords, Journal of Geophysical Research: Oceans, 119(6), 3767-3791, 2014.

61 "while at the same time entrapping considerably less light absorbing sediments" I'd be genuinely interested to know if you can provide data/refs to support this. I'm not sure we know what turbidity looks like in these sub-surface plumes and to what extent it represents particles from the ice melt/basal erosion, or resuspension, and how this changes through the year, although I would agree it would be expected to be low in this environment given the description of the fjord

We agree that some support in form of references and data is helpful here. We added a short reference to a study at Hansbreen, another polythermal Svalbard tidewater glacier. The study monitored SPM throughout the year and found the lowest SPM value in winter and spring at a depth of about 5-10 m, which fits to our study. The origin is resuspension, as well as subglacial discharge.

We suggest following addition to the introduction:

Sediment inputs during this time of the year are low with highest sediment concentrations deeper in the water column, indicating limited light limiting effects of surface primary production (Moskalik et al., 2018).

Moskalik, M., Ćwiąkała, J., Szczuciński, W., Dominiczak, A., Głowacki, O., Wojtysiak, K., and Zagórski, P: Spatiotemporal changes in the concentration and composition of suspended particulate matter in front of Hansbreen, a tidewater glacier in Svalbard, Oceanologia, 60(4), 446-463 2018.

63 "this spring upwelling mechanism could be the primary mechanism to significantly increase primary production" does not read well, try "could be a mechanism via which primary production is increased in tidewater fjords compared to similar fjords without these glaciers: : :" or similar.

We changed the text accordingly:

We suggest, that in the absence of wind induced mixing due to the seasonal presence of a fast ice cover, upwelling of subglacial outflows could be a mechanism increasing primary production in tidewater fjords compared to similar fjords without these glaciers, especially towards the end of the ice algal/phytoplankton spring bloom when nutrients become limiting (Leu et al. 2015).

65 I think you need a reference here. Retreat may generally co-occur with shoaling of the grounding line, but not always, and there may also be an increase in discharge which could offset shoaling to some extent as entrainment also depends on freshwater discharge volume. Also, you comment on upwelling being eliminated, but wind-driven upwelling will remain, so maybe be a bit more precise.

We clarified the statement in the following way and added a reference:

Higher glacial melt rates and earlier runoffs may initially increase tidewater glacier induced upwelling, due to increased subglacial runoff (Amundson and Carroll, 2018). However, their retreat and transformation into shallower tidewater glacier termini may lead to less pronounced upwelling, unless the shallower grounding line is compensated by the increased runoff (Amundson and Carroll, 2018). Eventually, the tidewater glaciers transform into land terminating glaciers, where wind induced mixing is still possible, but subglacial upwelling is eliminated (Amundson and Carroll, 2018) – potentially reducing primary production.

107 'shallow' Do you know the approximate depth?

We estimated the approximate depth based on bathymetric maps and added the information to the manuscript:

The fjord is separated from Isfjorden, a larger fjord connected to the West Spitsbergen current, by a shallow approximately 30 to 40 m sill (Norwegian Polar Institute, 2020),…

120 "were melted in 50 % vol/vol sterile filtered" Is there a reason for this?

Sea ice is typically melted in in sterile filtered seawater to avoid osmotic shock and lysis of organisms in the ice. Microorganisms in the ice live mostly in brine channels with high salinity, while the frozen ice around is very fresh. Melting the ice around would lead to an overall very low salinity, leading to severe stress to the high-salinity adapted organism.

We added following information:

…were melted in 50 % vol/vol sterile filtered (0.2 µm Sterivex filter, Sigma-Aldrich, St. Louis, MO, USA) seawater to avoid osmotic shock of cells (Garrison and Buck 1986), …

Garrison, D. L., and Buck, K. R.: Organism losses during ice melting: a serious bias in sea ice community studies, Polar Biol 6:237-239, 1986.

131 One metre or 1 m

Since it is the beginning if a sentence we suggest "One metre".

140 Does the exact salinity you use for your inflowing seawater have a large impact on your calculations, can you state what the value of 34.6 refers to? Also, can you clarify to what extent small changes in this would matter (you may want to calculate the saline endmember uncertainty and propagate it?

For the mixing calculations, we used initially the salinity of meltwater (0 PSU) and the bottom water at the glacier front. However, we realize that the average salinity of the well-mixed water column at the ice edge reference site with a salinity of 34.7 is better suited for the calculations. We changed the salinity and added the information where the value comes from and what the standard deviation is. Since the value of 34.7 as the bottom water endmember is very stable, variations would lead to <1% changes in the estimate of the calculations.

We suggest following changes:

The fraction of fjord water vs subglacial meltwater for the water samples was calculated assuming linear mixing of the two water sources with different salinities (glacial meltwater salinity = 0 PSU, average seawater salinity at IE=34.7 ± 0.03 standard deviation), since no other water masses in regard to temperature or salinity signature were present (Table 1). The variability of the IE sea water salinity leads to a small ( 0.09%) uncertainty in the estimated value of the relative contributions of sea water vs subglacial meltwater.

197 What does 'net haul' refer to? -> Mentioned one paragraph above

We gave details about the phytoplankton net hauls in line 181, but changed the term net haul to "phytoplankton net" for clarity.

Suggested change:

For qualitative counting of algal communities, the phytoplankton net and bottom sea-ice samples

We agree that these exceptionally high values have to be treated carefully. We took great care during the nutrient analysis itself and the calibration of the auto-analyzer, and we have no indications on instrument caused errors in our data record. Local remineralization and dissolution of algae biomass at the sea ice-water interface may provide part of the explanation, but other anomalies cannot be excluded since the values are indeed driven by only one sample with no correspondance or obvious source in the values below or above. Thus, we did not use this value for any mixing calculations or estimates, but used instead the value 1 m under the sea ice for all further calculations. The 1 m value is more consistent with the measurements in the water column below and sea ice above. Thus the exceptionally high values had been considered as outliers and not used in our estimates.

We suggest following change:

where NO$_X$ (10 µmol L$^{-1}$) and silicate (19 µmol L$^{-1}$) levels were exceptionally high (Fig. 4). As these values are driven by a single sample, we cannot exclude anomalies to be responsible for these high values. Wetherefor used the values measured 1 m under the sea ice for further calculations in this manuscript as surface water reference.

291 "N:P ratios were generally highest: : :" Somewhere it would be interesting to comment on what drives this trend? Is it a source of N, or a sink/dilution of P? If saline water inflow dominated the N and P supply, would you expect such strong shifts? I suspect you need some sort of local process leading to a net accumulation of N or loss of P to get these ratios (you do comment on this for NH4 briefly), and whilst there are no other spring studies I can think of looking at this, I think a few papers have commented on some not particularly well explained P loss in similar environments in summer 5,6.

We added suggest to add a more thorough discussion of the N:P ratios, including the suggested reference in the following way:

Ammonium regeneration and subsequent nitrification (Christman et al., 2011) may explain the exceptionally high nitrate concentration of the UIW at SG, which can be part of the explanation for the high N:P ratios. In fact, bacterial activity was higher at SG potentially allowing higher ammonium recycling. Another explanation for the high N:P ratios and low phosphate concentrations can be related to phosphate scavenging by iron as discussed by Cantoni et al. (2020).

300 (306) "Nutrient versus salinity profiles give indications of the endmembers (sources) of the nutrients (Fig. 5). A positive correlation for example would indicate conservative mixing (assuming high salinity Atlantic water endmember had higher concentrations than melt water). Biological uptake and remineralisation as well as physical processes, such as external inputs by meltwater could inverse or eliminate the correlation."
This isn't quite right and needs a bit more clarity, you will find a lot of literature on this in marine chemistry or in a good textbook. In simple terms, a linear correlation shows conservative mixing, the absence of a non-linear correlation suggests non-conservative processes (although there are some subtleties to this, some physical factors can also lead to non-linearity). The gradient, not the strength of the correlation, indicates whether fresh, or saline, endmembers have a higher concentration, i.e. an increasing nutrient concentration with salinity (positive gradient) suggests saline inflow has higher nutrient concentrations, whereas a decrease with salinity suggests (negative gradient) a higher freshwater concentration.

We suggest correcting the paragraph in the following way:

Nutrient versus salinity relations can provide indications of the endmembers (sources) of the nutrients (Fig. 5) with a linear correlation being indicative of conservative mixing. A positive correlation indicates higher concentrations of the nutrients of the saline Atlantic water endmember, while a negative correlation points to a higher concentration in the fresh glacial meltwater endmember. Biological uptake and remineralisation could weaken or eliminate this correlation, indicating non-conservative mixing. In the water column at NG and IE silicate (R$^2$=0.66, p=0.008), NO$_X$ (R$^2$=0.62, p=0.01) and phosphate (R$^2$=0.69, p=0.005) showed conservative positive mixing patterns with higher nutrient concentrations in the Atlantic water (Fig. 5a-c). SG showed a negative correlation for silicate pointing to a higher concentration in the glacial meltwater (R$^2$=0.86, p<0.0001). The absence of correlations for NO$_X$ and PO$_4$ indicate non-conservative mixing pointing to the relevance of biological uptake and release measurements (Fig. 5d-f).

We also suggest correcting the figure legend of Fig. 5 (See below).

310 "The contribution of nutrients by upwelling as well as freshwater inflow from glacial meltwater was estimated by linear mixing calculations". Can you show these, maybe in the supplement, I am a little confused mainly because of the unclear description above. Similarly for the % nutrient values, please clarify how these were calculated and consider the error on them – especially if it is the case that the single SG value with very high NOx and Si is basically dominating the trend and an outlier.

We suggest adding following calculations to the supplement. The mentioned outlier values in SG in the UIW sample was not used for the mixing calculations as explained above. For the meltwater fraction at the surface the error related to the average IE salinity is less than 0.1 % (see comment above), the main variation of the % meltwater contribution in the surface layer of SG is related to the salinity at the surface of SG (Fig. R1). We added the error estimate of 0.1 % to the table. For nutrients, the error was estimated based on the variability in the concentrations measured in the triplicates. For NOx the estimated range of contribution by upwelling is thereby 57-59 % (± 1 %) bottom water, for Silicate 89-95 % (± 3 %), and for phosphate 46-49 % (± 3 %).

Equations. Mixing calculations for estimates of the fraction of meltwater (MW$_{Sal}$) based on salinity, and for bottom water based on nutrient concentrations (BW$_{Nuts}$). Sal indicates the average salinities measured at the IE (Sal$_{IE}$), SG at 1m depth (Sal$_{SG1m}$), subglacial outflow (Sal$_{glac}$). Nut indicates the nutrient concentrations of nitrate and nitrite (NOX), silicate (Si), and phosphate (PO4) at 1m under the sea ice at SG (Nut$_{1mSG}$) and IE (Nut$_{1mIE}$), the bottom water of the IE (Nut$_{BW}$), or subglacial outflow water (Nut$_{glac}$).

$$MW_{Sal}[\%] = \frac{Sal_{IE} - Sal_{SG1m}}{Sal_{SG1m} - Sal_{glac} + Sal_{IE} - Sal_{SG1m}} * 100$$

$$MW_{Sal}[\%] = \frac{34.7\ PSU - 23.6\ PSU}{23.6\ PSU - 0\ PSU + 34.7\ PSU - 23.6 PSU} * 100 = 32\ \%$$

$$BW_{Nut}[\%] = \frac{Nut_{1mSG} - MW_{Sal}[\%] * Nut_{glac} - Nut_{1m_{IE}} + MW_{Sal}[\%] * Nut_{1m_{IE}}}{Nut_{BW} - Nut_{1m_{IE}}} * 100$$

$$BW_{NOX}[\%] = \frac{6.52\mu M - 0.32 * 2.06\ \mu M - 3.27\ \mu M + 0.32 * 3.27\ \mu M}{9.57\ \mu M - 3.27\ \mu M} * 100 = 58\ \%$$

$$BW_{Si}[\%] = \frac{4.30\ \mu M - 0.32 * 1.79\ \mu M - 1.59\ \mu M + 0.32 * 1.59\ \mu M}{4.46\ \mu M - 1.59\ \mu M} * 100 = 92\ \%$$

$$BW_{PO4}[\%] = \frac{0.41\ \mu M - 0.32 * 0.09\ \mu M - 0.34\ \mu M + 0.32 * 0.34\ \mu M}{0.67\ \mu M - 0.34\ \mu M} * 100 = 46\ \%$$

[Figure]

Figure R1. Estimated fractions of glacial meltwater in the surface layer of SG.

333 Can you clarify what you mean by the vertical export of Chl a and how this was calculated please

The vertical export flux of Chl a is based on Chl a measurements in the sediment traps. We first convert the measured Chl concentrations (mg m$^{-3}$) to mass (mg) in order to calculate the flux as the mass of Chlorophyll a per unit area and time sedimenting to a certain depth.

Suggested change:

This leads to higher (14 times) vertical export flux based on the sediment trap measurements than production at IE and considerably lower (5 %) export than production at SG (Table 2).

The sediment traps are cylindrical bottles, filled with sterile filtered water, incubated at different depths for about 1 day. The material (e.g. Chl a) sedimenting out (vertical flux) is collected in these cylinders. Since we know the concentration of Chl a in the sediment trap (C in mg m$^{-3}$) and the volume of the cylinders (V in m3), we can calculate the mass of Chl a sedimenting into the trap (mg). With the knowledge of the area above the sediment trap opening (A = m$^2$) we can calculate the amount of Chl per area (mg m$^{-2}$). With the information of the exact incubation time (t in days), we can then calculate the vertical flux (mg m$^{-2}$ d$^{-1}$). The calculation is described in Wiedmann et al. (2016), but we could also add the equation below to the supplement.

$$Vertical\ flux = \frac{C * V}{A * t}$$

462 These values are hard to compare as written because the first (250-500) refers to the vertical plume volume, whereas the second (1.1) refers to the volume transport over the fjord surface, with different units and spatial scales, it would be better to calculate a set of numbers with the same units for comparison.

We suggest adding our estimate in the same unit:

Compared to the massive subglacial plumes of summer systems (250-500 $m^3$ $s^{-1}$, Carroll et al., 2016), subglacial upwelling in spring is a small volume transport with only about 1.1 $m^3$ $m^{-2}$ $month^{-1}$ (approx. 2 $m^3$ $s^{-1}$) upwelling…

462 'careful' implies errors/uncertainties are quantified, at the moment I would say it was a little crude.

We agree and suggest following change:

This rough, but conservative estimate…

465-468 There are 2 sentences here basically saying the same thing

We suggest removing one of the sentences.

469 'depth of glacier front' it would be better to cite the physical studies which specifically show this rather than a review

We suggest citing Carroll et al., 2016 instead. Carroll et al. (2016) also reviews different studies, but for coming to a conclusion of the depth of the glacier front being related to the amount of upwelling, requires a review, or meta-analyses. Since we use the citation as evidence for this specific relationship, we suggest this review as most appropriate. The physical studies alone do not have sufficient data to come to this conclusion.

Carroll, D., Sutherland, D. A., Hudson, B., Moon, T., Catania, G. A., Shroyer, E. L., Nash, J. D., Bartholomaus, T. C., Felikson, D., Stearns, L. A., Noël, B. P. Y., and van den Broeke, M. R.: The impact of glacier geometry on meltwater plume structure and submarine melt in Greenland fjords, Geophys. Res. Lett., 43, 9739–9748, https://doi.org/10.1002/2016GL070170, 2016.

470 It would be useful to mention the grounding line estimate earlier in the text

We suggest adding the information already in the methods description.

486 required for photosynthesis or something more specific (primary production can occur in the dark)

We suggest adding following clarification:
…where light is not considered limiting for photosynthesis.

519 Yes I would agree, but I don't think the review cited explicitly shows this. You can however find a lot of work that suggests most of the Arctic is basically nitrate limited based on observed macronutrient distributions in summer7, and I think this has been explicitly tested closed to Svalbard showing no significant effect of Fe additions8.

We agree that the review is not the most appropriate reference and added the study by Krisch et al. (2020) instead.

520 I'm not sure what you mean by nutrient concentrations are higher at shallower depths, something due to relatively more evident benthic input as you mentioned for NH4 briefly?

We suggest that at a shallower water depth, less physical forcing not necessarily related to subglacial upwelling (e.g. tides (low in Adolfbukta), currents, or wind (unlikely under sea ice), is needed for vertical mixing leading bottom water to reach the surface.

We suggest following clarification:

Besides the subglacial upwelling, nutrient concentrations may be higher due to the shallower water depth at SG compared to IE, facilitating easier vertical mixing down to the bottom.

523 As above, I think this is not quite right. Non-conservative silicic acid behavior (but conservative N/P behavior) would suggest glacier associated input from dissolution of glacier-derived particles either directly into the water column or from sediments into the water column, although conservative silicic acid behavior is also equally often observed downstream of glaciers so this is not really a clear universal meltwater signature9. I think the only generalizations you could make would concern concentrations that melt is generally expected to be a low or not significant source of nitrate/phosphate, and a more important source of silicic acid, see Ref1 and the supplement for a summary.

We suggest following correction:

The differences in the relation of nutrient concentrations versus salinity indicate, that glacial meltwater was not a major source for N and P at SG while the different relation for Si provide evidence for supply through meltwater inflow (Hopwood et al., 2020).

530. I'm not sure these values are low compared to Greenland work if you compare to the dissolved values in freshwater. See the supplement for Ref for a summary1, I suspect if you calculate mean/median for data available for Greenland or Svalbard your values are likely not atypical (for silicic acid especially, I think the mean and range is high because 1 or 2 catchments have exceptionally high concentrations, but median concentrations are likely a few micromolar.) Note spelling Hawkings (I assume).

We are quite confident that the values are low, but would be thankful if the reviewer has any suggestions for references with lower Silicate values in glacial outflow water in Greenland. Overall the data for glacial outflow in Greenland are sparse. We do not think comparing Arctic rivers with our measurements of subglacial outflow would be useful, since additional processes, including additional weathering and uptake by freshwater diatoms would play a role. Overall, rivers have also higher Silicate values. The only samples with lower concentrations than our study are from icebergs (Meire et al., 2016a). The other values in the study by Meire et al. (2016a) are measured from glacial rivers, with the lowest value of 3.4 µmol L$^{-1}$, the lowest mean value of 5.5 µmol L$^{-1}$ and typical mean values around 10 µmol L$^{-1}$. For clarity, we suggest adding the values of our study and the range of the values from Greenland.

The nutrient concentrations in subglacial outflow water were lower (<1.5 – 2 µmol L$^{-1}$) than estimates in Greenland (Hawkings et al., 2017: 0.8-41.4 average 9.6 µmol L$^{-1}$ , Hatton et al., 2019: 9.2-56.9 average 20.8 µmol L$^{-1}$, Cape et al., 2019: 10 ± 8 µmol L$^{-1}$), indicating that direct fertilisation in spring may be even more important in other tidewater glacier influenced fjords.

533 This is curious, do you have any idea why? Based on the Ref10 cited, this source would be expected to be quite large (i.e. silicic acid entering solution from glacier mderived particles) compared to direct glacier inputs of dissolved silicic acid, but I'm not sure how much evidence there is for this, elsewhere around Svalbard I think the same summary can be made as herein that there doesn't seem to be strong evidence for a significant silicic acid source from glacier sediments6.

As indicated by rather low silicate concentrations in the subglacial outflow water, we suggest that the bedrock below Nordenskiøldbreen is overall poor in silicate, at least at the areas, where the subglacial drainage system is in contact with the bedrock. We suggest following change:

However, bottom water nutrient concentrations were similar at SG and IE, indicating a limited role of higher silicate inputs from sediment, presumably due to silicate-poor subglacial bedrock.

534 Besides: : : This is repetition, I don't think it's particularly controversial to assume NOx as the limiting nutrient in this environment11,12, I think a very brief comment about Fe would suffice, there is more than ample evidence for really high Fe inputs in and around Svalbard13,14 and low nitrate levels through most of the growing season.

We agree and suggest removing the sentence, since the information about iron is already given above.

539 Given the lack of relevance for atmospheric inputs to under-ice blooms, I don't think you need to discuss this, unless you are writing about incorporation of such nutrients into sea-ice

Since atmospheric inputs can be an important N source for sea ice algae, we suggest keeping the discussion. Especially at the SG station, we found high biomass of sea ice algae higher up in the ice, indicating that atmospheric inputs may play a role and need to be discussed.

549 I'm not sure which value in the cited review you're referring to here, it would be more useful to cite the specific studies that measured primary production (there are many studied on primary production for Svalbard including values specifically for spring which are presumably the best comparison)12,15,16.

The value is given in a table (Table 1 in Hopwood et al., 2020) and is based on many different studies (6 fjords, 33 datapoints), which makes citing one original research paper tricky. Discussing all studies separately would repeat the review effort of Hopwood et al. (2020) and go beyond the scope of our discussion. Thus, we suggest keeping the review article as main reference. We added however the range of PP in tidewater influenced fjords for clarification.

For a comparison of Kongsfjorden as a similar system on Svalbard, we also agree that adding more specific references to van de Poll et al. (2018) and Hodal et al. (2012) is helpful.

We suggest following changes:

The integrated primary production to 25 m at SG of 42.6 mg C m$^{-2}$ d$^{-1}$ is low compared to values from other marine terminating glacier influenced fjord systems in summer with mean integrated NPP of 480 ±403 mg C m$^{-2}$ d$^{-1}$ (reviewed by Hopwood et al., 2020), including studies in Kongsfjorden on Svalbard (250 -900 mg C m$^{-2}$ d$^{-1}$, Van de Poll et al. 2018). Also studies in the same time (1 May) observed higher primary production rates in a marine-terminating glacier influenced fjord system, such as Kongsfjorden (1520-1850 mg C m$^{-2}$ d$^{-1}$, Hodal et al., 2012).

570-575 As written this is fine, but please note I think the 'seed' hypothesis specifically referring to inner-fjord communities seeding outer-fjord/shelf areas is not particularly well supported by literature, especially since in the context of sea-ice covered fjords, I think the bloom generally occurs earlier outside the fjord than it does inside (I'm not sure if that is the case here). Elsewhere on seasonal timescales there is evidence of marine inflow changing the in-fjord bloom and not really of the opposite17,18.

Our main support is the paper by Hegseth et al. (2019), which describes microalgae derived from sediment upwelling/mixing in the fjord as crucial source of inoculum for a spring bloom. Especially in Billefjorden with little Atlantic water inflow due to the shallow sill, slow tidal currents, and a suspected net advection away from the glacier front, we expect this mechanism to also be important in Billefjorden. However, based on your literature, we realize that this hypothesis is not widely accepted and formulated the statement more carefully.

We suggest a more careful discussion in the following way:

…, may be a viable seed community for summer phytoplankton blooms, once the sea ice disappears and light levels increase (Hegseth et al., 2019).

585 These are averages you're referring to? It may be worth commenting on the variability, I expect there's a huge range when you're writing about all Arctic glacier-fjords

We agree and suggest giving the range instead of the average. We also add a reference citing the original study, instead of the review by Leu et al. (2015).

Only Greenland fjords (0.1-3.3 mg Chl m$^{-2}$) or pre- and post-bloom systems had comparably low biomass (Mikkelsen et al., 2008, Leu et al., 2015).

589 'limited by phosphate' do you actually show this, or do you mean than based on measured concentrations, there was a deficiency of phosphate?

We suggest changing the term "limited" to "deficient".

595 This appears very speculative, because I think you are comparing broad regional averages to a spot measurement?

We agree and realize that this discussion is not crucial for the paper and, thus, suggest removing it.

599 -8.3 above average doesn't make sense
We referred to 8.3 not -8.3 and corrected the error.

644 Here, in this section, I think you need to consider where sea-ice cover occurs and also how that and the timing of its breakup may also change in the future.

We agree that sea-ice cover and changes with climate change need to be discussed here and suggest following additions:

Another impact of climate change will be the reduction and earlier break-up of sea ice and Atlantification of fjords, leading to increased light and wind mixing. In the ice free Kongsfjorden, higher primary production rates have been measured in the same month, indicating that the lack of sea ice may lead to increased overall primary productivity at that time of the year (Iversen & Seuthe, 2010). However, Kongsfjorden is still influenced by subglacial upwelling, supplying nutrients to the bloom (Halbach et al., 2017). In systems not affected by subglacial upwelling the additional light through sea ice melt will most likely not lead to substantially higher primary production as indicated by lower measured rates in these type of fjords (Hopwood et al., 2020). Due to the shallow (20 m) grounding depth in our study site Billdefjorden the estimated fluxes and the nutrient entrainment factors are rather low. In an ice-covered fjord with limited wind and tidal mixing, we suggest subglacial upwelling as a major source for nutrients to the euphotic zone. However, if sea ice disappears, we hypothesize that wind induced mixing would be strong enough to exceed the role of subglacial upwelling. However, katabatic winds reaching the surface water could also increase subglacial upwelling as described by Halbach et al. (2017) and the overall effects are unclear. Direct silicate fertilization would likely have a limited effect in an ice-free fjord since the primary production in the fjord is nitrate and not silicate limited due to the later stage of the spring bloom (Hegseth et al., 2019). In summary, we suggest that subglacial upwelling in winter/spring is important for phytoplankton blooms, but only in a sea-ice covered fjord. The future of the winter/spring

650 "the seed material from the deeper sediments would not reach the water column, leading to a reduced and delayed phytoplankton summer bloom" Whilst I've read this hypothesis in a few places, I'm not sure there's much evidence for this, can you cite studies specifically showing this does affect the summer bloom?

Our main support is the paper by Hegseth et al. (2019), which describes microalgae derived from sediment upwelling/mixing in the fjord as crucial source of inoculum for a spring bloom. Especially in Billefjorden with little Atlantic water inflow due to the shallow sill, slow tidal currents, and a suspected net advection away from the glacier front, we expect this mechanism to also be important in Billefjorden. However, since the support lies in another study and not in our data, we suggest removing this sentence.

650-660 There are a lot of ideas in these paragraphs which are not extensively developed. I think it would be good to either develop these a bit more, or remove them. For the later comment, Holding et al., is probably the best ref I can think of – you also need to think about stratification19 if you want to write about changes in summertime, but I generally suggest you cut this given the spring focus of your manuscript. The writing concerning spring is much better developed and the comments concerning changes in summer bloom lack discussion of the many factors (changing discharge, stratification, circulation) that change seasonally and are generally beyond the scope of the manuscript.
In your comments about how significant/important this process is, maybe you could think about how it works with respect to the availability of nutrients and timing.
If I understood correctly, the entrainment occurs from only 20 m depth, so if it started slightly later in the season it would be presumably much less effective as nitrate would already have been drawdown and meltwater would just be mixing into an already nutrient deficient top 20 m layer? Presumably this means the relative timing of bloom onset, and early discharge is an important feature to think about in determining when/when this is important?
(And, also sea ice break up, the dates of which presumably are also changing?)

We suggest removing all references to summer and focus on spring changes and extend our discussion on sea-ice retreat, timing of the bloom and sea-ice retreat vs glacier retreat effects in the following way (See response to comment on line 644):

Another impact of climate change will be the reduction and earlier break-up of sea ice and Atlantification of fjords, leading to increased light and wind mixing. In the ice free Kongsfjorden, higher primary production rates have been measured in the same month, indicating that the lack of sea ice may lead to increased overall primary productivity at that time of the year (Iversen & Seuthe, 2010). However, Kongsfjorden is still influenced by subglacial upwelling, supplying nutrients to the bloom (Halbach et al., 2017). In systems not affected by subglacial upwelling the additional light through sea ice melt will most likely not lead to substantially higher primary production as indicated by lower measured rates in these type of fjords (Hopwood et al., 2020). Due to the shallow (20 m) grounding depth in our study site Billdefjorden the estimated fluxes and the nutrient entrainment factors are rather low. In an ice-covered fjord with limited wind and tidal mixing, we suggest subglacial upwelling as a major source for nutrients to the euphotic zone. However, if sea ice disappears, we hypothesize that wind induced mixing would be strong enough to exceed the role of subglacial upwelling. However, katabatic winds reaching the surface water could also increase subglacial upwelling as described by Halbach et al. (2017) and the overall effects are unclear. Direct silicate fertilization would likely have a limited effect in an ice-free fjord since the primary production in the fjord is nitrate and not silicate limited due to the later stage of the spring bloom (Hegseth et al., 2019). In summary, we suggest that subglacial upwelling in winter/spring is important for phytoplankton blooms, but only in a sea-ice covered fjord. The future of the winter/spring

Data files: These are generally well organized but I could not find the nutrient data in the file which the readme says it is in, did I miss something? -> I will double check after PhD submission( The same for finishing the ENA submission)

We added the missing data to the DATAVERSE archive.

Fig. 3 The blue line doesn't quite display properly in my version

We will upload a figure with higher quality. For the final paper, we will submit vector based PDF files for each figure.

Fig. 4 There are a couple of suspect anomalies here, along the line that represents the ice boundary there are a few nutrient concentrations that appear well above the trend for either ice or water column concentrations, are you sure these are real? -> mentioned the outliers in the results

As mentioned above, these values are based on 1 sample (UIW at SG for NOX and Silicate) and may well be outliers or anomalies based on sampling artifacts, or locally high remineralization/dissolution rates. Thus, we highlight them as outliers in the text and do not use them for the mixing calculations, or detailed discussions.

Fig. 5 As in text, the description of 'conservative mixing' isn't quite right. "Conservative mixing shows as a positive correlation, non-conservative mixing as a negative correlation". The strength of the correlation indicates roughly how conservative it is. The sign of the gradient indicates whether the concentrations are increasing or decreasing with salinity i.e. whether freshwater or saline water has the higher concentration. It would be useful to have the actual p values written somewhere.

As mentioned above, we agree and changed the text in the manuscript and figure legend accordingly.

Suggested change for figure legend:

Fig 5. Linear salinity-nutrient correlations of NG and IE water samples (a–c), NG, IE, and SG water stations (d–f) and sea ice samples of NG, IE and SG (g–i). A higher concentration in saline Atlantic water results in a positive correlation, a higher concentration in glacial meltwater in a negative correlation. Significant correlations (p<0.05) are asterisk marked behind the $R^2$ value.

Fig. 6 This took a while to read, there are a lot of abbreviations.

Due to the large amount of data in this figure, we argue that the amount of text, containing information and assumptions in the methods are necessary. We suggest writing out the abbreviations on top, to make the figure more understandable without reading the legend.

(1) Meire, L.; Meire, P.; Struyf, E.; Krawczyk, D. W.; Arendt, K. E.; Yde, J. C.; Juul Pedersen, T.; Hopwood, M. J.; Rysgaard, S.; Meysman, F. J. R. High Export of Dissolved Silica from the Greenland Ice Sheet. Geophys. Res. Lett. 2016, 43 (17), 9173–9182. https://doi.org/10.1002/2016GL070191.

(2) Meire, L.; Mortensen, J.; Rysgaard, S.; Bendtsen, J.; Boone, W.; Meire, P.; Meysman, F. J. R. Spring Bloom Dynamics in a Subarctic Fjord Influenced by Tidewater Outlet Glaciers (Godthåbsfjord, SW Greenland). J. Geophys. Res. Biogeosciences 2016, 121 (6), 1581–1592. https://doi.org/10.1002/2015JG003240.

(3) Schaffer, J.; Kanzow, T.; von Appen, W. J.; von Albedyll, L.; Arndt, J. E.; Roberts, D. H. Bathymetry Constrains Ocean Heat Supply to Greenland's Largest Glacier Tongue. Nat. Geosci. 2020. https://doi.org/10.1038/s41561-019-0529-x.

(4) Carroll, D.; Sutherland, D. A.; Hudson, B.; Moon, T.; Catania, G. A.; Shroyer, E. L.; Nash, J. D.; Bartholomaus, T. C.; Felikson, D.; Stearns, L. A.; Noël, B. P. Y.; van den Broeke, M. R. The Impact of Glacier Geometry on Meltwater Plume Structure and Submarine Melt in Greenland Fjords. Geophys. Res. Lett. 2016, 43 (18), 9739–9748. https://doi.org/10.1002/2016GL070170.

(5) Cape, M. R.; Straneo, F.; Beaird, N.; Bundy, R. M.; Charette, M. A. Nutrient Release to Oceans from Buoyancy-Driven Upwelling at Greenland Tidewater Glaciers. Nat. Geosci. 2018. https://doi.org/10.1038/s41561-018-0268-4.

(6) Cantoni, C.; Hopwood, M. J.; Clarke, J. S.; Chiggiato, J.; Achterberg, E. P.; Cozzi, S. Glacial Drivers of Marine Biogeochemistry Indicate a Future Shift to More Corrosive Conditions in an Arctic Fjord. J. Geophys. Res. Biogeosciences 2020, n/a (n/a), e2020JG005633. https://doi.org/https://doi.org/10.1029/2020JG005633.

(7) Codispoti, L. A.; Kelly, V.; Thessen, A.; Matrai, P.; Suttles, S.; Hill, V.; Steele, M.; Light, B. Synthesis of Primary Production in the Arctic Ocean: III. Nitrate and

Phosphate Based Estimates of Net Community Production. Prog. Oceanogr. 2013. https://doi.org/10.1016/j.pocean.2012.11.006.

(8) Krisch, S.; Browning, T. J.; Graeve, M.; Ludwichowski, K.-U.; Lodeiro, P.; Hopwood, M. J.; Roig, S.; Yong, J.-C.; Kanzow, T.; Achterberg, E. P. The Influence of Arctic Fe and Atlantic Fixed N on Summertime Primary Production in Fram Strait, North Greenland Sea. Sci. Rep. 2020, 10 (1), 15230. https://doi.org/10.1038/s41598-020-72100-9.

(9) Brown, M. T.; Lippiatt, S. M.; Bruland, K. W. Dissolved Aluminum, Particulate Aluminum, and Silicic Acid in Northern Gulf of Alaska Coastal Waters: Glacial/Riverine Inputs and Extreme Reactivity. Mar. Chem. 2010, 122 (1–4), 160–175.

(10) Hawkings, J. R.; Wadham, J. L.; Benning, L. G.; Hendry, K. R.; Tranter, M.; Tedstone, A.; Nienow, P.; Raiswell, R. Ice Sheets as a Missing Source of Silica to the Polar Oceans. Nat. Commun. 2017, 8, 14198.

(11) van De Poll,W. H.; Maat, D. S.; Fischer, P.; Rozema, P. D.; Daly, O. B.; Koppelle, S.; Visser, R. J.W.; Buma, A. G. J. Atlantic Advection Driven Changes in Glacial Meltwater: Effects on Phytoplankton Chlorophyll-a and Taxonomic Composition in Kongsfjorden, Spitsbergen . Frontiers in Marine Science . 2016, p 200.

(12) Van De Poll, W. H.; Kulk, G.; Rozema, P. D.; Brussaard, C. P. D.; Visser, R. J. W.; Buma, A. G. J. Contrasting Glacial Meltwater Effects on Post-Bloom Phytoplankton on Temporal and Spatial Scales in Kongsfjorden, Spitsbergen. Elementa 2018. https://doi.org/10.1525/elementa.307.

(13) Herbert, L. C.; Riedinger, N.; Michaud, A. B.; Laufer, K.; Røy, H.; Jørgensen, B. B.; Heilbrun, C.; Aller, R. C.; Cochran, J. K.; Wehrmann, L. M. Glacial Controls on Redox- Sensitive Trace Element Cycling in Arctic Fjord Sediments (Spitsbergen, Svalbard). Geochim. Cosmochim. Acta 2020. https://doi.org/10.1016/j.gca.2019.12.005.

(14) Hopwood, M. J.; Cantoni, C.; Clarke, J. S.; Cozzi, S.; Achterberg, E. P. The Heterogeneous Nature of Fe Delivery from Melting Icebergs. Geochemical Perspect. Lett. 2017, 3 (2), 200–209. https://doi.org/10.7185/geochemlet.1723.

(15) Hop, H.; Pearson, T.; Hegseth, E. N.; Kovacs, K. M.; Wiencke, C.; Kwasniewski, S.; Eiane, K.; Mehlum, F.; Gulliksen, B.; Wlodarska-Kowalczuk, M.; Lydersen, C.; Weslawski, J. M.; Cochrane, S.; Gabrielsen, G. W.; Leakey, R. J. G.; Lønne, O. J.; Zajaczkowski, M.; Falk-Petersen, S.; Kendall, M.; Wängberg, S.-Å.; Bischof, K.; Voronkov, A. Y.; Kovaltchouk, N. A.; Wiktor, J.; Poltermann, M.; Prisco, G.; Papucci, C.; Gerland, S. The Marine Ecosystem of Kongsfjorden, Svalbard. Polar Res. 2002, 21 (1), 167–208. https://doi.org/10.1111/j.1751-8369.2002.tb00073.x.

(16) Hodal, H.; Falk-Petersen, S.; Hop, H.; Kristiansen, S.; Reigstad, M. Spring Bloom Dynamics in Kongsfjorden, Svalbard: Nutrients, Phytoplankton, Protozoans and Primary Production. Polar Biol. 2012, 35 (2), 191–203. https://doi.org/10.1007/s00300-011-1053-7.

(17) Krawczyk, D. W.; Witkowski, A.; Juul-Pedersen, T.; Arendt, K. E.; Mortensen, J.; Rysgaard, S. Microplankton Succession in a SW Greenland Tidewater Glacial Fjord Influenced by Coastal Inflows and Run-off from the Greenland Ice Sheet. Polar Biol.

2015, 38 (9), 1515–1533. https://doi.org/10.1007/s00300-015-1715-y.

(18) Hegseth, E. N.; Tverberg, V. Effect of Atlantic Water Inflow on Timing of the Phytoplankton Spring Bloom in a High Arctic Fjord (Kongsfjorden, Svalbard). J. Mar. Syst. 2013, 113–114, 94–105. https://doi.org/https://doi.org/10.1016/j.jmarsys.2013.01.003.

(19) Holding, J. M.; Markager, S.; Juul-Pedersen, T.; Paulsen, M. L.; Møller, E. F.; Meire, L.; Sejr, M. K. Seasonal and Spatial Patterns of Primary Production in a High-Latitude Fjord Affected by Greenland Ice Sheet Run-Off. Biogeosciences 2019. https://doi.org/10.5194/bg-16-3777-2019.

---

## Referee Comment (RC2) · Anonymous Referee #2 · 7 Jan 2021

Subglacial upwelling in winter/spring increases under-ice primary production

Summary: This paper aims to explore the role of the release of subglacial meltwater in the winter and spring on under-ice primary production. The premise of the study is that though subglacial upwelling of nutrient rich deep marine waters has been shown to be a viable mechanism for stimulating primary production in the summer, very few studies have examined this topic with regards to spring under-ice primary production. The study is an interesting, under-explored topic, which is only likely to become more important with global warming and prolonged glacial melt seasons, and thus, worthy of eventual publication in this journal.

However, I think there are number of improvements that could be made to aid the study, which I outline below. Apart from issues with over-interpretation of the data

(detailed below), the writing is often disorganized and unclear. Also, there often a lack of consideration of the on-ice processes that are occurring that could be affecting the authors interpretations – i.e. enrichment of the glacial meltwater itself that has been stored at the bed overwinter and is released in the spring. The fact that the submarine discharge in the spring is likely quite different to the dilute discharge characteristic of summer drainage is a fact that makes this difficult to compare to previous summer studies of glacial discharge into the ocean. To this end often the authors seem to have a pre-ordained conclusion – i.e. that the mechanism of nutrient addition was via upwelling of "deep" bottom waters by the submarine discharge, but this seemed at odd with the shallow depth of this discharge (20-m). Finally, there is also a lack of clarity with how some of the calculations are made – this needs to be rectified for these calculations to be understood. I would urge the authors to address these points, and indeed try to focus their story on the novel spring measurements they have, to maximize the potential readership of this interesting study.

Title: Given the confusion regarding subglacial upwelling (see below) – do you mean submarine discharge or upwelling of deeper marine waters? – I would suggest a title change.. Perhaps: Spring submarine discharge plumes fuel under-ice primary production ?

Abstract: L25: "retreat of tidewater glaciers could lead to decreased under-ice phytoplankton primary production" when? in the spring? In winter? Or both?

My comment on the line above points to a broader problem which is evident in the title.. which is that I think by the lack of specificity regarding the timing, winter or spring is determinantal to the paper. Presumably if the focus is on spring primary production then the authors are speaking about subglacial upwelling in the spring?

*A minor point, but line numbers every line would be really very helpful*

Introduction: L37: unclear what "it" is referring too L39: "close to the glacier front".. meaning what? Suggest specifying. Also a reference would be helpful here. The

ranges of increased primary production in front of tidewater glaciers is quite variable so specification would be good. L41: "at some distance" .. again suggest specifying here. L46: I'm not sure I would necessarily agree that the lack of studies of subglacial discharge in the winter / spring is due to the perception of a lack of freshwater outflow. I think it's well known from a glacier hydrological perspective that temperate and even polythermal ice masses likely have winter / spring discharge. More likely it's due to a lack of opportunity given the challenge of Arctic field conditions and the difficulty in locating such an outflow which would presumably be of low flux. L52-53: Suggest defining what you mean by "Glacier terminus melt rates" L54: Svalbard glaciers are shallower compared to what? L55-56: Phrase "can persist throughout winter and specifically in early spring" is unclear. Are you suggesting that outflow persists through winter and into spring? L57: add phrase " various other mechanisms such as:" between the words "through" and "constant". Also suggest making the part re: temperate parts of the glacier" a discrete sentence. Presumably, with regards to winter / spring discharge you are speaking about polythermal glaciers? I think this section in general needs more specifics regarding the types of glaciers that typically have winter/spring discharge and the typical fluxes and chemical composition of this discharge. I would think that all of these points are worth mentioning to set-up the discussion of this paper. The point regarding chemical composition in particular has been glossed over as being sourced from meltwater stored from the previous melt season but this meltwater having been stored at the bed over winter would have a significantly different chemical character than dilute snow and ice-melt passing quickly through the system at the height of summer. Also, what about the possibility of basal ice melt? L59-60: "Even low rates of subglacial outflows can be sufficient to supply nutrients to the surface".. why? How? Is it because they would be sufficiently deep enough in the water column? Are you speaking of supplying nutrients via upwelling or via direct addition of nutrients in the subglacial discharge itself? If only the former, how can the latter be discounted since subglacial discharge in the spring would likely be more chemically enriched from greater contact times with the glacier bed or being sourced from basal ice melt? L60:

[Figure]

Why would spring subglacial discharge contain less sediment.. b/c of the low fluxes? Suggest specifying why. L63: Suggest setting up this argument a bit more progressively. Explain first what nutrients are generally fueling the under-ice spring bloom initially, and then go into the timing of glacial discharge and how that might positively affect under-ice primary production. As of now, the timing of the discharge and the initial bloom and end of bloom period are all not clearly laid out and this is problematic (in my opinion). L67: delete "the" before "primary" and add "in front of tidewater glaciers" after the word "production" L70: Re-arrange /re-write sentence to: Once sufficient light penetrates the snow and ice layers, ice algae start growing within sea ice between March and April.... Etc" L73: "nutrient additions from the water column" .. via what? How? Suggest specifying. L74: "subglacial upwelling" .. does this refer to spring subglacial upwelling? Suggest specifying. Again, I find the timeline within the year confusing with regards to glacial meltwater discharge and effect on bloom dynamics. Suggest more clearly spelling all of this out above. L78: "or at the ice edge related to ice edge induced upwelling" .. can you define this upwelling without using the words "ice edge"? L79: suggest replacing "coverage also" with "accumulates" L81: suggest replacing "Once" with "After" L83: suggest replacing "related" with "induced" L86: suggest deleting "to" and replacing "fuel" with "fueling" L87: the word "slow" is curious .. why is the subglacial upwelling slow? How do you know it's slow vs fast or continuous vs intermittent? Suggest deleting this word as it opens up a range of topics that haven't been discussed in enough detail above to warrant the use of this adjective here. L86-88: This pivot in this last sentence doesn't make a lot of sense to me as it seems to not really address the points brought up by the sentences immediately preceding it... i.e. namely reduced algal biomass due to brackish ice conditions .. suggest rectifying this last sentence. L90-91: How are the 2 freshwater inputs different? Suggest specifying versus keeping your reader in the dark here.. L92: "to investigate the effect of the glacier terminus" .. this is a big vague. Suggest specifying. L94: "nutrient rich meltwater".. I'm unclear what you are referring to here.. presumably since this phrase is followed by "bottom water to the surface" I think by nutrient-rich meltwater you are

referring to the subglacial discharge being enriched itself in nutrients versus upwelling of bottom waters but this has not been addressed above (though I suggest doing so) L95: suggest adding "under ice" before the words "primary production" if this is indeed what you are referring too? L95: "near the glacier front".. phrase is vague. Suggest specifying. L95-96: "low permeability of sea ice" .. phrase is also vague. Suggest specifying.

As noted above I think the introduction would benefit from some more specificity, especially regarding the types of glaciers where winter / spring discharge might occur, a timeline of how this discharge evolves from end of the season to the winter and spring, and how this discharge might affect spring bloom under-ice dynamics – considering both the possibility of upwelling of bottom waters and also addition of nutrients directly from the glacial meltwater itself as alluded to in the last paragraph. One thing that should also be likely addressed is that any spring discharge will presumably be of quite low flux.. given this how likely / effective will any upwelling be?

Methods: L120: ".. were melted in 50% vol/vol sterile filtered seawater..." what was the reasoning for this? L155-157: Estimates of bottom water fractional contributions based on conservative mixing of nitrate.. can you rule out nitrate addition from the glacial meltwater itself? Other studies have found this (see, Beaton et al., 2017 in ES&T: https://pubs.acs.org/doi/abs/10.1021/acs.est.7b03121), especially in the early season meltwater from a distributed subglacial drainage system. L215: I'm confused by the words "reciprocal transplant experiment" .. I don't think a "transplant experiment" is described above.. just primary production incubations. I also find the description of this experiment (L215-218) unclear and thus the overall purpose of the experiments to the study also unclear. As written, I cannot assess these experiments so I'd suggest a re-write of this paragraph. L225: Unclear what map you are referring to in sentence starting with "The map.." L232: I'm wondering why you chose to you swarm to cluster versus amplicon sequence variants (see Callahan et al., 2017: https://www.nature.com/articles/ismej2017119) L235: Was the data transformed in anyway before making the dissimilarity matrix? I'm only asking because it seems doing some type of transformation (e.g. Hellinger) is increasingly common.

Results: L243: replace "were having" with "had" L244: why is Fig 2 c, d referenced before Fig 2 a, b.. did I miss the reference to a, b somewhere? L265: Are there any photos of the subglacial outflow described in L267-268? Since there is a lack of field data at this time of year I think that these would be of value. L283: When reading about the very high nitrate+nitrite and silicate concentrations below the ice at SG I found myself really wondering if this could be coming from the subglacial meltwater itself versus upwelling of deeper marine waters. I believe you have data of the glacial meltwater itself? You mention these samples in lines 101-102 .. and I see further on that you present this data in L295. I'd suggest re-organizing so that this comes before the marine data. L295: missing units for silicate in the outflow water L300: The definition of conservative mixing is not quite right. The sentences in lines 300-302 are especially problematic. I see that the other reviewer has already adequately commented on this so I will defer to those comments. In the rest of the paragraph I would avoid the words "positive mixing patterns" and "positive relations". I also found the color scheme in Fig 5 (red and pink) challenging to interpretation. L310: I echo the other reviewer that these calculations of nutrients supplied via upwelling vs the glacial meltwater should be shown.. how were these calculated? What is the error on these calculations? This paragraph needs more explanation for these values to be believed especially considering (as pointed out by the other reviewer) the single outlier values that are driving the gradient in SG samples. Also, at SG, it seems, at least from Fig 5 d-f, that the lower salinity water had higher silicate concentrations but these concentrations were much higher than those reported for the glacial meltwater above. What is the source of this silicate? L333: Like the other reviewer I'm confused by the term "vertical export of Chl" – what it means, how it was estimated, and what the errors on this estimate are. L337: "assuming absence of grazing".. this doesn't really seem realistic? L348: I'd suggest explaining more fully again the goal of the "reciprocal transplant experiment" before giving the results. Fig 6: The quality of this

figure should be improved. The numbers in the parentheses are very difficult to read. Fig 7: The x-axis with the experiment name are not clear. What does "com" stand for? Fig 8: Define UIW in the legend as you have for the other abbreviations L355-356: "The first [NMDS1] axis separated sea ice from water communities with no overlapping samples".. this really isn't evident in Fig 8a.. sea ice is the square and what water and under ice water samples are the triangles. These regularly are in the same ellipses, unless I'm missing something? Also, is the glacier outflow sample actually a under ice water sample? What is the salinity of this sample? I guess I'm wondering if this is a true non-marine glacial outflow sample or one that could be diluted by marine water? I think this is an important point that needs to be clarified above. L358-360: What was the stress on this NMDS? How robust is this ordination you show? I'm always weary of interpreting the axes in this manner, i.e. axes one shows X and axes 2 shows Y .. i.e. similar to how one might view a PCA. I agree that looking at Fig 8a your communities are different but I don't think you can go as far to say that axis 1 is separating ice vs water and axis 2 is separating glacial vs marine. The ordination of this NMDS would likely change each time you ran it.. maybe something to consider? L371: "Overall the same NMDS clustering has been found as for the 16S rRNA sequencing" .. but in the 18S plot (Fig 8b) no ellipses are drawn.. does this indicate that these group divisions were not significant? The written text doesn't seem to match the figure. Fig8c – the separation in the samples is quite striking on this NMDS. How come there are no ellipses on this plot? Were the differences shown in the NMDS not significant? Could try a perMANOVA to test the significance of differences between the groups perhaps?

Discussion: L388-391: These first few lines are a great summary and really the abstract and introduction needs to be better set-up to frame these important points: (1) evidence for subglacial upwelling at a shallow tidewater glacier under sea ice and (2) that this upwelling persists in the winter / spring and supplies nutrient-rich glacial meltwater and upwelling of bottom water. . . I actually think part of the confusion is the use of the term "upwelling" to describe the release of submarine discharge into the ocean and also the upwelling of bottom water. Perhaps a change of language throughout would be helpful

– i.e. saying "submarine discharge" vs "subglacial upwelling". And as per points above the case about nutrient-rich glacial meltwater needs to be set-up and made earlier as it's really a central finding. L406: The phrasing "which does not allow basal glacial ice to melt" is unclear. The whole sentence is too long and should be made into 2, but are the authors saying that because there is not Atlantic inflow water there can be no basal ice melt? Basal ice melt can result from geothermal heat flux, overburden ice pressure, and sliding friction. Warm ocean water is not the only mechanism. I suggest looking at a textbook (e.g. the physics of glaciers) and reviews on this topic: e.g. Hubbard and Sharp, 1989 L407: "Subglacial meltwater itself is unlikely to lead to basal ice melting due to its low salinity". This sentence is very unclear to me. I'm not sure what this sentence is saying or trying to say. L407-408: "However, basal ice melt is likely more important in systems with Atlantic water inflows..." as per above this seems to ignore the possibility of basal ice melt underneath temperate and polythermal glaciers. This may not be what the authors mean but as written it reads this way. L420: "remains from the previous melting season" is unclear. Can you specify what you mean by remains. L433: Can you specify what data you are referring to when you say "estimated bacterial growth rates". I searched for this term in the paper and did not see it previously defined. It really should be so that the basis for this calculation of doubling time is clear. L442: Why does the supply have to be "constant" ? It seems like (from the methods) that samples for community analyses were only taken once at each station? How does a single-time point sample give an indication of the timescale of submarine discharge into the fjord? This might be a bit of a reach based on the community data alone – suggest tempering this statement. L442-444: When you say the "southern part of the glacier" is this part on land or in the ocean? If it's on land you should specify. I also think that this assumption that this outflow is being released under the marine-terminating portion can be backed up by your marine data? This sentence seems out of place here. L445- to end of paragraph: This explanation of glacier hydrology really needs to come earlier. As written this whole section on the potential magnitude of upwelling is poorly organized. Suggest first setting it up by talking about processes on the ice and then

what's happening in the ocean. L456: "Our mixing calculations estimate".. where are these calculations described? L457: At what depth is the submarine discharge exiting the glacier? I find myself wondering at what depth these different water masses occur (can you specify this) and how deep the DLAW is being entrained from? Is it sufficiently below the nutricline to be replete in nutrients? Also the calculated entrainment factor of 1.6, how was this calculated exactly? And you state "which pulled 1.6 times more DLAW" .. more than what? This is not clear. L458-459: "Fransson et al. (2020) found that 30-60% of glacier derived meltwater was incorporated in the bottom sea ice . . . again indicating that it is a widespread process at marine terminating glacier fronts" .. what is a widespread process? The release of submarine discharge and its incorporation into bottom sea ice OR the entrainment of different water masses (i.e. DLAW) as the plume rises (as discussed in the previous sentence). Again, this is a case in point of the organizational structure and lack of specificity of terms "submarine discharge" vs "upwelling of bottom waters" to be a source of confusion. L461: "Compared to the massive subglacial plumes of summer systems" .. where? This should be specified .. different glaciers have widely different discharge fluxes. The citation seems to be from Greenland but these glaciers will bear little resemblance to Svalbard, perhaps citing summer discharge fluxes from Svalbard glaciers too would be useful – particularly from your study site if the intent of this sentence is to contrast with spring discharge fluxes as seems to be the case. L462: "subglacial upwelling in spring is a small volume transport".. where is this data from? This study? This should be explicitly stated. Suggest re-writing this entire sentence. Also, the last part of the sentence regarding upwelling needed to maintain primary production should be a new sentence as this is a different point then the discharge flux. L464: "This careful estimate".. I'd remove the word "careful".. the more so because the sentence before this one is unclear! Is this estimate of freshwater input for Billefjorden in the summer or spring? It's unclear. The estimate from the Halbach paper is I believe from the summer so you want to make sure you are comparing like with like. L465-466: The fact that you have less entrainment than the Hopwood study is really not surprising at all considering the depth of

discharge and flux of discharge at the much deeper, larger glaciers in that study. I'm not sure what the purpose is of this statement? As written now it's failing to provide relevance to this study. L466-467: "each volume of SGO water pulled about the same volume of DLAW with it to surface".. this is unclear.. do you mean each volume over a certain timeframe (a day? A week? A month?) .. what is the volume exactly? What was the volume of DLAW entrained? This should be stated if you are speaking about volumes here. And again the comparisons to the Hopwood study don't' seem relevant if you are comparing to large Greenland glaciers. You should specify where and what type of glaciers in the Hopwood review you are comparing too. L470: This is the first mention of the depth of the discharge. As you say, 20-m is quite shallow. Are nutrient concentrations sufficiently high enough here to augment surface concentrations? In other words, is this depth below the nutricline. L473-to end of paragraph: This seems to directly contradict previous statements regarding the glacial meltwater discharge being enriched in nutrients (e.g. silicate?). Also many of the comparisons you are making are to summer discharge fluxes and summer entrainments.. the spring discharge will of course be lower but more chemically enriched from the glacial meltwater discharge? I think if you are going to use the summer values to compare, which you might have to do out of necessity and lack of other comparisons, you need to state so explicitly, and the limitations of such comparisons. L480: The word "Surprisingly" seems to not be the right word choice here. L438: "Substantial subglacial upwelling" .. I'm unclear was to what you are referring to here – is this submarine discharge of glacial meltwater or upwelling of bottom waters? In either case the word "substantial" seems ill-advised here given the preceding discussion and should be removed. Could it be that you didn't observe much light limitation because the plumes were not that "massive" (compared to summer).. i.e. you just have a much smaller discharge flux and therefore plume in the spring? This seems likely and unsurprising. L485-86: Unclear what the phrase "where light is not considered limiting" is referring too. Line 511: "rations" should be "ratios"? L515: Can you really call it "deep water upwelling" if the water is being entrained from only 20-m? This is problematic (at least for me) and needs to be clearly

addressed I think. L517-519: The discussion on iron seems unrelated and as written is unconvincing. L520: "nutrient concentrations may simply be higher due to the shallower depth at SG" .. why? It's unclear what you are trying to say. Suggest re-writing with more detail and explicity. L529: Was the Frasson study done at this same site? L530: "The values" .. vague.. specify what kind of values you are referring to. L535: Paragraph ending here is rambling and needs to be re-written. Suggest taking out the iron since you have no data on this to compare. L536: "related".. what do you mean by this word? Specify. L538: Were are you proposing this nitrification is occurring? In the ocean or in the glacial meltwater? Could the high nitrate come from the subglacial waters itself? See papers by Beaton et al. in Greenland, Jemma Wadham, Boyd et al., 2011 (AEM) and Wynn et al., 2007 (Chemical Geology). Do you have measurements of the outflow un-diluted by seawater so you can rule this possibility out? L566: Were you able to resolve any low-light level species in your molecular community composition data to back this statement up? L581: "their" .. unclear what this is referring to. L646: "In winter and spring, this would result in the lack of subglacial upwelling".. but with more melt there would be longer melt seasons and presumably more submarine discharge and associated upwelling – at least in the shorter term?

---

## Author Comment (AC2) · 25 Jan 2021

We want to thank the reviewer sincerely for the thorough and very helpful review. Weaddressed all comments as described in the attached document. We believe that thechanges improved the manuscript considerably.

In cases of over-interpretations, we either rephrased the interpretation more careful, often via adding details for clarification, or removed the statements (details in the supplement). We rewrote the sections that the reviewer considered disorganized and unclear (details in the supplement) with the most substantial changes regarding glacial processes and the chapter about subglacial upwelling and entrainment factors. We tried to clarify the relevance of on-ice processes by i) introducing the processes in more

detail in the introduction, ii) mentioning the nutrient concentrations of the undiluted subglacial meltwater that we measured earlier in the results, and iii) giving more references to the role of nutrient enrichment under the glacier (weathering during bedrock contact, solute expulsion during freezing). However, our nutrient measurements of the undiluted meltwater still showed lower concentrations compared to the fjord bottom water. The concentrations are enriched compared to sea ice and UIW samples at NG and IE, but we consider upwelling of bottom water more important for nutrient dynamics in this area. We further clarified these findings by referring more strongly to the undiluted meltwater nutrient concentrations in the text. Please note that Svalbard studies by van der Poll (e.g. van der Poll et al., 2018) agree with our conclusion. Referee 2 suggests that shallow water depth might limit the relevance of this process. We suggest that the freshwater input occurred below the sharp halocline in 4-5m depth, explaining the nutrient differences between 15 and 1 m. Additionally this process is supported through a) the absence of any substantial external advection of inorganic nutrients (e.g through tides and wind), and b) strong salinity driven stratification preventing mixing apart from upwelling. Detailed calculations were added to the text or supplement.

We used following font colors and highlighting in the attached document: - Grey text:Reviewers' comments - Black text: Our response - green: Changes in the manuscript aftger RW2 - Yellow highlights: Changes in the manuscript based on RW1.

Please also note the supplement to this comment:
https://tc.copernicus.org/preprints/tc-2020-326/tc-2020-326-AC2-supplement.pdf

**Supplement:**

Subglacial upwelling in winter/spring increases under-ice primary production

Summary: This paper aims to explore the role of the release of subglacial meltwater in the winter and spring on under-ice primary production. The premise of the study is that though subglacial upwelling of nutrient rich deep marine waters has been shown to be a viable mechanism for stimulating primary production in the summer, very few studies have examined this topic with regards to spring under-ice primary production. The study is an interesting, under-explored topic, which is only likely to become more important with global warming and prolonged glacial melt seasons, and thus, worthy of eventual publication in this journal.

We want to thank the reviewer sincerely for the comprehensive review that helps to improve the clarity of the manuscript considerably. All comments are clear and very useful. We addressed all comments as outlined in detail below. Suggested changes in the text of the manuscript are highlighted in green.

However, I think there are number of improvements that could be made to aid the study, which I outline below. Apart from issues with over-interpretation of the data (detailed below), the writing is often disorganized and unclear. Also, there often a lack of consideration of the on-ice processes that are occurring that could be affecting the authors interpretations – i.e. enrichment of the glacial meltwater itself that has been stored at the bed overwinter and is released in the spring. The fact that the submarine discharge in the spring is likely quite different to the dilute discharge characteristic of summer drainage is a fact that makes this difficult to compare to previous summer studies of glacial discharge into the ocean. To this end often the authors seem to have a pre-ordained conclusion – i.e. that the mechanism of nutrient addition was via upwelling of "deep" bottom waters by the submarine discharge, but this seemed at odd with the shallow depth of this discharge (20-m). Finally, there is also a lack of clarity with how some of the calculations are made – this needs to be rectified for these calculations to be understood. I would urge the authors to address these points, and indeed try to focus their story on the novel spring measurements they have, to maximize the potential readership of this interesting study.

In cases of over-interpretations, we either rephrased the interpretation more careful, often via adding details for clarification, or removed the statements (details below). We rewrote the sections that the reviewer considered disorganized and unclear (details below) with the most substantial changes regarding glacial processes and the chapter about subglacial upwelling and entrainment factors. We tried to clarify the relevance of on-ice processes by i) introducing the processes in more detail in the introduction, ii) mentioning the nutrient concentrations of the undiluted subglacial meltwater that we measured earlier in the results, and iii) giving more references to the role of nutrient enrichment under the glacier (weathering during bedrock contact, solute expulsion during freezing). However, our nutrient measurements of the undiluted meltwater still showed lower concentrations compared to the fjord bottom water. The concentrations are enriched compared to sea ice and UIW samples at NG and IE, but we consider upwelling of bottom water more important for nutrient dynamics in this area. We further clarified these findings by referring more strongly to the undiluted meltwater nutrient concentrations in the text. Please note that Svalbard studies by van der Poll (e.g. van der Poll et al., 2018) agree with our conclusion. Referee 2 suggests that shallow water depth might limit the relevance of this process. We suggest that the freshwater input occurred below the sharp halocline in 4-5m depth, explaining the nutrient differences between 15 and 1 m. Additionally this process is supported through a) the absence of any substantial external advection of inorganic nutrients (e.g through tides and wind), and b) strong salinity driven stratification preventing mixing apart from upwelling. Detailed calculations were added to the text or supplement.

Title: Given the confusion regarding subglacial upwelling (see below) – do you mean submarine discharge or upwelling of deeper marine waters? – I would suggest a title change.. Perhaps: Spring submarine discharge plumes fuel under-ice primary production ?

This has been a very good suggestion by the referee – we agree and changed the title accordingly to "==Early spring submarine discharge plumes fuel under-ice primary production==""

Abstract:

L25: "retreat of tidewater glaciers could lead to decreased under-ice phytoplankton primary production" when? in the spring? In winter? Or both? My comment on the line above points to a broader problem which is evident in the title.. which is that I think by the lack of specificity regarding the timing, winter or spring is determinantal to the paper. Presumably if the focus is on spring primary production then the authors are speaking about subglacial upwelling in the spring?

Based on the simple date definition spring starts at the 20th of March. However, the definition of winter and spring is more difficult in Arctic studies, as biological processes like algal blooms are not tight to the calendar but to changes in e.g. light and ice regime. Also algal growth (as indicator of spring) in the ice might occur prior to algal growth in the water column. Spring may also be defined as the onset of snowmelt and temperatures above freezing point (mostly in terrestrial ecology), or by the return of light. Since we sampled at a time of subzero temperatures and ice cover (winter), but with sufficient light for algae blooms (spring), we had used the term "winter/spring" in the submitted version. However, since light availabililty is often most important in Arctic marine systems to define the onset of spring we changed the term to "==early spring==" throughout the manuscript and added the information where it was lacking (including L25).

*A minor point, but line numbers every line would be really very helpful*

We added the line numbers as suggested.

Introduction:

L37: unclear what "it" is referring too

We replaced "it" with "==the submarine discharge==""

L39: "close to the glacier front".. meaning what? Suggest specifying. Also a reference would be helpful here. The ranges of increased primary production in front of tidewater glaciers is quite variable so specification would be good.

The exact distance is highly variable, depending on sediment load, glacier terminus depth, discharge volume and flux e.g.. Hence, it is not possible to provide accurate numbers. However, based on an earlier study (Halbach et al., 2019) which found a phytoplankton bloom at 0.1 -1.9km distance from the glacier, we included a distance range into the manuscript ==(hundreds of meters to kilometers)==.

L41: "at some distance" .. again suggest specifying here.

See comment above

L46: I'm not sure I would necessarily agree that the lack of studies of subglacial discharge in the winter / spring is due to the perception of a lack of freshwater outflow. I think it's well known from a glacier hydrological perspective that temperate and even polythermal ice masses likely have winter / spring discharge. More likely it's due to a lack of opportunity given the challenge of Arctic field conditions and the difficulty in locating such an outflow which would presumably be of low flux.

We agree and changed this statement in the following way: " ==Due to the challenges of Arctic field work in early spring and the difficulties of locating such an outflow, only few studies investigated submarine discharge during that time window. The few studies available suggest overall little discharge (e.g. Fransson et al., 2020, Schaffer et al., 2020) compared to summer values. The limited amount of data makes the generalized quantification of subglacial outflow difficult. In addition, studies focusing on the potential impacts of the early spring discharge on sea ice and pelagic primary production are lacking.==""

L52-53: Suggest defining what you mean by "Glacier terminus melt rates"

The term "glacier terminus melt rate" is adopted from the mentioned publications, but we added a short clarification. "Glacier terminus melt rate occurring at the glacier-marine interface".

L54: Svalbard glaciers are shallower compared to what?

They (the water depth at the glacier terminus) are shallower than Greenland glaciers. We added following clarification: "submarine glacier termina on Svalbard occur typically at shallower water depths than on Greenland …"

L55-56: Phrase "can persist throughout winter and specifically in early spring" is unclear. Are you suggesting that outflow persists through winter and into spring?

We included the suggested sentence by the referee and rephrased the sentence in the following way: "can persist through winter and into spring"

L57: add phrase " various other mechanisms such as:" between the words "through" and "constant". Also suggest making the part re: temperate parts of the glacier" a discrete sentence. Presumably, with regards to winter / spring discharge you are speaking about polythermal glaciers? I think this section in general needs more specifics regarding the types of glaciers that typically have winter/spring discharge and the typical fluxes and chemical composition of this discharge. I would think that all of these points are worth mentioning to set-up the discussion of this paper. The point regarding chemical composition in particular has been glossed over as being sourced from meltwater stored from the previous melt season but this meltwater having been stored at the bed over winter would have a significantly different chemical character than dilute snow and ice-melt passing quickly through the system at the height of summer. Also, what about the possibility of basal ice melt?

We replaced the sentence with a more comprehensive paragraph addressing the missing information and background: "Hodgkins (1997) described the release of subglacial meltwater stored from the previous summer melt season from various Svalbard glaciers, including cold-based glaciers. Winter drainage occurred mostly periodically during events of ice-dam breakage. During the storage period, the meltwater can change its chemical composition. During prolonged contact with silicon-rich bedrock, the meltwater becomes enriched in the macronutrient silicate (Hodgkins, 1997). Additionally during freezing of the meltwater, solutes are expelled leading to higher ion concentrations in the liquid fraction (Hodgkins, 1997). Under polythermal glaciers, various other mechanisms such as constant supply from groundwater, and basal ice melt via geothermal heat, pressure, or frictional dissipation can also be a continuous, but low flux meltwater source in winter (Schoof et al., 2014)."

L59-60: "Even low rates of subglacial outflows can be sufficient to supply nutrients to the surface".. why? How? Is it because they would be sufficiently deep enough in the water column? Are you speaking of supplying nutrients via upwelling or via direct addition of nutrients in the subglacial discharge itself? If only the former, how can the latter be discounted since subglacial discharge in the spring would likely be more chemically enriched from greater contact times with the glacier bed or being sourced from basal ice melt?

We suggest that low supply rates via upwelling can have a considerable impact due to the absence of other sources in a sea ice covered fjord with very weak advection (tidal currents, wind, Atlantic water) and a strongly stratified water column. Direct addition can of course also play a role. We added the following clarification: "We hypothesis that subglacial discharge can lead to significantly increased primary production, due to upwelling of nutrient rich deeper water or through its own nutrient load, especially towards the end …"

L60: Why would spring subglacial discharge contain less sediment.. b/c of the low fluxes? Suggest specifying why.

The referee is correct in his/her suggestion. The reduction is likely caused by the low fluxes and thereby reduced advective forcing. We added a reference to a study measuring the seasonal variation of sediment outputs at a Svalbard tidewater glacier as additional support as described in the response to RW1 (Moskalik et al., 2018) and added a specification of "due to lower fluxes".

L63: Suggest setting up this argument a bit more progressively. Explain first what nutrients are generally fueling the under-ice spring bloom initially, and then go into the timing of glacial discharge and how that might positively affect under-ice primary production. As of now, the timing of the discharge and the initial bloom and end of bloom period are all not clearly laid out and this is problematic (in my opinion).

We suggest following additions: "After light becomes available in spring, ice algae and phytoplankton may start forming blooms fueled by nutrients supplied via winter mixing with different onsets in different parts of the Arctic. The blooms are typically terminated by limitation by macronutrients, either nitrate or silicate (Leu et al., 2015). We suggest that in the absence of wind induced mixing due to the seasonal presence of fast ice cover in spring, submarine discharge of glacial meltwater can directly (ion enrichment over the subglacial storage period) or indirectly (upwelling) be a significant source of inorganic nutrients significantly increasing primary production after nutrients supplied via winter mixing are incoporated into algal biomass."

L67: delete "the" before "primary" and add "in front of tidewater glaciers" after the word "production"

We changed the sentence accordingly.

L70: Re-arrange /re-write sentence to: Once sufficient light penetrates the snow and ice layers, ice algae start growing within sea ice between March and April: : :. Etc"

We changed the sentence accordingly

L73: "nutrient additions from the water column" .. via what? How? Suggest specifying.

We suggest replacing "nutrient addition" with "advection of nutrient-rich seawater" for clarification

L74: "subglacial upwelling" .. does this refer to spring subglacial upwelling? Suggest specifying. Again, I find the timeline within the year confusing with regards to glacial meltwater discharge and effect on bloom dynamics. Suggest more clearly spelling all of this out above.

Yes, we refer to spring. We add the term "early spring" for clarification.

L78: "or at the ice edge related to ice edge induced upwelling" .. can you define this upwelling without using the words "ice edge"?

We suggest using the term "wind-induced Ekman upwelling" as described by Mundy et al. (2009).

L79: suggest replacing "coverage also" with "accumulates"

We replaced "coverage also" with "accumulation"

L81: suggest replacing "Once" with "After"

We change the term "Once" with "After" as suggested.

L83: suggest replacing "related" with "induced"

We change the term "related" with "induced" as suggested.

L86: suggest deleting "to" and replacing "fuel" with "fueling"

We change the formulation "to fuel" with "fueling" as suggested.

L87: the word "slow" is curious .. why is the subglacial upwelling slow? How do you know it's slow vs fast or continuous vs intermittent? Suggest deleting this word as it opens up a range of topics that haven't been discussed in enough detail above to warrant the use of this adjective here.

We suggest replacing "slow" with "of low total flux", which would include continuous and intermittent discharge.

L86- 88: This pivot in this last sentence doesn't make a lot of sense to me as it seems to not really address the points brought up by the sentences immediately preceding it: : : i.e. namely reduced algal biomass due to brackish ice conditions .. suggest rectifying this last sentence.

We agree with the referee to change this section. We suggest removing the last part of the sentence "…and cause different succession patterns for phytoplankton and sea ice algae." Since the succession patters are not clearly introduced or explained and not a major objective of the paper.

L90-91: How are the 2 freshwater inputs different? Suggest specifying versus keeping your reader in the dark here.

We suggest replacing "with different freshwater inputs" with "with only one glacier front supplying submarine freshwater discharge". We agree that the previous formulation is unclear and misleading, since we mostly argue for the absence of freshwater inputs at NG.

L92: "to investigate the effect of the glacier terminus" .. this is a big vague. Suggest specifying.

We suggest adding following details: "… to investigate the effect of the glacier terminus, and subglacial outflow related upwelling on the light and nutrient regime in the fjord and thereby on early spring primary productivity…"

L94: "nutrient rich meltwater".. I'm unclear what you are referring to here.. presumably since this phrase is followed by "bottom water to the surface" I think by nutrient-rich meltwater you are referring to the subglacial discharge being enriched itself in nutrients versus upwelling of bottom waters but this has not been addressed above (though I suggest doing so)

We refer to the meltwater coming from the glacier itself. We suggest following clarification: "nutrient rich glacial meltwater" and "upwelling of marine bottom water"

L95: suggest adding "under ice" before the words "primary production" if this is indeed what you are referring too?

We added the formulation "under ice" as suggested.

L95: "near the glacier front".. phrase is vague. Suggest specifying.

We suggest adding a distance estimate in the following way: "near (<500 m) the glacier front".

L95-96: "low permeability of sea ice" .. phrase is also vague. Suggest specifying. As noted above I think the introduction would benefit from some more specificity, especially regarding the types of glaciers where winter / spring discharge might occur, a timeline of how this discharge evolves from end of the season to the winter and spring, and how this discharge might affect spring bloom under-ice dynamics – considering both the possibility of upwelling of bottom waters and also addition of nutrients directly from the glacial meltwater itself as alluded to in the last paragraph. One thing that should also be likely addressed is that any spring discharge will presumably be of quite low flux.. given this how likely / effective will any upwelling be?

We suggest adding following specification: "as a result of low permeability sea ice limiting nutrient exchange and inhabitable space"

As mentioned above (RW comment on L57 and L63), we also added a more detailed introduction of the potential discharge of different glacier types and the chemical characteristics of fresh vs stored subglacial meltwater with a potential of direct nutrient input with the meltwater. We also added the statement of low fluxes in spring as mentioned above (RW comment on 87). We believe we explained the role of lower salinity waters for forming less permeable sea ice already in former lines 84ff. We added the following sentence to line 85: , 1999). Sea ice with reduced bulk salinity has a reduced permeability to more saline ice at identitical temperatures (Golden et al. 1998).

Reference: Golden KM, Ackley SF, Lytle VI (1998) The percolation phase transition in sea ice. Science 282:2238-2241

Methods:

L120: ".. were melted in 50% vol/vol sterile filtered seawater: : :" what was the reasoning for this?

Sea ice is commonly melted in 50% vol/vol sterile seawater in order to avoid osmotic shock. Since most sea ice organisms live in the brine channels with high salinity, but the salinity of a melted bulk ice core is very low, direct melting leads to osmolysis. We added following sentence for clarification: "…to avoid osmotic shock of cells (Garrison and Buck 1986)"

L155-157: Estimates of bottom water fractional contributions based on conservative mixing of nitrate.. can you rule out nitrate addition from the glacial meltwater itself? Other studies have found this (see, Beaton et al., 2017 in ES&T: https://pubs.acs.org/doi/abs/10.1021/acs.est.7b03121), especially in the early season meltwater from a distributed subglacial drainage system.

We realize that our formulation was not clear. We also measured NOx concentrations from the subglacial outflow itself. We found subglacial outflow water exiting the glacier and sampled it directly (Salinity 0). The nutrient values of the glacial outflow, bottom water, and surface water were used for the calculations. We added following clarification in the methods text: "assuming linear mixing of subglacial meltwater, bottom water (at station IE) and surface water concentration using the NOX concentration measured at IE (Table 1).

As mentioned by RW1 we added details and equations on how the mixing calculations were done.

In the manuscript we added the equations to the supplement, we added the error estimates in Table 1, and we added details about the different water types in the header of Table 1.

Here the response to RW1 which outlines our changes:

"We suggest adding following calculations to the supplement. The mentioned outlier values in SG in the UIW sample was not used for the mixing calculations as explained above. For the meltwater fraction at the surface the error related to the average IE salinity is less than 0.1 % (see comment above), the main variation of the % meltwater contribution in the surface layer of SG is related to the salinity at the surface of SG (Fig. R1). We added the error estimate of 0.1 % to the table. For nutrients, the error was estimated based on the variability in the concentrations measured in the triplicates. For NOx the estimated range of contribution by upwelling is thereby 57-59 % (± 1 %) bottom water, for Silicate 89-95 % (± 3 %), and for phosphate 46-49 % (± 3 %).

Equations. Mixing calculations for estimates of the fraction of meltwater ($MW_{Sal}$) based on salinity, and for bottom water based on nutrient concentrations ($BW_{Nuts}$). Sal indicates the average salinities measured at the IE ($Sal_{IE}$), SG at 1m depth ($Sal_{SG1m}$), subglacial outflow ($Sal_{glac}$). Nut indicates the nutrient concentrations of nitrate and nitrite ($NO_X$), silicate (Si), and phosphate (PO4) at 1m under the sea ice at SG ($Nut_{1mSG}$) and IE ($Nut_{1mIE}$), the bottom water of the IE ($Nut_{BW}$), or subglacial outflow water ($Nut_{glac}$).

$$MW_{Sal}[\%] = \frac{Sal_{IE} - Sal_{SG1m}}{Sal_{SG1m} - Sal_{glac} + Sal_{IE} - Sal_{SG1m}} * 100$$

$$MW_{Sal}[\%] = \frac{34.7\ PSU - 23.6\ PSU}{23.6\ PSU - 0\ PSU + 34.7\ PSU - 23.6 PSU} * 100 = 32\ \%$$

$$BW_{Nut}[\%] = \frac{Nut_{1mSG} - MW_{Sal}[\%] * Nut_{glac} - Nut_{1m_{IE}} + MW_{Sal}[\%] * Nut_{1m_{IE}}}{Nut_{BW} - Nut_{1m_{IE}}} * 100$$

$$BW_{NOX}[\%] = \frac{6.52\mu M - 0.32 * 2.06\ \mu M - 3.27\ \mu M + 0.32 * 3.27\ \mu M}{9.57\ \mu M - 3.27\ \mu M} * 100 = 58\ \%$$

$$BW_{Si}[\%] = \frac{4.30\ \mu M - 0.32 * 1.79\ \mu M - 1.59\ \mu M + 0.32 * 1.59\ \mu M}{4.46\ \mu M - 1.59\ \mu M} * 100 = 92\ \%$$

$$BW_{PO4}[\%] = \frac{0.41\ \mu M - 0.32 * 0.09\ \mu M - 0.34\ \mu M + 0.32 * 0.34\ \mu M}{0.67\ \mu M - 0.34\ \mu M} * 100 = 46\ \%$$

"

Change in Table 1:

Table 1. Properties of 1) marine surface and 2) Marine deep water (both station IE), 3) subglacial discharge melt water and 4) station SG surface water and the relative contribution of the water types 1 to 3 to form water type 4. The calculations are given in the Supplement and are based on different salinities and nutrients in the 4 water masses.

| | 1) Surface water (IE 1m) | | 2) Bottom water (IE) | | 3) Subglacial discharge Meltwater | | 4) SG (1 m) |
|---|---|---|---|---|---|---|---|
| Salinity [PSU] | 34.7 | | 34.7 | | 0 | 32 ± 0.1 % | 23.6 |
| Temperature [°C] | -1.4 | | -1.4 | | 0 | | -0.4 |
| Silicate [µmol L⁻¹] | 1.59 | 0 % | 4.46 | > 84 % | 1.79 | 32 % | 4.30 |
| NO$_X$ [µmol L⁻¹] | 3.27 | 10 ± 3 % | 9.57 | 58 ± 1 % | 2.06 | 32 % | 6.52 |
| Phosphate [µmol L⁻¹] | 0.34 | 19 ± 3 % | 0.67 | 49 ± 3 % | 0.09 | 32 % | 0.42 |

L215: I'm confused by the words "reciprocal transplant experiment" .. I don't think a "transplant experiment" is described above.. just primary production incubations. I also find the description of this experiment (L215-218) unclear and thus the overall purpose of the experiments to the study also unclear. As written, I cannot assess these experiments so I'd suggest a re-write of this paragraph.

The words "reciprocal transplant experiment" is mostly used in plant ecology, when plants are planted/grown on different soil/ environments in order to see if the different soil/ environment has an effect on their fitness or growth. We did an analogue experiment in which we incubated algae communities in different water/ environments in order to test if the water chemistry has an effect on algae growth. We considered other more descriptive terms such as "water exchange experiment", but

prefer keeping the term "reciprocal transplant experiment" due to its established and wide use in ecology. We rewrote the paragraph to clarify the experimental design in the following way:

"For testing the effect of the water chemistry on phytoplankton growth, we designed a reciprocal transplant experiment where the phytoplankton communities at SG and IE (1 m and 15 m) were transplanted into the water of both SG and IE. 50 ml of the water containing the phytoplankton community were transferred into 50 ml sterile filtered (0.2 μm) seawater of SG or IE in 100 ml polyethylene bottles. The bottles were then incubated in situ at the appropriate depth and primary production measured as described above. The aim of the experiment is to test if water chemistry alone is sufficient to increase primary production, or if the different communities, light regimes, or temperatures are more important."

L225: Unclear what map you are referring to in sentence starting with "The map.."

We refer to the map in Figure 1 and added the figure reference. (Fig. 1)

L232: I'm wondering why you chose to you swarm to cluster versus amplicon sequence variants (see Callahan et al., 2017: https://www.nature.com/articles/ismej2017119)

We are familiar with both approaches. ASVs would indeed give more details on ecotype level. However, the aim of the study was not to dive into detailed taxonomic differences and identities, but to a) identify larger groups (e.g. flagellates, diatoms) and their potential functions and ecological role in relation to the biogeochemical data and b) to show and discuss overall community differences between the samples/sites. For this purpose we believe that swarm clustering of OTUs is appropriate.

L235: Was the data trans-formed in anyway before making the dissimilarity matrix? I'm only asking because it seems doing some type of transformation (e.g. Hellinger) is increasingly common.

We did do Square root transformations and Wisconsin double standardizations and added this for clarity to the text. "… (NMDS) plots are based on Bray-Curtis dissimilarities of square root transformed and double Wisconsin standardized OTU tables…"

Results:

L243: replace "were having" with "had"

We replace "were having" with "had" as suggested.

L244: why is Fig 2 c, d referenced before Fig 2 a, b.. did I miss the reference to a, b somewhere?

Since Fig 2c,d (Salinity profiles) is mentioned before a,b (Temperature profiles) in the text we suggest changing the order of the figure panels (Salinity profiles: a,b, Temperature profiles: c,d)

L265: Are there any photos of the subglacial outflow described in L267-268? Since there is a lack of field data at this time of year I think that these would be of value.

We do have a photo that shows the sampling location of the subglacial discharge water, but the picture is not very clear since the liquid water was sampled below a layer of ice (Icing). We showed the picture in an earlier version of the manuscript, but removed it after the editor pointed out that the picture does not show clearly where we took the sample. We suggest adding the photo again in the supplement with a description where the sample was taken.

[Figure]

Figure S1. Sampling site for the subglacial discharge water. a) Aufeis on land in front of the southern part of the glacier and location of the ice cave shown in b-d (red arrow). b-d) Inside the ice cave with red arrow pointing to the liquid water sampled. The liquid meltwater was mostly covered by a layer of ice. Picture credits: a,c) Josef Elster, b) Marie Sabacka, d) Tobias Vonnahme.

L283: When reading about the very high nitrate+nitrite and silicate concentrations below the ice at SG I found myself really wondering if this could be coming from the subglacial meltwater itself versus upwelling of deeper marine waters. I believe you have data of the glacial meltwater itself? You mention these samples in lines 101-102 .. and I see further on that you present this data in L295. I'd suggest re-organizing so that this comes before the marine data.

We agree that the glacial meltwater data should be shown earlier to answer this question before it arises. We suggest moving the sentences about the subglacial outflow to the start of the paragraph.

L295: missing units for silicate in the outflow water

Thanks for spotting this omission. We add the units of $\mu mol\ L^{-1}$

L300: The definition of conservative mixing is not quite right. The sentences in lines 300- 302 are especially problematic. I see that the other reviewer has already adequately commented on this so I will defer to those comments. In the rest of the paragraph I would avoid the words "positive mixing patterns" and "positive relations". I also found the color scheme in Fig 5 (red and pink) challenging to interpretation.

Concerning the color scheme in Fig 5, we used the same colors as in the rest of the manuscript for consistency. However, we agree that the colors appear too similar in Fig 5 and added a black outline to the red circles which will help improve clarity while keeping it consistent.

Concerning the conservative mixing we changed the text in the following way as described in the response to RW 1:

Nutrient versus salinity relations can provide indications of the endmembers (sources) of the nutrients (Fig. 5) with a linear correlation being indicative of conservative mixing. A positive correlation indicates higher concentrations of the nutrients of the saline Atlantic water endmember, while a negative correlation points to a higher concentration in the fresh glacial meltwater endmember. Biological uptake and remineralisation could weaken or eliminate this correlation, indicating non-conservative mixing. In the water column at NG and IE silicate ($R^2$=0.66, p=0.008), $NO_X$ ($R^2$=0.62, p=0.01) and phosphate ($R^2$=0.69, p=0.005) showed conservative positive mixing patterns with higher nutrient concentrations in the Atlantic water (Fig. 5a-c). SG showed a negative correlation for silicate pointing to a higher concentration in the glacial meltwater ($R^2$=0.86, p<0.0001). The absence of correlations for $NO_X$ and $PO_4$ at station SG indicate non-conservative mixing pointing towards the relevance of biological uptake and release measurements (Fig. 5d-f).

L310: I echo the other reviewer that these calculations of nutrients supplied via upwelling vs the glacial meltwater should be shown.. how were these calculated? What is the error on these calculations? This paragraph needs more explanation for these values to be believed especially considering (as pointed out by the other reviewer) the single outlier values that are driving the gradient in SG samples. Also, at SG, it seems, at least from Fig 5 d-f, that the lower salinity water had higher silicate concentrations but these concentrations were much higher than those reported for the glacial meltwater above. What is the source of this silicate?

Concerning the source of silicate, we prefer to keep this as part of the discussion. (Se ch. 4.4.3 first paragraph). Briefly, the mixing calculations show that the high Si values can be attributed to the subglacial discharge water itself AND bottom water reaching the surface. So, the bottom water appears an important source.

Concerning the calculations and error estimates, we provided following response to RW1 (the error estimates will be added to the text and tabl1 (See above)) that explains our methodology and the inclusion of text as supplement:

We suggest adding the following calculations to the supplement. The mentioned outlier values in SG in the UIW sample were not used for the mixing calculations as explained before. For the meltwater fraction at the surface the error related to the average IE salinity is less than 0.1 % (see comment above), the main variation of the % meltwater contribution in the surface layer of SG is related to the salinity at the surface of SG (Fig. R1). We added the error estimate of 0.1 % to the table. For nutrients, the estimation error was estimated based on the variability in the concentrations measured in the triplicates from each water type. For NOx the estimated range of contribution by upwelling is thereby 57-59 % (± 1 %) bottom water, for Silicate 89-95 % (± 3 %), and for phosphate 46-49 % (± 3 %).

Equations. Mixing calculations for estimates of the fraction of meltwater ($MW_{Sal}$) based on salinity, and for bottom water based on nutrient concentrations ($BW_{Nuts}$). Sal indicates the average salinities measured at the IE ($Sal_{IE}$), SG at 1m depth ($Sal_{SG1m}$), subglacial outflow ($Sal_{glac}$). Nut indicates the nutrient concentrations of nitrate and nitrite ($NO_X$), silicate (Si), and phosphate ($PO_4$) at 1m under the sea ice at SG ($Nut_{1mSG}$) and IE ($Nut_{1mIE}$), the bottom water of the IE ($Nut_{BW}$), or subglacial outflow water ($Nut_{glac}$).

$$MW_{Sal}[\%] = \frac{Sal_{IE} - Sal_{SG1m}}{Sal_{SG1m} - Sal_{glac} + Sal_{IE} - Sal_{SG1m}} * 100$$

$$MW_{Sal}[\%] = \frac{34.7\ PSU - 23.6\ PSU}{23.6\ PSU - 0\ PSU + 34.7\ PSU - 23.6PSU} * 100 = 32\ \%$$

$$BW_{Nut}[\%] = \frac{Nut_{1mSG} - MW_{Sal}[\%] * Nut_{glac} - Nut_{1m_{IE}} + MW_{Sal}[\%] * Nut_{1m_{IE}}}{Nut_{BW} - Nut_{1m_{IE}}} * 100$$

$$BW_{NOX}[\%] = \frac{6.52\mu M - 0.32 * 2.06\ \mu M - 3.27\ \mu M + 0.32 * 3.27\ \mu M}{9.57\ \mu M - 3.27\ \mu M} * 100 = 58\ \%$$

$$BW_{Si}[\%] = \frac{4.30\ \mu M - 0.32 * 1.79\ \mu M - 1.59\ \mu M + 0.32 * 1.59\ \mu M}{4.46\ \mu M - 1.59\ \mu M} * 100 = 92\ \%$$

$$BW_{PO4}[\%] = \frac{0.41\ \mu M - 0.32 * 0.09\ \mu M - 0.34\ \mu M + 0.32 * 0.34\ \mu M}{0.67\ \mu M - 0.34\ \mu M} * 100 = 46\ \%$$

[Figure]

Figure R1. Estimated fractions of glacial meltwater in the surface layer of SG.

L333: Like the other reviewer I'm confused by the term "vertical export of Chl" – what it means, how it was estimated, and what the errors on this estimate are.

See response to RW1 (The error is based on Chl a tripiclates and given in Fig. 6):

The vertical export flux of Chl a is based on Chl a measurements in the sediment traps. We first convert the measured Chl concentrations (mg m$^{-3}$) to mass (mg) in order to calculate the flux as the mass of Chlorophyll a per unit area and time sedimenting to a certain depth.

Suggested change:

This leads to higher (14 times) vertical export flux based on the sediment trap measurements than production at IE and considerably lower (5 %) export than production at SG (Table 2).

L337: "assuming absence of grazing".. this doesn't really seem realistic?

The assumption is necessary since we did not estimate grazing rates. If grazing would be considered the loss rate would be higher. For clarity, we added following sentence.

"As grazing was not estimated in this study, the suggested loss terms of chl a based on the sediment trap data are likely underestimations."

L348: I'd suggest explaining more fully again the goal of the "reciprocal transplant experiment" before giving the results.

See changes in the methods:

"For testing the effect of the water chemistry on phytoplankton growth, we designed a reciprocal transplant experiment where the phytoplankton communities at SG and IE (1 m and 15 m) were transplanted into the water of both SG and IE. 50 ml of the water containing the phytoplankton community were transferred into 50 ml sterile filtered (0.2 μm) seawater of SG or IE in 100 ml polyethylene bottles. The bottles were then incubated in situ at the appropriate depth and primary production measured as described above. The aim of the experiment is to test if water chemistry alone is sufficient to increase primary production, or if the different communities, light regimes, or temperatures are more important."

We also suggest adding a short introduction of the experiment to the results:

"The reciprocal transplant experiment aimed to show the effect of water chemistry on primary production in the absence of effects related to different communities, temperature, or light. The results (Fig. 7) showed clearly …"

Fig 6: The quality of this figure should be improved. The numbers in the parentheses are very difficult to read.

For the final version the quality will be substantially better due to the use of vector files (pdf) instead of png (as in the current pre-print file). We will also increase the font size of the error ranges in the parentheses)

Fig 7: The x-axis with the experiment name are not clear. What does "com" stand for?

We suggest writing "community" instead of "com"

Fig 8: Define UIW in the legend as you have for the other abbreviations

We will write it out as "Under ice water" as suggested.

L355-356: "The first [NMDS1] axis separated sea ice from water communities with no overlapping samples".. this really isn't evident in Fig 8a.. sea ice is the square and what water and under ice water samples are the triangles. These regularly are in the same ellipses, unless I'm missing something? Also, is the glacier outflow sample actually a under ice water sample? What is the salinity of this sample? I guess I'm wondering if this is a true non-marine glacial outflow sample or one that could be diluted by marine water? I think this is an important point that needs to be clarified above.

We agree that the figure needs some clarifications.

1. We agree that sea ice and water samples are not directly separated by axis 1, but by axis 1 and 2 and remove the reference: "Sea ice and water communities are clearly separated with no overlapping samples."
2. The ellipses include subglacial meltwater (Salinity=0), glacier ice (Salinity=0), surface water and sea ice at SG in 2019 and 2018, and the remaining water and sea ice samples (including deeper water samples from SG). For clarity, we suggest coloring the ellipses and add a legend for the eclipse colors. In the figure caption we suggest following clarification: "… Groups highlighted in eclipses: glacier ice (top right), undiluted subglacial outflow (top left), surface samples (UIW, sea ice) at station SG 2019 (top blue), surface samples (1m water, sea ice) at

station SG 2018 (bottom blue) and others including deeper water samples at SG (bottom). The fraction of shared OTUs (in %) are shown as lines scaled to the fraction [%] of shared OTUs.

3. We also suggest using a separate symbol for glacial outflow to avoid confusion about the origin (under the sea ice, or from the subglacial outflow)
4. The aim of the eclipses is to support the discussion of OTU turnover between trhe subglacial outflow and marine samples, which we use for a rough estimate of fluxes and connectivity. Since we only do the analyses for 16S samples (due to short generation time and availability of complete glacier samples), we did not show ellises for the eukaryotic communities.

L358-360: What was the stress on this NMDS? How robust is this ordination you show? I'm always weary of interpreting the axes in this manner, i.e. axes one shows X and axes 2 shows Y .. i.e. similar to how one might view a PCA. I agree that looking at Fig 8a your communities are different but I don't think you can go as far to say that axis 1 is separating ice vs water and axis 2 is separating glacial vs marine. The ordination of this NMDS would likely change each time you ran it.. maybe something to consider?

The stress values are given on top of the NMDS plots (0.07 for 16S, 0.14 for 18S and LM). The stress values are indicative of a very good to good representation in the reduced dimensions. For clarity, we added the information also in the figure caption. We removed the description of which axis separates the community. With the R function used (metaMDS) the ordinations stay the same (The plot is reproducible with the same code).

L371: "Overall the same NMDS clustering has been found as for the 16S rRNA sequencing" .. but in the 18S plot (Fig 8b) no ellipses are drawn.. does this indicate that these group divisions were not significant? The written text doesn't seem to match the figure.

The aim of the eclipses is to support the discussion of 16S OTU turnover between the subglacial outflow and marine samples, which we use to estimate fluxes and connectivity. Since we only do this analyses for 16S samples (due to short generation time and availability of complete glacier samples), we did not show ellipses for the eukaryotic communities. However, for comparability and due to descriptions of clusters in the written text, we suggest adding the ellipses for Fig. 8b, but do not show OTU turnover since the information is not used in the manuscript. We tested for significance using ANOSIM and describe the significant ($p<0.005$) differences in the text.

Fig8c – the separation in the samples is quite striking on this NMDS. How come there are no ellipses on this plot? Were the differences shown in the NMDS not significant? Could try a perMANOVA to test the significance of differences between the groups perhaps?

For Fig 8c we prefer to not add ellipses, since the sampling design differs and the information is not used further in the manuscript. As described in the text, differences between sea water and sea ice are significant (ANOSIM, $p<0.005$), but not the differences between SG surface samples, and other stations. For Fig 8c we also suggest following changes in the text for clarity: "Furthermore sea ice species composition at SG station differed from NG and IE (Fig. 8c)."

Discussion:

L388-391: These first few lines are a great summary and really the abstract and introduction needs to be better set-up to frame these important points: (1) evidence for subglacial upwelling at a shallow tidewater glacier under sea ice and (2) that this upwelling persists in the winter / spring and supplies nutrient-rich glacial meltwater and upwelling of bottom water: : : I actually think part of the confusion is the use of the term "upwelling" to describe the release of submarine discharge into the ocean and also the upwelling of bottom water. Perhaps a change of language throughout would be helpful – i.e. saying "submarine discharge" vs "subglacial upwelling". And as per points above the case about nutrient-rich glacial meltwater needs to be set-up and made earlier as it's really a central finding.

The referee has a good point that subglacial upwelling and submarine discharge are two different processes. We suggest changing the terminology of submarine upwelling to submarine discharge where necessary ( e.g: "(1) evidence for submarine discharge at a shallow tidewater glacier under sea ice and (2) that this submarine discharge persists in the winter") throughout the manuscript. As mentioned above, we also moved the results description of nutrients in subglacial meltwater to the beginning of the nutrient section and added an introduction about the effect of water storage underneath a glacier over winter on the water chemistry (silicate enrichment by prolonged contact with the bedrock -> weathering, ion concentration by solute expulsion during freezing of stored meltwater)

L406: The phrasing "which does not allow basal glacial ice to melt" is unclear. The whole sentence is too long and should be made into 2, but are the authors saying that because there is not Atlantic inflow water there can be no basal ice melt? Basal ice melt can result from geothermal heat flux, overburden ice pressure, and sliding friction. Warm ocean water is not the only mechanism. I suggest looking at a textbook (e.g. the physics of glaciers) and reviews on this topic: e.g. Hubbard and Sharp, 1989

We realize that we used the wrong terminology here. We are discussing glacier terminus (glacier-marine interface) ice melt, and not basal (glacier-bedrock interface) ice melt. We corrected the terminology throughout the discussion. We also agree that the sentence can be splitted in 2.

L407: "Subglacial meltwater itself is unlikely to lead to basal ice melting due to its low salinity". This sentence is very unclear to me. I'm not sure what this sentence is saying or trying to say.

We agree that this sentence is very unclear and suggest removing it.

L407-408: "However, basal ice melt is likely more important in systems with Atlantic water inflows: : :" as per above this seems to ignore the possibility of basal ice melt underneath temperate and polythermal glaciers. This may not be what the authors mean but as written it reads this way.

As mentioned above, we meant glacier terminus ice melt and not basal ice melt and correct the terminology.

L420: "remains from the previous melting season" is unclear. Can you specify what you mean by remains.

We refer to fresh meltwater that entered the fjord during the previous melting season (summer), remaining at the surface (due to its lower density) throughout winter due to limited mixing and advection. We suggest following change: "may be meltwater introduced during the last summer to fall melting season and remaining throughout winter."

L433: Can you specify what data you are referring to when you say "estimated bacterial growth rates". I searched for this term in the paper and did not see it previously defined. It really should be so that the basis for this calculation of doubling time is clear.

The estimated bacterial growth rate is given in table 2 as bacteria biomass production. We suggest replacing the term "growth rate" with "biomass production" for consistency and to add a reference to table 2 in the text.

L442: Why does the supply have to be "constant" ? It seems like (from the methods) that samples for community analyses were only taken once at each station? How does a single-time point sample give an indication of the timescale of submarine discharge into the fjord? This might be a bit of a reach based on the community data alone – suggest tempering this statement.

We agree that "constant" appears to be the wrong term. We suggest using the term "continuous" instead. The argument is that we assume that the Bacteria that are only present in subglacial outflow and surface SG water are inactive and not growing. Considering the doubling time of the entire bacteria community, these inactive not-growing bacteria would be replaced by active bacteria in the time frame of the

doubling time. In addition to overgrowth, inactive bacteria would also be exposed to losses due to grazing, viral lysis, and sedimentation. We acknowledge that these assumptions are very simplified and suggest to also add some terms to show the uncertainty of this estimate: "Thus, we suggest that the presence of shared OTUs between SG and the glacial outflow may indicate a continuous supply of fresh inoculum to sustain these taxa."

L442-444: When you say the "southern part of the glacier" is this part on land or in the ocean? If it's on land you should specify. I also think that this assumption that this outflow is being released under the marine-terminating portion can be backed up by your marine data? This sentence seems out of place here.

Yes, we refer to the land-terminating part. We suggest adding the detail in the following way "land-terminating part south of the glacier". We also agree that we have marine data to support this hypothesis (e.g. Salinity profiles). The observed subglacial outflow on land is simply an additional piece of evidence. We suggest replacing "the clearest evidence" with "clear evidence" and adding a sentence about the marine evidence. "In addition, our marine evidence by salinity and nutrient profiles, turbidity, and communities support the occurrence of submarine discharge in early spring."

L445- to end of paragraph: This explanation of glacier hydrology really needs to come earlier. As written this whole section on the potential magnitude of upwelling is poorly organized. Suggest first setting it up by talking about processes on the ice and then what's happening in the ocean.

We addressed this comment by 1) introducing the glacier hydrology more extensively in the introduction and 2) moving the section about glacier hydrology (442-451) to the end of chapter 4.1 since it is part of the evidence for submarine discharge and not directly for the magnitude/ flux.

L456: "Our mixing calculations estimate".. where are these calculations described?

See comment above. We added the equations and calculations to the supplement.

L457: At what depth is the submarine discharge exiting the glacier? I find myself wondering at what depth these different water masses occur (can you specify this) and how deep the DLAW is being entrained from? Is it sufficiently below the nutricline to be replete in nutrients? Also the calculated entrainment factor of 1.6, how was this calculated exactly? And you state "which pulled 1.6 times more DLAW" .. more than what? This is not clear.

Considering the estimated depth at the glacier terminus of 20 m, this would be the depth of the discharge exiting the glacier. Nutrients are depleted at the surface, but not at 15m, indicating that the discharge happens below the nutricline and has therefore the potential for upwelling.

We suggest adding this information in the following way: "Nutrients were depleted in the UIW, but not at 15 m depth, showing that the nutricline had to be shallower than 15 m. Hence, submarine discharge at a glacier terminus depth of 20 m would cause upwelling ofc nutrient rich DLAW to the surface."

The entrainment factor is the proportion of contributions from DLAW to SGO at the surface (53% DLAW: 32% SGO = 1.6 DLAW:SGO at 1m depth). We suggest replacing "more" with "as much" for clarification. We also specified the calculation by replacing the "(53%)" by "(53 % DLAW : 32 % SGO = ratio of 1.6)" in the manuscript.

L458-459: "Fransson et al. (2020) found that 30-60% of glacier derived meltwater was incorporated in the bottom sea ice : : : again indicating that it is a widespread process at marine terminating glacier fronts" .. what is a widespread process? The release of submarine discharge and its incorporation into bottom sea ice OR the entrainment of different water masses (i.e. DLAW) as the plume rises (as discussed in the previous sentence). Again, this is a case in point of the organizational structure and lack

of specificity of terms "submarine discharge" vs "upwelling of bottom waters" to be a source of confusion.

We suggest following clarification "… indicating that winter/ spring submarine discharge and the resulting formation of sea ice with low porosity is a widespread process…".

L461: "Compared to the massive subglacial plumes of summer systems" .. where? This should be specified .. different glaciers have widely different discharge fluxes. The citation seems to be from Greenland but these glaciers will bear little resemblance to Svalbard, perhaps citing summer discharge fluxes from Svalbard glaciers too would be useful – particularly from your study site if the intent of this sentence is to contrast with spring discharge fluxes as seems to be the case.

We agree that the structure of the entire chapter needed improvement. Thus, we rewrote the entire chapter, considering all comments. Concerning this specific comment, we specified the location and time of each tidewater glacier system compared. We start with stating the conditions in our study, continue with the most similar glacier on Svalbard, and finish with a wider picture by comparing the data to the larger and deeper Greenland glaciers.

The sentence mentioned by the RW was rewritten in the following way: "Our study suggests that subglacial upwelling in spring results in a small volume transport with only about $>1.1$ $m^3$ $m^{-2}$ $month^{-1}$ (approx. 2 $m^3$ $s^{-1}$). This estimate is based on the flux of nutrient rich bottom water needed to maintain the measured primary production assuming steady state conditions and is therefore a rough, but conservative estimate. The most comparable estimate on the magnitude of the upwelling is available at Kronebreen for summer. This Svalbard tidewater glacier is of similar size and had one order of magnitude higher upwelling rates compared to our study (31-127 $m^3$ $s^{-1}$, Halbach et al., 2019). Due to their size, summer subglacial upwelling in Greenland is two to four times higher than at Kronebreen (250-500 $m^3$ $s^{-1}$, Carroll et al., 2016)."

L462: "subglacial upwelling in spring is a small volume transport".. where is this data from? This study? This should be explicitly stated. Suggest re-writing this entire sentence. Also, the last part of the sentence regarding upwelling needed to maintain primary production should be a new sentence as this is a different point then the discharge flux.

The data are from this study. We agree that this should be stated. We also agree that the information "needed to maintain primary production should be moved to a seperate sentence. We rewrote the entire chapter, considering all comments. As suggested by RW1 we also converted the discharge units of the three studies (Greenland, Kongsfjorden, our study) to the same units for comparability. Concerning this comment, following changes were made:

"Our study estimated only a small volume transport through subglacial upwelling in spring of only about 1.1 $m^3$ $m^{-2}$ $month^{-1}$ (approx. 2 $m^3$ $s^{-1}$). This estimate is based on the flux of nutrient rich bottom water needed to maintain the measured primary production assuming steady state conditions and is therefore a rough, but conservative estimate."

L464: "This careful estimate".. I'd remove the word "careful".. the more so because the sentence before this one is unclear! Is this estimate of freshwater input for Billefjorden in the summer or spring? It's unclear. The estimate from the Halbach paper is I believe from the summer so you want to make sure you are comparing like with like.

As pointed out by RW1, "careful estimate" is a misleading formulation. We suggested to replace it with "rough, but conservative". We also realized that the reason for comparing our spring study with summer values is not clear and suggest specifying that we do not know of any other spring studies with similar estimates. The study in Kongsfjorden is the most comparable estimate to our study (glacier size, terminus depth, location). We suggest following changes: "To our knowledge, our study provides currently the only available estimate of subglacial upwelling in early spring. …. The most comparable

estimate on the magnitude of the upwelling is available at Kronebreen for summer. This Svalbard tidewater glacier is of similar size and had one order of magnitude higher upwelling rates compared to our study (31-127 m$^3$ s$^{-1}$, Halbach et al., 2019)."

L465-466: The fact that you have less entrainment than the Hopwood study is really not surprising at all considering the depth of discharge and flux of discharge at the much deeper, larger glaciers in that study. I'm not sure what the purpose is of this statement? As written now it's failing to provide relevance to this study.

We agree that this fact is not surprising and rephrased the statement. We still argue that it is necessary to compare entrainment rates and state that the glacier terminus depth is typically the controlling factor, apparently independent of the time of the year.

"In our study about 1.6 times as much bottom water (DLAW) as subglacial outflow water (SOW) reached the surface at SG (Entrainment factor of 1.6 – see above) through the upwelling process. The entrainment factor is mostly dependent on the depth of the glacier front (Carroll et al., 2016). The glacier terminus at SG was shallower (approx. 20 m) than any other studied tidewater glacier on Svalbard (70 m depth at Kronebreen, Halbach et al., 2019) or Greenland (> 100m, Hopwood et al., 2020). Hence, the higher summer entrainment factors estimated in Kongsfjorden (3, Halbach et al., 2019) and Greenland (6 to 10, Hopwood et al., 2020) are not surprising. Overall, glacier terminus depth appears to be the main control of entrainment rates, likely independent of the time of the year. However, turbulent mixing may cause increased entrainment during times of very high subglacial discharge rates."

L466-467: "each volume of SGO water pulled about the same volume of DLAW with it to surface".. this is unclear.. do you mean each volume over a certain timeframe (a day? A week? A month?) .. what is the volume exactly? What was the volume of DLAW entrained? This should be stated if you are speaking about volumes here. And again the comparisons to the Hopwood study don't' seem relevant if you are comparing to large Greenland glaciers. You should specify where and what type of glaciers in the Hopwood review you are comparing too.

We refer to proportion of volumes (Vol DLAW : Vol SOW), which is a value comparable to chemical volume percentages (e.g. 70% Ethanol in MQ vol/vol). Thereby an exact volume is meaningless. To avoid confusion, we rephrased the sentence in the following way.

"In our study about 1.6 times as much bottom water (DLAW) as subglacial outflow water (SOW) reached the surface at SG (Entrainment factor of 1.6 – see above)"

We also specified the type (depth, size, location) and time (summer) of the compared studies as mentioned above.

To our knowledge, our study provides currently the only available estimate of subglacial upwelling in early spring. ….The entrainment factor is mostly dependent on the depth of the glacier front (Carroll et al., 2016). The glacier terminus at SG was shallower (approx. 20 m) than any other studied tidewater glacier on Svalbard (70 m depth at Kronebreen, Halbach et al., 2019) or Greenland (> 100m, Hopwood et al., 2020). Hence, the higher summer entrainment factors estimated in Kongsfjorden (3, Halbach et al., 2019) and Greenland (6 to 10, Hopwood et al., 2020) are not surprising. Glacier terminus depth appears to be the main control of entrainment rates, likely independent of the time of the year."

L470: This is the first mention of the depth of the discharge. As you say, 20-m is quite shallow. Are nutrient concentrations sufficiently high enough here to augment surface concentrations? In other words, is this depth below the nutricline.

As mentioned above, we now mention the depth earlier in the chapter. We also provide information on the depth of discharge in relation to nutricline (see comments above).

"The entrainment factor is mostly dependent on the depth of the glacier front (Carroll et al., 2016). The glacier terminus at SG was shallower (approx. 20 m) than any other studied tidewater glacier on Svalbard (70 m depth at Kronebreen, Halbach et al., 2019) or Greenland (> 100m, Hopwood et al., 2020)."

We also mentioned that the submarine discharge enters the fjord below the nutricline in the end of the chapter.

"In spite of the shallow depth, and the low discharge and entrainment rate of our study, subglacial upwelling appears to be the main mechanism to replenish bottom water with high nutrient concentrations to the surface and can substantially increase spring primary production due to; (i) submarine outflow below (approx. 20 m) the nutricline (<15 m), (ii) the absence of any other terrestrials inputs, (iii) Atlantic water blocked by a shallow sill (Skogseth et al., 2020), (iv) very weak tidal currents (Kowalik et al., 2015), and (iv) wind mixing blocked by sea ice in Billefjorden, and (v) undiluted subglacial meltwater having lower nutrient concentrations than the DLAW."

L473-to end of paragraph: This seems to directly contradict previous statements regarding the glacial meltwater discharge being enriched in nutrients (e.g. silicate?). Also many of the comparisons you are making are to summer discharge fluxes and summer entrainments.. the spring discharge will of course be lower but more chemically enriched from the glacial meltwater discharge? I think if you are going to use the summer values to compare, which you might have to do out of necessity and lack of other comparisons, you need to state so explicitly, and the limitations of such comparisons.

The glacial meltwater is enriched in silicate, considering its salinity (0) and compared to UIW and sea ice at NG and IE, but not compared to the bottom water. We tried to clarify it by following statement:

"…(v) undiluted subglacial meltwater having lower nutrient concentrations than the DLAW"

As mentioned above, we fully agree with the confusions about the comparisons. We rewrote the entire chapter in the following way:

"To our knowledge, our study provides currently the only available estimate of subglacial upwelling in early spring. Our study suggests that subglacial upwelling in spring results in a small volume transport with only about >1.1 m$^3$ m$^{-2}$ month$^{-1}$ (approx. 2 m$^3$ s$^{-1}$). This estimate is based on the flux of nutrient rich bottom water needed to maintain the measured primary production assuming steady state conditions and is therefore a rough, but conservative estimate. The most comparable estimate on the magnitude of the upwelling is available at Kronebreen for summer. This Svalbard tidewater glacier is of similar size and had one order of magnitude higher upwelling rates compared to our study (31-127 m$^3$ s$^{-1}$, Halbach et al., 2019). Due to their size, summer subglacial upwelling in Greenland is two to four times higher than at Kronebreen (250-500 m$^3$ s$^{-1}$, Carroll et al., 2016). Due to their size, summer subglacial upwelling in Greenland is two to four times higher than at Kronebreen (250-500 m$^3$ s$^{-1}$, Carroll et al., 2016). In our study about 1.6 times as much bottom water (DLAW) as subglacial outflow water (SOW) reached the surface at SG (Entrainment factor of 1.6 – see above). The entrainment factor is mostly dependent on the depth of the glacier front (Carroll et al., 2016). The glacier terminus at SG was shallower (approx. 20 m) than any other studied tidewater glacier on Svalbard (70 m depth at Kronebreen, Halbach et al., 2019) or Greenland (> 100m, Hopwood et al., 2020). Hence, the higher summer entrainment factors estimated in Kongsfjorden (3, Halbach et al., 2019) and Greenland (6 to 10, Hopwood et al., 2020) are not surprising. Glacier terminus depth appears to be the main control of entrainment rates, likely independent of the time of the year. Kronebreen is the most comparable tidewater glacier in terms of glacier terminus depth and entrainment rate. Although the estimated entrainment rate was low at Kronebreen (3), it substantially increased summer primary production in Kongsfjorden (Halbach et al., 2019). In spite of the shallow depth, and the low discharge and entrainment rate of our study, subglacial upwelling appears to be the main mechanism to replenish bottom water with high nutrient concentrations to the surface and can substantially increase spring primary production due to; (i) submarine outflow

below (approx. 20 m) the nutricline (<15 m), (ii) the absence of any other terrestrials inputs, (iii) Atlantic water blocked by a shallow sill (Skogseth et al., 2020), (iv) very weak tidal currents (Kowalik et al., 2015), and (iv) wind mixing blocked by sea ice in Billefjorden, (v) undiluted subglacial meltwater having lower nutrient concentrations than the DLAW."

L480: The word "Surprisingly" seems to not be the right word choice here.

We suggest removing the word "Surprisingly".

L438: "Substantial subglacial upwelling" .. I'm unclear was to what you are referring to here – is this submarine discharge of glacial meltwater or upwelling of bottom waters? In either case the word "substantial" seems ill-advised here given the preceding discussion and should be removed. Could it be that you didn't observe much light limitation because the plumes were not that "massive" (compared to summer).. i.e. you just have a much smaller discharge flux and therefore plume in the spring? This seems likely and unsurprising.

We agree that the formulation is misleading and removed it.

L485-86: Unclear what the phrase "where light is not considered limiting" is referring too.

We suggest specifying in the following way: "where light sufficient for photosynthesis".

Line 511: "rations" should be "ratios"?

We replaced the term "rations" with "ratios".

L515: Can you really call it "deep water upwelling" if the water is being entrained from only 20-m? This is problematic (at least for me) and needs to be clearly addressed I think.

We suggest replacing the term "deep water" with "bottom water".

L517-519: The discussion on iron seems unrelated and as written is unconvincing.

We consider a short discussion of iron important for a comprehensive discussion. Without the information the reader may consider iron as important micronutrient not considered and potentially important, which would weaken the robustness of the study. By acknowledging that iron may be imported in large amounts, but is not limiting in coastal Arctic systems, we clarify this potential question briefly. We suggest adding following clarification and an additional reference: "However, iron limitation is untypical in coastal Arctic systems (Krisch et al., 2020)."

L520: "nutrient concentrations may simply be higher due to the shallower depth at SG" .. why? It's unclear what you are trying to say. Suggest re-writing with more detail and explicity.

Nutrients are typically higher close to the sea floor due to benthic regeneration of organic matter in the sediments. If the surface water is only 30m over the bottom, vertical mixing via diffusion or advection needs consequently less time and/or physical forcing than at 150 m depth. We suggest following clarification: "nutrient concentrations may be higher due to less physical forcing and time needed for vertical mixing at the shallower SG compared to IE.

L529: Was the Frasson study done at this same site?

No the study was done at the neighboring fjord. We added the information in the following way: "The role of bedrock derived minerals and particles for composition of sea ice chemistry have been described in detail in the neighboring fjord (Tempelfjorden) by Fransson et al. (2020)."

L530: "The values" .. vague.. specify what kind of values you are referring to.

We replaced "The values", with "Silicate concentrations"

L535: Paragraph ending here is rambling and needs to be re-written. Suggest taking out the iron since you have no data on this to compare.

We agree and removed the last sentence about iron.

We suggest following clarification: "…which was introduced via subglacial upwelling in Kongsfjorden…"

L538: Were are you proposing this nitrification is occurring? In the ocean or in the glacial meltwater? Could the high nitrate come from the subglacial waters itself? See papers by Beaton et al. in Greenland, Jemma Wadham, Boyd et al., 2011 (AEM) and Wynn et al., 2007 (Chemical Geology). Do you have measurements of the outflow un-diluted by seawater so you can rule this possibility out?

We propose the nitrification to happen in the UIW. We suggest adding the following information: "Ammonium regeneration and subsequent nitrification under the sea ice…". We disregard high nitrate inputs from the glacial meltwater itself since we did not measure high nitrate concentration in our samples from the outflow of undiluted meltwater (see Table 1). For clarification we suggest adding the following statement: "Nitrate can be supplied through the subglacial meltwater itself (Wynn et al., 2008), however we did not find high nitrate concentrations in the undiluted subglacial outflow water in our study."

L566: Were you able to resolve any low-light level species in your molecular community composition data to back this statement up?

In general, diatoms are know to be quite well adapted to low light levels. Diatoms were also th most common taxon of the UIW phytoplankton community (based on light micsroscopy, which is more quantitative). We suggest adding a statement of the capability of diatoms to grow under low light conditions. "In particular diatoms, the most common taxa of under ice phytoplankton blooms (von Quillfeldt, 2000, this study) are known to be well adapted to low light conditions (Furnas, 1990)."

Furnas MJ (1990) In situ growth rates of marine phytoplankton: approaches to measurement, community and species growth rate. J Plankton Res 12:1117–1151

We suggest replacing "their production" with "primary production"

L646: "In winter and spring, this would result in the lack of subglacial upwelling".. but with more melt there would be longer melt seasons and presumably more submarine discharge and associated upwelling – at least in the shorter term?

We suggest adding following information: "In the shorter term, a longer melt season and presumably increased submarine discharge may lead to increased subglacial upwelling in winter and spring. However, on longer time scales , tidewater glaciers will retreat and transform towards land terminating glaciers (Błaszczyk et al., 2009), which would result in the lack of subglacial upwelling and systems more similar to the IE with less nutrients and light available for phytoplankton."

---

## Author Response (AR1)

Vonnahme et al., conduct spring-time measurements in a tidewater glacier fjord and provide evidence that entrainment of subsurface-fjord waters by early season freshwater discharge is a measurable nutrient source to under-ice phytoplankton blooms which would otherwise be nutrient-limited at this time of year. The hypothesis and general idea is quite novel, supported by some in situ data as well as incubation studies, and I think it is an interesting addition to the field. I have no specific expertise in seaice or in the analysis of algae community composition- I defer to a more appropriate reviewer on these aspects. I do have quite a few comments throughout the text, but nevertheless found the discussion paper an interesting read suitable for the journal. Most of my comments are minor, or simply requests for a little more clarity.

We want to thank the reviewer sincerely for the thorough and very helpful review. We addressed all comments as described below and believe that the changes improved the manuscript considerably.
We used following font colors and highlighting for the changes:
- Grey text: Reviewers comments
- Black text: Our response
- Red text: Text from the manuscript with the changes
- Yellow highlights: Parts of the manuscript that we changed in some Arctic fjord systems, many don't have this and some of the few available case studies don't have high primary production.

We agree that the current formulation is inadequate, since not all fjords have tidewater glaciers, and the primary production in these tidewater-influenced fjord is typically high compared to other summer systems, but not high compared to spring blooms.

We did the following change:

Subglacial upwelling of nutrient rich bottom water can sustain elevated summer primary production in tidewater glacier influenced fjord systems.

You do briefly comment on this once, but considering the timescale required to get significant shifts in the groundling line of these glaciers, could you also think about trends in sea-ice loss as well. Could the timing of sea-ice loss from some of these fjords shift, or could sea-ice even completely disappear, before significant changes in grounding line are evident?

We agree that sea-ice loss is certainly important and should be discussed together with the glacier retreat.

We did following change in the abstract:

We suggest that climate change caused retreat of tidewater glaciers could lead to decreased under-ice phytoplankton primary production, while sea ice algae production and biomass may become increasingly important, unless sea ice disappears before, in which case spring phytoplankton primary production may increase.

We also added more details about sea-ice loss and changed in timing of the spring bloom due to earlier sea-ice break up, increased DOM and sediment inputs (brownification) to the discussion.

35 "During summer" Here I think the reference you mean is a similar paper by the same author that
deduces the 'upwelling' effect in summer has a measurable impact on silicic acid availability1, the paper
cited is from the same area but concerns the spring bloom in the same fjord (which has some sea-ice
cover in spring)2. I think you could be a little more precise here.

We thank the reviewer for the comment and changed the reference accordingly.

39 Is time not also important here? If you didn't have any sediment close to the glacier front, the nature
of the upwelling followed by outflow at the surface would surely lead to relatively low primary production
at the glacier front anyway, because the freshly upwelled water is being laterally advected away from the
front i.e. I assume you would never see the highest cell counts here even in the absence of high turbdity?
Looking specifically at the system studied by Meire et al., the bloom also seems to peak a fair way
downstream of the large glaciers, even though there isn't much turbidity in the inner fjord.

We agree that time and advection are also important here. While phytoplankton biomass increases
towards a bloom, it is already advected away from the glacier front.

We did following change (including suggestions by RW 2):

Primary production is typically low in direct proximity to the glacier front (hundreds of meters to
kilometres, Halbach et al., 2019) due to high sediment loads of the plumes absorbing light and thereby
inhibit primary production close to the glacier front., but potentially also due to lateral advection (Meire
et al., 2016ab; Halbach et al., 2019).

46 I don't think there's a perception of no freshwater release from these systems overwinter, there are
several papers demonstrating this (for a good recent Arctic example3), although I agree there is a big
problem with bias in the distribution of data towards the peak meltwater season, and model discharge
curves do I think include a little early/late season discharge comparable to that observed here4. I think a
more accurate statement would be that whilst it's known that a little discharge occurs early/late season
there simply isn't much data to quantify it.

We agree that the formulation need clarification. There are indeed studies on the subglacial upwelling in
winter and spring, but they are rare, compared to summer studies.

We add the suggested references and did following change (including suggestions by RW 2):

Due to the challenges of Arctic field work in early spring and the difficulties of locating such an outflow,
only few studies investigated submarine discharge during that time window. The few studies available
suggest overall little discharge (e.g. Fransson et al., 2020; Schaffer et al., 2020) compared to summer
values. The limited amount of data makes the generalized quantification of subglacial outflow difficult. In
addition, studies focusing on the potential impacts of the early spring discharge on sea ice and pelagic
primary production are lacking.

48 I think you should distinguish 3 sources here, as written icebergs and terminus melt, but also subglacial
discharge at the grounding line

We agree and did following change:

In addition to subglacial discharge at the grounding line, tidewater glacier related upwelling mechanisms
can also be caused by the melting of deep icebergs (Moon et al., 2018), or the melting of the glacier
terminus in contact with warm seawater (Moon et al., 2018, Sutherland et al., 2019).
51 in summer. (Reference?)
We added the references by Moon et al. (2018) and Sutherland et al. (2014)

Moon, T., Sutherland, D. A., Carroll, D., Felikson, D., Kehrl, L., and Straneo, F.: Subsurface iceberg melt
key to Greenland fjord freshwater budget, Nat Geosci, 11(1), 49-54, https://org/10.1038/s41561-017-
0018-z, 2018.

Sutherland, D. A., Straneo, F., & Pickart, R. S.: Characteristics and dynamics of two major Greenland
glacial fjords, Journal of Geophysical Research: Oceans, 119(6), 3767-3791, 2014.

61 "while at the same time entrapping considerably less light absorbing sediments" I'd be genuinely
interested to know if you can provide data/refs to support this. I'm not sure we know what turbidity looks
like in these sub-surface plumes and to what extent it represents particles from the ice melt/basal
erosion, or resuspension, and how this changes through the year, although I would agree it would be
expected to be low in this environment given the description of the fjord

We agree that some support in form of references and data is helpful here. We added a short reference
to a study at Hansbreen, another polythermal Svalbard tidewater glacier. The study monitored SPM
throughout the year and found the lowest SPM value in winter and spring at a depth of about 5-10 m,
which fits to our study. The origin is resuspension, as well as subglacial discharge.

We added following detail to the introduction:

Sediment inputs into the fjord during this time of the year are low with peaks deeper in the water column,
indicating limited impacts on surface primary production (Moskalik et al., 2018).

Moskalik, M., Ćwiąkała, J., Szczuciński, W., Dominiczak, A., Głowacki, O., Wojtysiak, K., and Zagórski, P:
Spatiotemporal changes in the concentration and composition of suspended particulate matter in front
of Hansbreen, a tidewater glacier in Svalbard, Oceanologia, 60(4), 446-463 2018.

63 "this spring upwelling mechanism could be the primary mechanism to significantly increase primary
production" does not read well, try "could be a mechanism via which primary production is increased in
tidewater fjords compared to similar fjords without these glaciers: : :" or similar.

We changed the text accordingly (including suggestions by RW 2):

We suggest that in the absence of wind induced mixing, due to the seasonal presence of fast ice cover in
spring, submarine discharge of glacial meltwater can directly (ion enrichment over the subglacial storage
period) or indirectly (upwelling) be a significant source of inorganic nutrient increasing primary
production in front of tidewater glaciers compared to similar fjords without these glaciers. Especially after
nutrients supplied via winter mixing are incorporated into algal biomass (Leu et al. 2015) this additional
nutrient source may become important.

65 I think you need a reference here. Retreat may generally co-occur with shoaling of the grounding line,
but not always, and there may also be an increase in discharge which could offset shoaling to some extent
as entrainment also depends on freshwater discharge volume. Also, you comment on upwelling being
eliminated, but wind-driven upwelling will remain, so maybe be a bit more precise.

We clarified the statement in the following way and added a reference:

Higher glacial melt rates and earlier runoffs may initially increase tidewater glacier induced upwelling,
due to increased subglacial runoff (Amundson and Carroll, 2018). However, their retreat and
transformation into shallower tidewater glacier termini may lead to less pronounced upwelling, unless
the shallower grounding line is compensated by the increased runoff (Amundson and Carroll, 2018).
Eventually, the tidewater glaciers transform into land terminating glaciers, where wind induced mixing
is still possible, but subglacial upwelling is eliminated (Amundson and Carroll, 2018) – potentially
reducing primary production.

107 'shallow' Do you know the approximate depth?

We estimated the approximate depth based on bathymetric maps and added the information to the
manuscript:

The fjord is separated from Isfjorden, a larger fjord connected to the West Spitsbergen current, by a
shallow approximately 30 to 40 m deep sill (Norwegian Polar Institute, 2020),…

120 "were melted in 50 % vol/vol sterile filtered" Is there a reason for this?

Sea ice is typically melted in in sterile filtered seawater to avoid osmotic shock and lysis of organisms in
the ice. Microorganisms in the ice live mostly in brine channels with high salinity, while the frozen ice
around is very fresh. Melting the ice around would lead to an overall very low salinity, leading to severe
stress to the high-salinity adapted organism.

We added following information:

…were melted in 50 % vol/vol sterile filtered (0.2 µm Sterivex filter, Sigma-Aldrich, St. Louis, MO, USA)
seawater to avoid osmotic shock of cells (Garrison and Buck 1986), …

Garrison, D. L., and Buck, K. R.: Organism losses during ice melting: a serious bias in sea ice community
studies, Polar Biol 6:237-239, 1986.

131 One metre or 1 m

Since it is the beginning if a sentence we changed it to "One metre".

140 Does the exact salinity you use for your inflowing seawater have a large impact on your calculations,
can you state what the value of 34.6 refers to? Also, can you clarify to what extent small changes in this
would matter (you may want to calculate the saline endmember uncertainty and propagate it?

For the mixing calculations, we used initially the salinity of meltwater (0 PSU) and the bottom water at
the glacier front. However, we realize that the average salinity of the well-mixed water column at the ice
edge reference site with a salinity of 34.7 is better suited for the calculations. We changed the salinity
and added the information where the value comes from and what the standard deviation is. Since the
value of 34.7 as the bottom water endmember is very stable, variations would lead to <1% changes in the
estimate of the calculations.

We did following changes:

The fraction of fjord water vs subglacial meltwater for the water samples was calculated assuming
linear mixing (Equations 1-2) of the two water sources with different salinities (glacial meltwater salinity
= 0 PSU, average seawater salinity at IE=34.7 ± 0.03 standard deviation), since no other water masses in
regard to temperature or salinity signature were present (Table 1). The variability of the IE sea water
salinity leads to a small ( 0.1 %) uncertainty in the estimated value of the relative contributions of sea
water vs subglacial meltwater.

197 What does 'net haul' refer to? -> Mentioned one paragraph above

We gave details about the phytoplankton net hauls in line 181, but changed the term net haul to
"phytoplankton net" for clarity.

Change:

For qualitative counting of algal communities, the phytoplankton net and bottom sea-ice samples

285 "where NOX (10 _mol L-1) and silicate (19 _mol L-1) levels were exceptionally high (Fig. 4)." Am I right
that this is basically driven by one data point? If so, how do you explain such high NOx concentrations? Is
it possible that this value is an anomaly? And what are the implications of this? From your profile, and
from the range of the other samples I guess that this is an outlier for both NOx and Si, with the NOx harder
to explain from environmental processes. If this is an outlier, I think the calculations throughout need
amending or flagging to reflect this, noting that there is a large difference depending on whether or not
this data point is included.

We agree that these exceptionally high values have to be treated carefully. We took great care during the
nutrient analysis itself and the calibration of the auto-analyzer, and we have no indications on instrument
caused errors in our data record. Local remineralization and dissolution of algae biomass at the sea ice-
water interface may provide part of the explanation, but other anomalies cannot be excluded since the
values are indeed driven by only one sample with no correspondance or obvious source in the values
below or above. Thus, we did not use this value for any mixing calculations or estimates, but used instead
the value 1 m under the sea ice for all further calculations. The 1 m value is more consistent with the
measurements in the water column below and sea ice above. Thus the exceptionally high values had been
considered as outliers and not used in our estimates.

We did following change:

where $NO_X$ (10 µmol $L^{-1}$) and silicate (19 µmol $L^{-1}$) levels were exceptionally high (Fig. 4). As these values
are driven by a single sample, we cannot exclude anomalies to be responsible for these high values.
Wetherefor used the values measured 1 m under the sea ice for further calculations in this manuscript
as surface water reference.

291 "N:P ratios were generally highest: : :" Somewhere it would be interesting to comment on what drives
this trend? Is it a source of N, or a sink/dilution of P? If saline water inflow dominated the N and P supply,
would you expect such strong shifts? I suspect you need some sort of local process leading to a net
accumulation of N or loss of P to get these ratios (you do comment on this for NH4 briefly), and whilst
there are no other spring studies I can think of looking at this, I think a few papers have commented on
some not particularly well explained P loss in similar environments in summer 5,6.

We added a more thorough discussion of the N:P ratios, including the suggested reference in the
following way:

Ammonium regeneration and subsequent nitrification (Christman et al., 2011) under the sea ice may
explain the exceptionally high nitrate concentration of the UIW at SG, which can partially explain the
high N:P ratios. In fact, bacterial activity was higher at SG potentially allowing higher ammonium
recycling. Another explanation for the high N:P ratios and low phosphate concentrations could be
phosphate scavenging by iron as discussed by Cantoni et al. (2020).

300 (306) "Nutrient versus salinity profiles give indications of the endmembers (sources) of the nutrients
(Fig. 5). A positive correlation for example would indicate conservative mixing (assuming high salinity
Atlantic water endmember had higher concentrations than melt water). Biological uptake and
remineralisation as well as physical processes, such as external inputs by meltwater could inverse or
eliminate the correlation."
This isn't quite right and needs a bit more clarity, you will find a lot of literature on this in marine chemistry
or in a good textbook. In simple terms, a linear correlation shows conservative mixing, the absence of a
non-linear correlation suggests non-conservative processes (although there are some subtleties to this,
some physical factors can also lead to non-linearity). The gradient, not the strength of the correlation,
indicates whether fresh, or saline, endmembers have a higher concentration, i.e. an increasing nutrient
concentration with salinity (positive gradient) suggests saline inflow has higher nutrient concentrations,
whereas a decrease with salinity suggests (negative gradient) a higher freshwater concentration.

We corrected the paragraph in the following way:

Nutrient versus salinity profiles can give indications of the endmembers (sources) of the nutrients (Fig. 5)
based on a linear correlation indicating conservative mixing. A positive correlation for example would
indicate conservative mixing (assuming high salinityindicates higher concentrations of the nutrients in
the saline Atlantic water endmember had, while a negative correlation points to a higher concentrations
than melt water) in the fresh glacial meltwater endmember. Biological uptake and remineralisation as
well as physical processes, such as external inputs by meltwater could inverse or weaken or eliminate the
correlation, indicating non-conservative mixing. In the water column at NG and IE silicate (R2=0.66,
p=0.008), NOX (R2=0.62, p=0.01) and phosphate (R2=0.69, p=0.005) showed conservative positive mixing
patterns with higher contributions of Atlantic water (Fig. 5a-c). At SG silicate was negatively correlated to
salinity showed a negative correlation for silicate pointing to a higher contributionconcentration ofin
glacial meltwater (R2=0.86, p<0.0001). The absence of but not positive relationscorrelations for NOX and
PO4 indicate non-conservative mixing pointing towards the relevance of biological uptake and release
measurements (Fig. 5d-f).¨

We also corrected the figure legend of Fig. 5 (See below).

310 "The contribution of nutrients by upwelling as well as freshwater inflow from glacial meltwater was
estimated by linear mixing calculations". Can you show these, maybe in the supplement, I am a little
confused mainly because of the unclear description above. Similarly for the % nutrient values, please
clarify how these were calculated and consider the error on them – especially if it is the case that the
single SG value with very high NOx and Si is basically dominating the trend and an outlier.

We added following calculations to the appendix. The mentioned outlier values in SG in the UIW sample
was not used for the mixing calculations as explained above. For the meltwater fraction at the surface
the error related to the average IE salinity is less than 0.1 % (see comment above), the main variation of
the % meltwater contribution in the surface layer of SG is related to the salinity at the surface of SG (Fig.
R1). We added the error estimate of 0.1 % to the table. For nutrients, the error was estimated based on
the variability in the concentrations measured in the triplicates. For NOx the estimated range of
contribution by upwelling is thereby 57-59 % (± 1 %) bottom water, for Silicate 89-95 % (± 3 %), and for
phosphate 46-49 % (± 3 %). We added the error estimates to the text and table.

Equations. Mixing calculations for estimates of the fraction of meltwater ($MW_{Sal}$) based on salinity, and
for bottom water based on nutrient concentrations ($BW_{Nuts}$). Sal indicates the average salinities measured
at the IE ($Sal_{IE}$), SG at 1m depth ($Sal_{SG1m}$), subglacial outflow ($Sal_{glac}$). Nut indicates the nutrient
concentrations of nitrate and nitrite (NOX), silicate (Si), and phosphate (PO4) at 1m under the sea ice at
SG ($Nut_{1mSG}$) and IE ($Nut_{1mIE}$), the bottom water of the IE ($Nut_{BW}$), or subglacial outflow water ($Nut_{glac}$).

$$MW_{Sal}[\%] = \frac{Sal_{IE} - Sal_{SG1m}}{Sal_{SG1m} - Sal_{glac} + Sal_{IE} - Sal_{SG1m}} * 100$$

$$MW_{Sal}[\%] = \frac{34.7\,PSU - 23.6\,PSU}{23.6\,PSU - 0\,PSU + 34.7\,PSU - 23.6 PSU} * 100 = 32\,\%$$

$$BW_{Nut}[\%] = \frac{Nut_{1mSG} - MW_{Sal}[\%] * Nut_{glac} - Nut_{1m_{IE}} + MW_{Sal}[\%] * Nut_{1m_{IE}}}{Nut_{BW} - Nut_{1m_{IE}}} * 100$$

$$BW_{NOX}[\%] = \frac{6.52\mu M - 0.32 * 2.06\,\mu M - 3.27\,\mu M + 0.32 * 3.27\,\mu M}{9.57\,\mu M - 3.27\,\mu M} * 100 = 58\,\%$$

$$BW_{Si}[\%] = \frac{4.30\,\mu M - 0.32 * 1.79\,\mu M - 1.59\,\mu M + 0.32 * 1.59\,\mu M}{4.46\,\mu M - 1.59\,\mu M} * 100 = 92\,\%$$

$$BW_{PO4}[\%] = \frac{0.41\,\mu M - 0.32 * 0.09\,\mu M - 0.34\,\mu M + 0.32 * 0.34\,\mu M}{0.67\,\mu M - 0.34\,\mu M} * 100 = 46\,\%$$

[Figure]

Figure R1. Estimated fractions of glacial meltwater in the surface layer of SG.
The vertical export flux of Chl a is based on Chl a measurements in the sediment traps. We first convert
the measured Chl concentrations (mg m$^{-3}$) to mass (mg) in order to calculate the flux as the mass of
Chlorophyll a per unit area and time sedimenting to a certain depth.
Change:
This leads to higher (14 times) vertical export flux based on the sediment trap measurements than
production at IE and considerably lower (5 %) export than production at SG (Table 2).
The sediment traps are cylindrical bottles, filled with sterile filtered water, incubated at different depths
for about 1 day. The material (e.g. Chl a) sedimenting out (vertical flux) is collected in these cylinders.
Since we know the concentration of Chl a in the sediment trap (**C** in mg m$^{-3}$) and the volume of the
cylinders (**V** in m3), we can calculate the mass of Chl a sedimenting into the trap (mg). With the knowledge
of the area above the sediment trap opening (**A** = m$^2$) we can calculate the amount of Chl per area (mg
m$^{-2}$). With the information of the exact incubation time (**t** in days), we can then calculate the vertical flux
(mg m$^{-2}$ d$^{-1}$). The calculation is described in Wiedmann et al. (2016), but we also added the equation below
to the appendix.

$$Vertical\ flux = \frac{C*V}{A*t}$$

We added our estimate in the same unit:
To our knowledge, our study provides currently the only available estimate of subglacial upwelling in early
spring. Our study suggests that subglacial upwelling in spring causes in Billefjorden a small volume
transport of only about >1.1 m3 m-2 month-1 (approx. 2 m3 s−1). …The most comparable estimate on
the magnitude of the upwelling is available at Kronebreen for summer. This Svalbard tidewater glacier is of similar size and had one to two order of magnitude higher upwelling rates compared to our study (31-
127 m3 s-1, Halbach et al., 2019).

462 'careful' implies errors/uncertainties are quantified, at the moment I would say it was a little crude.

We agree and did following change:

This estimate is based on the flux of nutrient rich bottom water needed to maintain the measured primary
production assuming steady state conditions and is therefore a rough, but conservative estimate.

465-468 There are 2 sentences here basically saying the same thing

We removed one of the sentences.

469 'depth of glacier front' it would be better to cite the physical studies which specifically show this
rather than a review

We cited now Carroll et al., 2016 instead. Carroll et al. (2016) also reviews different studies, but for
coming to a conclusion of the depth of the glacier front being related to the amount of upwelling, requires
a review, or meta-analyses. Since we use the citation as evidence for this specific relationship, we suggest
this review as most appropriate. The physical studies alone do not have sufficient data to come to this
conclusion.

Carroll, D., Sutherland, D. A., Hudson, B., Moon, T., Catania, G. A., Shroyer, E. L., Nash, J. D.,
Bartholomaus, T. C., Felikson, D., Stearns, L. A., Noël, B. P. Y., and van den Broeke, M. R.: The impact of
glacier geometry on meltwater plume structure and submarine melt in Greenland fjords, Geophys. Res.
Lett., 43, 9739–9748, https://doi.org/10.1002/2016GL070170, 2016.

470 It would be useful to mention the grounding line estimate earlier in the text

We added the information already in the methods description.

486 required for photosynthesis or something more specific (primary production can occur in the dark)

We added following clarification:
… where light is sufficient for photosynthesis.

519 Yes I would agree, but I don't think the review cited explicitly shows this. You can however find a lot
of work that suggests most of the Arctic is basically nitrate limited based on observed macronutrient
distributions in summer7, and I think this has been explicitly tested closed to Svalbard showing no
significant effect of Fe additions8.

We agree that the review is not the most appropriate reference and added the study by Krisch et al.
(2020) instead.

520 I'm not sure what you mean by nutrient concentrations are higher at shallower depths, something
due to relatively more evident benthic input as you mentioned for NH4 briefly?

We suggest that at a shallower water depth, less physical forcing not necessarily related to subglacial
upwelling (e.g. tides (low in Adolfbukta), currents, or wind (unlikely under sea ice), is needed for vertical
mixing leading bottom water to reach the surface.

We added following clarification:

Besides the subglacial upwelling, nutrient concentrations may simply be higher due to due to lower physical forcing and time needed for vertical mixing at the shallower water depth at SG compared to IE, facilitating vertical mixing down to the bottom.

As above, I think this is not quite right. Non-conservative silicic acid behavior (but conservative N/P behavior) would suggest glacier associated input from dissolution of glacier-derived particles either directly into the water column or from sediments into the water column, although conservative silicic acid behavior is also equally often observed downstream of glaciers so this is not really a clear universal meltwater signature9. I think the only generalizations you could make would concern concentrations that melt is generally expected to be a low or not significant source of nitrate/phosphate, and a more important source of silicic acid, see Ref1 and the supplement for a summary.

We did following correction:

The differences in the relation of nutrient concentrations versus salinity indicate, that glacial meltwater was not a major source for N and P at SG while the different relation for Si provide evidence for supply through meltwater inflow (Hopwood et al., 2020).

530. I'm not sure these values are low compared to Greenland work if you compare to the dissolved values in freshwater. See the supplement for Ref for a summary1, I suspect if you calculate mean/median for data available for Greenland or Svalbard your values are likely not atypical (for silicic acid especially, I think the mean and range is high because 1 or 2 catchments have exceptionally high concentrations, but median concentrations are likely a few micromolar.) Note spelling Hawkings (I assume).

We are quite confident that the values are low, but would be thankful if the reviewer has any suggestions for references with lower Silicate values in glacial outflow water in Greenland. Overall, the data for glacial outflow in Greenland are sparse. We do not think comparing Arctic rivers with our measurements of subglacial outflow would be useful, since additional processes, including additional weathering and uptake by freshwater diatoms would play a role. Overall, rivers have also higher Silicate values. The only samples with lower concentrations than our study are from icebergs (Meire et al., 2016a). The other values in the study by Meire et al. (2016a) are measured from glacial rivers, with the lowest value of 3.4 $\mu$mol L$^{-1}$, the lowest mean value of 5.5 $\mu$mol L$^{-1}$ and typical mean values around 10 $\mu$mol L$^{-1}$. For clarity, we added the values of our study and the range of the values from Greenland.

The nutrient concentrations in subglacial outflow water were lower (<1.5 – 2 $\mu$mol L$^{-1}$) than estimates in Greenland (Hawkings et al., 2017: 0.8-41.4 average 9.6 $\mu$mol L$^{-1}$ , Hatton et al., 2019: 9.2-56.9 average 20.8 $\mu$mol L$^{-1}$, Cape et al., 2019: 10 ± 8 $\mu$mol L$^{-1}$), indicating that direct fertilisation in spring may be even more important in other tidewater glacier influenced fjords.

This is curious, do you have any idea why? Based on the Ref10 cited, this source would be expected to be quite large (i.e. silicic acid entering solution from glacier mderived particles) compared to direct glacier inputs of dissolved silicic acid, but I'm not sure how much evidence there is for this, elsewhere around Svalbard I think the same summary can be made as herein that there doesn't seem to be strong evidence for a significant silicic acid source from glacier sediments6.

As indicated by rather low silicate concentrations in the subglacial outflow water, we suggest that the bedrock below Nordenskiøldbreen is overall poor in silicate, at least at the areas, where the subglacial drainage system is in contact with the bedrock. We did following change:

However, bottom water nutrient concentrations were similar at SG and IE, indicating a limited role of higher silicate inputs from sediment, presumably due to silicate-poor subglacial bedrock.

Besides: : : This is repetition, I don't think it's particularly controversial to assume NOx as the limiting nutrient in this environment11,12, I think a very brief comment about Fe would suffice, there is more than ample evidence for really high Fe inputs in and around Svalbard13,14 and low nitrate levels through most of the growing season.

We agree and removed the sentence, since the information about iron is already given above.

539 Given the lack of relevance for atmospheric inputs to under-ice blooms, I don't think you need to
discuss this, unless you are writing about incorporation of such nutrients into sea-ice

Since atmospheric inputs can be an important N source for sea ice algae, we kept the discussion.
Especially at the SG station, we found high biomass of sea ice algae higher up in the ice, indicating that
atmospheric inputs may play a role and need to be discussed.

549 I'm not sure which value in the cited review you're referring to here, it would be more useful to cite
the specific studies that measured primary production (there are many studied on primary production
for Svalbard including values specifically for spring which are presumably the best comparison)12,15,16.

The value is given in a table (Table 1 in Hopwood et al., 2020) and is based on many different studies (6
fjords, 33 datapoints), which makes citing one original research paper tricky. Discussing all studies
separately would repeat the review effort of Hopwood et al. (2020) and go beyond the scope of our
discussion. Thus, we kept the review article as main reference. We added however the range of PP in
tidewater influenced fjords for clarification.

For a comparison of Kongsfjorden as a similar system on Svalbard, we also agree that adding more specific
references to van de Poll et al. (2018) and Hodal et al. (2012) is helpful.

We did following changes:

The integrated primary production to 25 m at SG of 42.6 mg C m$^{-2}$ d$^{-1}$ is low compared to values from
other marine terminating glacier influenced fjord systems in summer with mean integrated NPP of 480
±403 mg C m$^{-2}$ d$^{-1}$ (reviewed by Hopwood et al., 2020), including studies in Kongsfjorden on Svalbard (250
-900 mg C m$^{-2}$ d$^{-1}$, Van de Poll et al. 2018). A study conducted during a similar time window as ours (1
May) observed higher primary production rates in a marine-terminating glacier influenced fjord system,
in Kongsfjorden (1520-1850 mg C m-2 d-1, Hodal et al., 20120).

570-575 As written this is fine, but please note I think the 'seed' hypothesis specifically referring to inner-
fjord communities seeding outer-fjord/shelf areas is not particularly well supported by literature,
especially since in the context of sea-ice covered fjords, I think the bloom generally occurs earlier outside
the fjord than it does inside (I'm not sure if that is the case here). Elsewhere on seasonal timescales there
is evidence of marine inflow changing the in-fjord bloom and not really of the opposite17,18.

Our main support is the paper by Hegseth et al. (2019), which describes microalgae derived from
sediment upwelling/mixing in the fjord as crucial source of inoculum for a spring bloom. Especially in
Billefjorden with little Atlantic water inflow due to the shallow sill, slow tidal currents, and a suspected
net advection away from the glacier front, we expect this mechanism to also be important in Billefjorden.
However, based on your literature, we realize that this hypothesis is not widely accepted and formulated
the statement more carefully.

We did a more careful discussion in the following way:

..., may be a viable seed community for summer phytoplankton blooms, once the sea ice disappears and
light levels increase (Hegseth et al., 2019).

585 These are averages you're referring to? It may be worth commenting on the variability, I expect
there's a huge range when you're writing about all Arctic glacier-fjords

We agree and gave the range instead of the average. We also add a reference citing the original study,
instead of the review by Leu et al. (2015).

Only Greenland fjords (0.1-3.3 mg Chl m$^{-2}$) or pre- and post-bloom systems had comparably low biomass (Mikkelsen et al., 2008, Leu et al., 2015).

'limited by phosphate' do you actually show this, or do you mean than based on measured concentrations, there was a deficiency of phosphate?

We changed the term "limited" to "deficient".

This appears very speculative, because I think you are comparing broad regional averages to a spot measurement?

We agree and realize that this discussion is not crucial for the paper and, thus, removed it.

-8.3 above average doesn't make sense
We referred to 8.3 not -8.3 and corrected the error.

Here, in this section, I think you need to consider where sea-ice cover occurs and also how that and the timing of its breakup may also change in the future.
We agree that sea-ice cover and changes with climate change need to be discussed here and did following additions:

Another impact of climate change will be the reduction and earlier break-up of sea ice and Atlantification of fjords, leading to increased light, and wind mixing. In the ice free Kongsfjorden, higher primary production rates have been measured in the same month, indicating that the lack of sea ice may lead to increased overall primary production (Iversen & Seuthe, 2010). However, Kongsfjorden is still influenced by subglacial upwelling, supplying nutrients for the bloom (Halbach et al., 2017). In systems not affected by subglacial upwelling the additional light will most likely not lead to substantially higher primary production as indicated by lower measured rates in these type of fjords (Hopwood et al., 2020). Since the entrainment in our study occurs at only approximately 20 m depth, upwelling under sea ice-free conditions would have much less effect, since wind induced vertical mixing plays a more important role. Direct silicate fertilisation would also have less effect in an ice-free fjord since the fjord phytoplankton biomass is likely more nitrate than silicate limited, due to the later stage of the spring bloom (Hegseth et al., 2019). In summary, we suggest that subglacial upwelling in early spring is important for phytoplankton blooms, but only in a sea-ice covered fjord. The future of the spring phytoplankton blooms depends on what happens first, disappearance of sea ice, or retreat of the glacier to land.

"the seed material from the deeper sediments would not reach the water column, leading to a reduced and delayed phytoplankton summer bloom" Whilst I've read this hypothesis in a few places, I'm not sure there's much evidence for this, can you cite studies specifically showing this does affect the summer bloom?

Our main support is the paper by Hegseth et al. (2019), which describes microalgae derived from sediment upwelling/mixing in the fjord as crucial source of inoculum for a spring bloom. Especially in Billefjorden with little Atlantic water inflow due to the shallow sill, slow tidal currents, and a suspected net advection away from the glacier front, we expect this mechanism to also be important in Billefjorden. However, since the support lies in another study and not in our data, we removed this sentence.

650-660 There are a lot of ideas in these paragraphs which are not extensively developed. I think it would be good to either develop these a bit more, or remove them. For the later comment, Holding et al., is probably the best ref I can think of – you also need to think about stratification19 if you want to write about changes in summertime, but I generally suggest you cut this given the spring focus of your manuscript. The writing concerning spring is much better developed and the comments concerning changes in summer bloom lack discussion of the many factors (changing discharge, stratification, circulation) that change seasonally and are generally beyond the scope of the manuscript.
In your comments about how significant/important this process is, maybe you could think about how it works with respect to the availability of nutrients and timing.

 If I understood correctly, the entrainment occurs from only 20 m depth, so if it started slightly later in the
season it would be presumably much less effective as nitrate would already have been drawdown and
meltwater would just be mixing into an already nutrient deficient top 20 m layer? Presumably this means
the relative timing of bloom onset, and early discharge is an important feature to think about in
determining when/when this is important?
(And, also sea ice break up, the dates of which presumably are also changing?)

We removed all references to summer and focus on spring changes and extend our discussion on sea-ice
retreat, timing of the bloom and sea-ice retreat vs glacier retreat effects in the following way (See
response to comment on line 644):

Another impact of climate change will be the reduction ==and earlier break-up== of sea ice and Atlantification
of fjords, leading to increased light, and wind mixing. In the ice free Kongsfjorden, higher primary
production rates have been measured in the same month, indicating that the lack of sea ice may lead to
increased overall primary production (Iversen & Seuthe, 2010). However, Kongsfjorden is still influenced
by subglacial upwelling, supplying nutrients for the bloom (Halbach et al., 2017). In systems not affected
by subglacial upwelling the additional light will most likely not lead to substantially higher primary
production as indicated by lower measured rates in these type of fjords (Hopwood et al., 2020). ==Since the==
==entrainment in our study occurs at only approximately 20 m depth, upwelling under sea ice-free==
==conditions would have much less effect, since wind induced vertical mixing plays a more important role.==
==Direct silicate fertilisation would also have less effect in an ice-free fjord since the fjord phytoplankton==
==biomass is likely more nitrate than silicate limited, due to the later stage of the spring bloom (Hegseth et==
==al., 2019). In summary, we suggest that subglacial upwelling in early spring is important for phytoplankton==
==blooms, but only in a sea-ice covered fjord. The future of the spring phytoplankton blooms depends on==
==what happens first, disappearance of sea ice, or retreat of the glacier to land.==

Data files: These are generally well organized but I could not find the nutrient data in the file which the
readme says it is in, did I miss something? -> I will double check after PhD submission( The same for
finishing the ENA submission)

We added the missing data to the DATAVERSE archive.

Fig. 3 The blue line doesn't quite display properly in my version

We uploaded a figure with higher quality and thicker lines. For the final paper, we will submit vector
based PDF files for each figure.

Fig. 4 There are a couple of suspect anomalies here, along the line that represents the ice boundary there
are a few nutrient concentrations that appear well above the trend for either ice or water column
concentrations, are you sure these are real? -> mentioned the outliers in the results

As mentioned above, these values are based on 1 sample (UIW at SG for NOX and Silicate) and may well
be outliers or anomalies based on sampling artifacts, or locally high remineralization/dissolution rates.
Thus, we highlight them as outliers in the text and do not use them for the mixing calculations, or detailed
discussions.

Fig. 5 As in text, the description of 'conservative mixing' isn't quite right. "Conservative mixing shows as
a positive correlation, non-conservative mixing as a negative correlation". The strength of the correlation
indicates roughly how conservative it is. The sign of the gradient indicates whether the concentrations
are increasing or decreasing with salinity i.e. whether freshwater or saline water has the higher
concentration. It would be useful to have the actual p values written somewhere.

As mentioned above, we agree and changed the text in the manuscript and figure legend accordingly.

Change in the figure legend:

Fig 5. Linear salinity-nutrient correlations of NG and IE water samples (a–c), NG, IE, and SG water
stations (d–f) and sea ice samples of NG, IE and SG (g–i). A higher concentration in saline Atlantic water
results in a positive correlation, a higher concentration in glacial meltwater in a negative correlation.
Significant correlations ($p < 0.05$) are asterisk marked behind the $R^2$ value.

Fig. 6 This took a while to read, there are a lot of abbreviations.

Due to the large amount of data in this figure, we argue that the amount of text, containing information
and assumptions in the methods are necessary. We wrote out the abbreviations on top, to make the
figure more understandable without reading the legend. We also increased the font size of the numbers
within the figure.

(1) Meire, L.; Meire, P.; Struyf, E.; Krawczyk, D. W.; Arendt, K. E.; Yde, J. C.; Juul Pedersen, T.; Hopwood,
M. J.; Rysgaard, S.; Meysman, F. J. R. High Export of Dissolved Silica from the Greenland Ice Sheet.
Geophys. Res. Lett. 2016, 43 (17), 9173–9182. https://doi.org/10.1002/2016GL070191.

(2) Meire, L.; Mortensen, J.; Rysgaard, S.; Bendtsen, J.; Boone, W.; Meire, P.; Meysman, F. J. R. Spring
Bloom Dynamics in a Subarctic Fjord Influenced by Tidewater Outlet Glaciers (Godthåbsfjord, SW
Greenland). J. Geophys. Res. Biogeosciences 2016, 121 (6), 1581–1592.
https://doi.org/10.1002/2015JG003240.

(3) Schaffer, J.; Kanzow, T.; von Appen, W. J.; von Albedyll, L.; Arndt, J. E.; Roberts, D.
H. Bathymetry Constrains Ocean Heat Supply to Greenland's Largest Glacier Tongue.
Nat. Geosci. 2020. https://doi.org/10.1038/s41561-019-0529-x.

(4) Carroll, D.; Sutherland, D. A.; Hudson, B.; Moon, T.; Catania, G. A.; Shroyer, E.
L.; Nash, J. D.; Bartholomaus, T. C.; Felikson, D.; Stearns, L. A.; Noël, B. P. Y.; van
den Broeke, M. R. The Impact of Glacier Geometry on Meltwater Plume Structure and
Submarine Melt in Greenland Fjords. Geophys. Res. Lett. 2016, 43 (18), 9739–9748.
https://doi.org/10.1002/2016GL070170.

(5) Cape, M. R.; Straneo, F.; Beaird, N.; Bundy, R. M.; Charette, M. A. Nutrient Release
to Oceans from Buoyancy-Driven Upwelling at Greenland Tidewater Glaciers. Nat.
Geosci. 2018. https://doi.org/10.1038/s41561-018-0268-4.

(6) Cantoni, C.; Hopwood, M. J.; Clarke, J. S.; Chiggiato, J.; Achterberg, E. P.; Cozzi,
S. Glacial Drivers of Marine Biogeochemistry Indicate a Future Shift to More Corrosive
Conditions in an Arctic Fjord. J. Geophys. Res. Biogeosciences 2020, n/a (n/a), e2020JG005633.
https://doi.org/https://doi.org/10.1029/2020JG005633.

(7) Codispoti, L. A.; Kelly, V.; Thessen, A.; Matrai, P.; Suttles, S.; Hill, V.; Steele,
M.; Light, B. Synthesis of Primary Production in the Arctic Ocean: III. Nitrate and
Phosphate Based Estimates of Net Community Production. Prog. Oceanogr. 2013.
https://doi.org/10.1016/j.pocean.2012.11.006.

(8) Krisch, S.; Browning, T. J.; Graeve, M.; Ludwichowski, K.-U.; Lodeiro, P.; Hopwood,
M. J.; Roig, S.; Yong, J.-C.; Kanzow, T.; Achterberg, E. P. The Influence of Arctic Fe and
Atlantic Fixed N on Summertime Primary Production in Fram Strait, North Greenland
Sea. Sci. Rep. 2020, 10 (1), 15230. https://doi.org/10.1038/s41598-020-72100-9.

(9) Brown, M. T.; Lippiatt, S. M.; Bruland, K. W. Dissolved Aluminum, Particulate
Aluminum, and Silicic Acid in Northern Gulf of Alaska Coastal Waters:
Glacial/Riverine Inputs and Extreme Reactivity. Mar. Chem. 2010, 122 (1–4), 160–
175.

(10) Hawkings, J. R.; Wadham, J. L.; Benning, L. G.; Hendry, K. R.; Tranter, M.; Tedstone,
A.; Nienow, P.; Raiswell, R. Ice Sheets as a Missing Source of Silica to the Polar
Oceans. Nat. Commun. 2017, 8, 14198.
(11) van De Poll,W. H.; Maat, D. S.; Fischer, P.; Rozema, P. D.; Daly, O. B.; Koppelle, S.;
Visser, R. J.W.; Buma, A. G. J. Atlantic Advection Driven Changes in Glacial Meltwater:
Effects on Phytoplankton Chlorophyll-a and Taxonomic Composition in Kongsfjorden,
Spitsbergen . Frontiers in Marine Science . 2016, p 200.
(12) Van De Poll, W. H.; Kulk, G.; Rozema, P. D.; Brussaard, C. P. D.; Visser, R. J.
W.; Buma, A. G. J. Contrasting Glacial Meltwater Effects on Post-Bloom Phytoplankton
on Temporal and Spatial Scales in Kongsfjorden, Spitsbergen. Elementa 2018.
https://doi.org/10.1525/elementa.307.
(13) Herbert, L. C.; Riedinger, N.; Michaud, A. B.; Laufer, K.; Røy, H.; Jørgensen, B. B.;
Heilbrun, C.; Aller, R. C.; Cochran, J. K.; Wehrmann, L. M. Glacial Controls on Redox- Sensitive Trace
Element Cycling in Arctic Fjord Sediments (Spitsbergen, Svalbard).
Geochim. Cosmochim. Acta 2020. https://doi.org/10.1016/j.gca.2019.12.005.
(14) Hopwood, M. J.; Cantoni, C.; Clarke, J. S.; Cozzi, S.; Achterberg, E. P. The Heterogeneous
Nature of Fe Delivery from Melting Icebergs. Geochemical Perspect. Lett.
2017, 3 (2), 200–209. https://doi.org/10.7185/geochemlet.1723.
(15) Hop, H.; Pearson, T.; Hegseth, E. N.; Kovacs, K. M.; Wiencke, C.; Kwasniewski,
S.; Eiane, K.; Mehlum, F.; Gulliksen, B.; Wlodarska-Kowalczuk, M.; Lydersen, C.; Weslawski,
J. M.; Cochrane, S.; Gabrielsen, G. W.; Leakey, R. J. G.; Lønne, O. J.; Zajaczkowski,
M.; Falk-Petersen, S.; Kendall, M.; Wängberg, S.-Å.; Bischof, K.; Voronkov,
A. Y.; Kovaltchouk, N. A.; Wiktor, J.; Poltermann, M.; Prisco, G.; Papucci, C.; Gerland,
S. The Marine Ecosystem of Kongsfjorden, Svalbard. Polar Res. 2002, 21 (1), 167–
208. https://doi.org/10.1111/j.1751-8369.2002.tb00073.x.
(16) Hodal, H.; Falk-Petersen, S.; Hop, H.; Kristiansen, S.; Reigstad, M. Spring Bloom
Dynamics in Kongsfjorden, Svalbard: Nutrients, Phytoplankton, Protozoans and Primary
Production. Polar Biol. 2012, 35 (2), 191–203. https://doi.org/10.1007/s00300-
011-1053-7.
(17) Krawczyk, D. W.; Witkowski, A.; Juul-Pedersen, T.; Arendt, K. E.; Mortensen, J.;
Rysgaard, S. Microplankton Succession in a SW Greenland Tidewater Glacial Fjord
Influenced by Coastal Inflows and Run-off from the Greenland Ice Sheet. Polar Biol.
2015, 38 (9), 1515–1533. https://doi.org/10.1007/s00300-015-1715-y.
(18) Hegseth, E. N.; Tverberg, V. Effect of Atlantic Water Inflow on Timing of the Phytoplankton
Spring Bloom in a High Arctic Fjord (Kongsfjorden, Svalbard). J. Mar. Syst.
2013, 113–114, 94–105. https://doi.org/https://doi.org/10.1016/j.jmarsys.2013.01.003.
(19) Holding, J. M.; Markager, S.; Juul-Pedersen, T.; Paulsen, M. L.; Møller, E.
F.; Meire, L.; Sejr, M. K. Seasonal and Spatial Patterns of Primary Production in a
High-Latitude Fjord Affected by Greenland Ice Sheet Run-Off. Biogeosciences 2019.
https://doi.org/10.5194/bg-16-3777-2019.

**Author's response to:**

**Anonymous Referee #2**

Subglacial upwelling in winter/spring increases under-ice primary production

Summary: This paper aims to explore the role of the release of subglacial meltwater in the winter and
spring on under-ice primary production. The premise of the study is that though subglacial upwelling of
nutrient rich deep marine waters has been shown to be a viable mechanism for stimulating primary
production in the summer, very few studies have examined this topic with regards to spring under-ice
primary production. The study is an interesting, under-explored topic, which is only likely to become
more important with global warming and prolonged glacial melt seasons, and thus, worthy of eventual
publication in this journal.

We want to thank the reviewer sincerely for the comprehensive review that helps to improve the clarity
of the manuscript considerably. All comments are clear and very useful. We addressed all comments as
outlined in detail below. Changes in the text of the manuscript are highlighted in green.

However, I think there are number of improvements that could be made to aid the study, which I outline
below. Apart from issues with over-interpretation of the data (detailed below), the writing is often
disorganized and unclear. Also, there often a lack of consideration of the on-ice processes that are
occurring that could be affecting the authors interpretations – i.e. enrichment of the glacial meltwater
itself that has been stored at the bed overwinter and is released in the spring. The fact that the submarine
discharge in the spring is likely quite different to the dilute discharge characteristic of summer drainage
is a fact that makes this difficult to compare to previous summer studies of glacial discharge into the
ocean. To this end often the authors seem to have a pre-ordained conclusion – i.e. that the mechanism
of nutrient addition was via upwelling of "deep" bottom waters by the submarine discharge, but this
seemed at odd with the shallow depth of this discharge (20-m). Finally, there is also a lack of clarity
with how some of the calculations are made – this needs to be rectified for these calculations to be
understood. I would urge the authors to address these points, and indeed try to focus their story on the
novel spring measurements they have, to maximize the potential readership of this interesting study.

In cases of over-interpretations, we either rephrased the interpretation more careful, often via adding
details for clarification, or removed the statements (details below). We rewrote the sections that the
reviewer considered disorganized and unclear (details below) with the most substantial changes regarding glacial processes and the chapter about subglacial upwelling and entrainment factors. We tried
to clarify the relevance of on-ice processes by i) introducing the processes in more detail in the
introduction, ii) mentioning the nutrient concentrations of the undiluted subglacial meltwater that we
measured earlier in the results, and iii) giving more references to the role of nutrient enrichment under
the glacier (weathering during bedrock contact, solute expulsion during freezing). However, our nutrient
measurements of the undiluted meltwater still showed lower concentrations compared to the fjord
bottom water. The concentrations are enriched compared to sea ice and UIW samples at NG and IE, but
we consider upwelling of bottom water more important for nutrient dynamics in this area. We further
clarified these findings by referring more strongly to the undiluted meltwater nutrient concentrations in
the text. Please note that Svalbard studies by van der Poll (e.g. van der Poll et al., 2018) agree with our
conclusion. Referee 2 suggests that shallow water depth might limit the relevance of this process. We
suggest that the freshwater input occurred below the sharp halocline in 4-5m depth, explaining the
nutrient differences between 15 and 1 m. Additionally this process is supported through a) the absence
of any substantial external advection of inorganic nutrients (e.g through tides and wind), and b) strong
salinity driven stratification preventing mixing apart from upwelling. Detailed calculations were added
to the appendix.

Title: Given the confusion regarding subglacial upwelling (see below) – do you mean submarine
discharge or upwelling of deeper marine waters? – I would suggest a title change.. Perhaps: Spring
submarine discharge plumes fuel under-ice primary production ?

This has been a very good suggestion by the referee – we agree and changed the title accordingly to "Early spring submarine discharge plumes fuel under-ice primary production"

Abstract:

L25: "retreat of tidewater glaciers could lead to decreased under-ice phytoplankton primary production" when? in the spring? In winter? Or both? My comment on the line above points to a broader problem which is evident in the title.. which is that I think by the lack of specificity regarding the timing, winter or spring is determinantal to the paper. Presumably if the focus is on spring primary production then the authors are speaking about subglacial upwelling in the spring?

Based on the simple date definition spring starts at the 20th of March. However, the definition of winter and spring is more difficult in Arctic studies, as biological processes like algal blooms are not tight to the calendar but to changes in e.g. light and ice regime. Also algal growth (as indicator of spring) in the ice might occur prior to algal growth in the water column. Spring may also be defined as the onset of snowmelt and temperatures above freezing point (mostly in terrestrial ecology), or by the return of light. Since we sampled at a time of subzero temperatures and ice cover (winter), but with sufficient light for algae blooms (spring), we had used the term "winter/spring" in the submitted version. However, since light availabililty is often most important in Arctic marine systems to define the onset of spring we changed the term to "early spring" throughout the manuscript and added the information where it was lacking (including L25).

*A minor point, but line numbers every line would be really very helpful*

We added the line numbers as suggested.

Introduction:

L37: unclear what "it" is referring too

We replaced "it" with "the submarine discharge"

L39: "close to the glacier front".. meaning what? Suggest specifying. Also a reference would be helpful here. The ranges of increased primary production in front of tidewater glaciers is quite variable so specification would be good.

The exact distance is highly variable, depending on sediment load, glacier terminus depth, discharge volume and flux e.g.. Hence, it is not possible to provide accurate numbers. However, based on an earlier study (Halbach et al., 2019) which found a phytoplankton bloom at 0.1 -1.9km distance from the glacier, we included a distance range into the manuscript (hundreds of meters to kilometers).

L41: "at some distance" .. again suggest specifying here.

See comment above

L46: I'm not sure I would necessarily agree that the lack of studies of subglacial discharge in the winter / spring is due to the perception of a lack of freshwater outflow. I think it's well known from a glacier hydrological perspective that temperate and even polythermal ice masses likely have winter / spring discharge. More likely it's due to a lack of opportunity given the challenge of Arctic field conditions and the difficulty in locating such an outflow which would presumably be of low flux.

We agree and changed this statement in the following way: " Due to the challenges of Arctic field work in early spring and the difficulties of locating such an outflow, only few studies investigated submarine discharge during that time window. The few studies available suggest overall little discharge (e.g. Fransson et al., 2020; Schaffer et al., 2020) compared to summer values. The limited amount of data makes the generalized quantification of subglacial outflow difficult. In addition, studies focusing on the potential impacts of the early spring discharge on sea ice and pelagic primary production are lacking."

L52-53: Suggest defining what you mean by "Glacier terminus melt rates"

The term "glacier terminus melt rate" is adopted from the mentioned publications, but we added a short clarification. "Glacier terminus melt rate occurring at the glacier-marine interface".

L54: Svalbard glaciers are shallower compared to what?

They (the water depth at the glacier terminus) are shallower than Greenland glaciers. We added following clarification: "submarine glacier termina on Svalbard occur typically at shallower water depths than on Greenland …"

L55-56: Phrase "can persist throughout winter and specifically in early spring" is unclear. Are you suggesting that outflow persists through winter and into spring?

We included the suggested sentence by the referee and rephrased the sentence in the following way: "can persist through winter and into spring"

L57: add phrase " various other mechanisms such as:" between the words "through" and "constant". Also suggest making the part re: temperate parts of the glacier" a discrete sentence. Presumably, with regards to winter / spring discharge you are speaking about polythermal glaciers? I think this section in general needs more specifics regarding the types of glaciers that typically have winter/spring discharge and the typical fluxes and chemical composition of this discharge. I would think that all of these points are worth mentioning to set-up the discussion of this paper. The point regarding chemical composition in particular has been glossed over as being sourced from meltwater stored from the previous melt season but this meltwater having been stored at the bed over winter would have a significantly different chemical character than dilute snow and ice-melt passing quickly through the system at the height of summer. Also, what about the possibility of basal ice melt?

We replaced the sentence with a more comprehensive paragraph addressing the missing information and background: "However, subglacial outflows can persist through winter and into spring through the release of subglacial meltwater stored from the previous summer and fall melt season as observed in several Svalbard glaciers, including cold-based glaciers (Hodgkins, 1997). Winter drainage occurred mostly periodically during events of ice-dam breakage. During the storage period, the meltwater can change its chemical composition. For example, prolonged contact with silicon-rich bedrock increased the silicate concentrations (Hodgkins, 1997). Additionally, freezing of some of the meltwater leads to higher ion concentrations in the remaining liquid fraction (Hodgkins, 1997). Under polythermal glaciers, various additional mechanisms such as supply from groundwater, and basal ice melt via geothermal heat, pressure, or frictional dissipation can also contribute to a continuous but low flux meltwater source in winter and spring (Schoof et al., 2014)."

L59-60: "Even low rates of subglacial outflows can be sufficient to supply nutrients to the surface".. why? How? Is it because they would be sufficiently deep enough in the water column? Are you speaking of supplying nutrients via upwelling or via direct addition of nutrients in the subglacial discharge itself? If only the former, how can the latter be discounted since subglacial discharge in the spring would likely be more chemically enriched from greater contact times with the glacier bed or being sourced from basal ice melt?

We suggest that low supply rates via upwelling can have a considerable impact due to the absence of other sources in a sea ice covered fjord with very weak advection (tidal currents, wind, Atlantic water) and a strongly stratified water column. Direct addition can of course also play a role. We added the following clarification: "We hypothesize that subglacial discharge can lead to significantly increased primary production, due to upwelling of nutrient rich deeper water or through its own nutrient load, especially towards the end …"

L60: Why would spring subglacial discharge contain less sediment.. b/c of the low fluxes? Suggest specifying why.

The referee is correct in his/her suggestion. The reduction is likely caused by the low fluxes and thereby reduced advective forcing. We added a reference to a study measuring the seasonal variation of sediment outputs at a Svalbard tidewater glacier as additional support as described in the response to RW1 (Moskalik et al., 2018) and added a specification of "due to lower fluxes".

L63: Suggest setting up this argument a bit more progressively. Explain first what nutrients are generally fueling the under-ice spring bloom initially, and then go into the timing of glacial discharge and how that might positively affect under-ice primary production. As of now, the timing of the discharge and the initial bloom and end of bloom period are all not clearly laid out and this is problematic (in my opinion).

We did following additions: "With the return of the sunlight after the polar night, Arctic ice algae and phytoplankton start forming blooms sustained by the winter mixing replenished nutrients with different onsets in different parts of the Arctic. The blooms are typically terminated by limitation of macronutrients, mainly nitrate or silicate (Leu et al., 2015). We suggest that in the absence of wind induced mixing, due to the seasonal presence of fast ice cover in spring, submarine discharge of glacial meltwater can directly (nutrient ion enrichment over the subglacial storage period) or indirectly (upwelling) be a significant source of inorganic nutrient increasing primary production in front of tidewater glaciers compared to similar fjords without these glaciers. Especially after nutrients supplied via winter mixing are incorporated into algal biomass (Leu et al. 2015) this additional nutrient source may become important."

L67: delete "the" before "primary" and add "in front of tidewater glaciers" after the word "production"

We changed the sentence accordingly.

L70: Re-arrange /re-write sentence to: Once sufficient light penetrates the snow and ice layers, ice algae start growing within sea ice between March and April: : :. Etc"

We changed the sentence accordingly

L73: "nutrient additions from the water column" .. via what? How? Suggest specifying.

We replaced "nutrient addition" with "advection of nutrient-rich seawater" for clarification

L74: "subglacial upwelling" .. does this refer to spring subglacial upwelling? Suggest specifying. Again, I find the timeline within the year confusing with regards to glacial meltwater discharge and effect on bloom dynamics. Suggest more clearly spelling all of this out above.

Yes, we refer to spring. We added the term "early spring" for clarification.

L78: "or at the ice edge related to ice edge induced upwelling" .. can you define this upwelling without using the words "ice edge"?

We used the term "wind-induced Ekman upwelling" as described by Mundy et al. (2009).

L79: suggest replacing "coverage also" with "accumulates"

We replaced "coverage also" with "accumulation"

L81: suggest replacing "Once" with "After"

We change the term "Once" with "After" as suggested.

L83: suggest replacing "related" with "induced"

We change the term "related" with "induced" as suggested.

L86: suggest deleting "to" and replacing "fuel" with "fueling"

We change the formulation "to fuel" with "fueling" as suggested.

L87: the word "slow" is curious .. why is the subglacial upwelling slow? How do you know it's slow vs fast or continuous vs intermittent? Suggest deleting this word as it opens up a range of topics that haven't been discussed in enough detail above to warrant the use of this adjective here.

We replaced "slow" with "of low total flux", which would include continuous and intermittent discharge.

L86- 88: This pivot in this last sentence doesn't make a lot of sense to me as it seems to not really address the points brought up by the sentences immediately preceding it: : : i.e. namely reduced algal biomass due to brackish ice conditions .. suggest rectifying this last sentence.

We agree with the referee to change this section. We removed the last part of the sentence "…and cause different succession patterns for phytoplankton and sea ice algae." Since the succession patters are not clearly introduced or explained and not a major objective of the paper.

L90-91: How are the 2 freshwater inputs different? Suggest specifying versus keeping your reader in the dark here.

We replaced "with different freshwater inputs" with "with only one glacier front supplying submarine freshwater discharge". We agree that the previous formulation is unclear and misleading, since we mostly argue for the absence of freshwater inputs at NG.

L92: "to investigate the effect of the glacier terminus" .. this is a big vague. Suggest specifying.

We added following details: "… to investigate the effect of the glacier terminus, and subglacial outflow related upwelling on the light and nutrient regime in the fjord and thereby on early spring primary productivity…"

L94: "nutrient rich meltwater".. I'm unclear what you are referring to here.. presumably since this phrase is followed by "bottom water to the surface" I think by nutrient-rich meltwater you are referring to the subglacial discharge being enriched itself in nutrients versus upwelling of bottom waters but this has not been addressed above (though I suggest doing so)

We refer to the meltwater coming from the glacier itself. We added following clarification: "nutrient rich glacial meltwater" and "upwelling of marine bottom water"

L95: added "under ice" before the words "primary production" if this is indeed what you are referring too?

We added the formulation "under ice" as suggested.

L95: "near the glacier front".. phrase is vague. Suggest specifying.

We added a distance estimate in the following way: "near (<500 m) the glacier front".

L95-96: "low permeability of sea ice" .. phrase is also vague. Suggest specifying. As noted above I think the introduction would benefit from some more specificity, especially regarding the types of glaciers where winter / spring discharge might occur, a timeline of how this discharge evolves from end of the season to the winter and spring, and how this discharge might affect spring bloom under-ice dynamics – considering both the possibility of upwelling of bottom waters and also addition of nutrients directly from the glacial meltwater itself as alluded to in the last paragraph. One thing that should also be likely addressed is that any spring discharge will presumably be of quite low flux.. given this how likely / effective will any upwelling be?

We added following specification: "as a result of low permeability sea ice limiting nutrient exchange and inhabitable space"

As mentioned above (RW comment on L57 and L63), we also added a more detailed introduction of the
potential discharge of different glacier types and the chemical characteristics of fresh vs stored subglacial
meltwater with a potential of direct nutrient input with the meltwater. We also added the statement of
low fluxes in spring as mentioned above (RW comment on 87). We believe we explained the role of
lower salinity waters for forming less permeable sea ice already in former lines 84ff. We added the
following clarifications:  "We also suggest that the unique sea ice features could increase the under-ice
light intensity. Sea ice formed from brackish water has a low bulk salinity, brine volume fraction and
permeability (Golden et al., 1998) and resulting low total ice algal biomass as observed e.g. in the Baltic
Sea (Haecky & Andersson, 1999). This lower algal biomass will reduce ice algal light absorption
allowing more light to reach the under-ice phytoplankton."

Reference: Golden KM, Ackley SF, Lytle VI (1998) The percolation phase transition in sea ice. Science
282:2238-2241

Methods:

L120: ".. were melted in 50% vol/vol sterile filtered seawater: : :" what was the reasoning for this?

Sea ice is commonly melted in 50% vol/vol sterile seawater in order to avoid osmotic shock. Since most
sea ice organisms live in the brine channels with high salinity, but the salinity of a melted bulk ice core
is very low, direct melting leads to osmolysis. We added following sentence for clarification: "…to
avoid osmotic shock of cells (Garrison and Buck 1986)"

L155-157: Estimates of bottom water fractional contributions based on conservative mixing of nitrate..
can you rule out nitrate addition from the glacial meltwater itself? Other studies have found this (see,
Beaton et al., 2017 in ES&T: https://pubs.acs.org/doi/abs/10.1021/acs.est.7b03121), especially in the
early season meltwater from a distributed subglacial drainage system.

We realize that our formulation was not clear. We also measured NOx concentrations from the subglacial
outflow itself. We found subglacial outflow water exiting the glacier and sampled it directly (Salinity
0). The nutrient values of the glacial outflow, bottom water, and surface water were used for the
calculations. We added following clarification in the methods text: "assuming linear mixing (Equations
1-2) of the two salinities (glacial meltwater salinity = 0 PSU, average seawater salinity at IE=34.7 PSU
± 0.03 standard deviation), since no other water masses in regard to temperature or salinity signature
were present (Table 1)."

As mentioned by RW1 we added details and equations on how the mixing calculations were done.

In the manuscript we added the equations to the appendix, we added the error estimates in Table 1, and
we added details about the different water types in the header of Table 1.

Here the response to RW1 which outlines our changes:

"We added following calculations to the appendix. The mentioned outlier values in SG in the UIW sample
was not used for the mixing calculations as explained above. For the meltwater fraction at the surface the
error related to the average IE salinity is less than 0.1 % (see comment above), the main variation of the
% meltwater contribution in the surface layer of SG is related to the salinity at the surface of SG (Fig.
R1). We added the error estimate of 0.1 % to the table. For nutrients, the error was estimated based on
the variability in the concentrations measured in the triplicates. For NOx the estimated range of
contribution by upwelling is thereby 57-59 % (± 1 %) bottom water, for Silicate 89-95 % (± 3 %), and
for phosphate 46-49 % (± 3 %).

Equations. Mixing calculations for estimates of the fraction of meltwater (MWSal) based on salinity, and
for bottom water based on nutrient concentrations (BWNuts). Sal indicates the average salinities
measured at the IE (SalIE), SG at 1m depth (SalSG1m), subglacial outflow (Salglac). Nut indicates the
nutrient concentrations of nitrate and nitrite (NOX), silicate (Si), and phosphate (PO4) at 1m under the
sea ice at SG (Nut1mSG) and IE (Nut1mIE), the bottom water of the IE (NutBW), or subglacial outflow
water (Nutglac).

$$MW_{Sal}[\%] = \frac{Sal_{IE} - Sal_{SG1m}}{Sal_{SG1m} - Sal_{glac} + Sal_{IE} - Sal_{SG1m}} * 100$$

$$MW_{Sal}[\%] = \frac{34.7\ PSU - 23.6\ PSU}{23.6\ PSU - 0\ PSU + 34.7\ PSU - 23.6 PSU} * 100 = 32\ \%$$

$$BW_{Nut}[\%] = \frac{Nut_{1mSG} - MW_{Sal}[\%] * Nut_{glac} - Nut_{1m_{IE}} + MW_{Sal}[\%] * Nut_{1m_{IE}}}{Nut_{BW} - Nut_{1m_{IE}}} * 100$$

$$BW_{NOX}[\%] = \frac{6.52 \mu M - 0.32 * 2.06\ \mu M - 3.27\ \mu M + 0.32 * 3.27\ \mu M}{9.57\ \mu M - 3.27\ \mu M} * 100 = 58\ \%$$

$$BW_{Si}[\%] = \frac{4.30\ \mu M - 0.32 * 1.79\ \mu M - 1.59\ \mu M + 0.32 * 1.59\ \mu M}{4.46\ \mu M - 1.59\ \mu M} * 100 = 92\ \%$$

$$BW_{PO4}[\%] = \frac{0.41\ \mu M - 0.32 * 0.09\ \mu M - 0.34\ \mu M + 0.32 * 0.34\ \mu M}{0.67\ \mu M - 0.34\ \mu M} * 100 = 46\ \%$$

Change in Table 1:

Table 1. Properties of 1) marine surface and 2) Marine deep water (both station IE), 3) subglacial discharge melt water and 4) station SG surface water and the relative contribution of the water types 1 to 3 to form water type 4. The calculations are given in Equations 1-6 and are based on different salinities and nutrients in the 4 water masses.

| | 1) Surface water (IE 1m) | | 2) Bottom water (IE) | | 3) Subglacial discharge Meltwater | | 4) SG (1 m) |
|---|---|---|---|---|---|---|---|
| Salinity [PSU] | 34.7 | | 34.7 | | 0 | 32 ± 0.1 % | 23.6 |
| Temperature [°C] | -1.4 | | -1.4 | | 0 | | -0.4 |
| Silicate [µmol L⁻¹] | 1.59 | 0 % | 4.46 | > 84 % | 1.79 | 32 % | 4.30 |
| NOₓ [µmol L⁻¹] | 3.27 | 10 ± 3 % | 9.57 | 58 ± 1 % | 2.06 | 32 % | 6.52 |
| Phosphate [µmol L⁻¹] | 0.34 | 19 ± 3 % | 0.67 | 49 ± 3 % | 0.09 | 32 % | 0.42 |

L215: I'm confused by the words "reciprocal transplant experiment" .. I don't think a "transplant experiment" is described above.. just primary production incubations. I also find the description of this experiment (L215-218) unclear and thus the overall purpose of the experiments to the study also unclear. As written, I cannot assess these experiments so I'd suggest a re-write of this paragraph.

The words "reciprocal transplant experiment" is mostly used in plant ecology, when plants are planted/grown on different soil/ environments in order to see if the different soil/ environment has an effect on their fitness or growth. We did an analogue experiment in which we incubated algae communities in different water/ environments in order to test if the water chemistry has an effect on algae growth. We considered other more descriptive terms such as "water exchange experiment", but prefer keeping the term "reciprocal transplant experiment" due to its established and wide use in ecology. We rewrote the paragraph to clarify the experimental design in the following way:

"For testing the effect of the water chemistry on phytoplankton growth, we designed a reciprocal transplant primary production experiment where the phytoplankton communities at SG and IE (1 m and 15 m) each were transplanted into sterile filtered water of both SG and IE. 50 ml of the water containing the respective original phytoplankton community were transferred into 50 ml sterile filtered (0.2 μm) seawater of SG or IE each in 100 ml polyethylene bottles. The bottles were then incubated in situ at the original depth and primary production measured as described above. The aim of the experiment is to test if water chemistry alone is sufficient to increase primary production, or if differences in algal communities, light regimes, or temperatures are more important. These samples were incubated and processed together with the other PP incubations at the adequate depths as described above."

L225: Unclear what map you are referring to in sentence starting with "The map.."

We refer to the map in Figure 1 and added the figure reference. (Fig. 1)

L232: I'm wondering why you chose to you swarm to cluster versus amplicon sequence variants (see Callahan et al., 2017: https://www.nature.com/articles/ismej2017119)

We are familiar with both approaches. ASVs would indeed give more details on ecotype level. However, the aim of the study was not to dive into detailed taxonomic differences and identities, but to a) identify larger groups (e.g. flagellates, diatoms) and their potential functions and ecological role in relation to the biogeochemical data and b) to show and discuss overall community differences between the samples/sites. For this purpose we believe that swarm clustering of OTUs is appropriate.

L235: Was the data trans-formed in anyway before making the dissimilarity matrix? I'm only asking because it seems doing some type of transformation (e.g. Hellinger) is increasingly common.

We did do Square root transformations and Wisconsin double standardizations and added this for clarity to the text. "… (NMDS) plots are based on Bray-Curtis dissimilarities of square root transformed and double Wisconsin standardized OTU tables…"

Results:

L243: replace "were having" with "had"

We replace "were having" with "had" as suggested.

L244: why is Fig 2 c, d referenced before Fig 2 a, b.. did I miss the reference to a, b somewhere?

We made sure to mention Fig 2a,b before c,d. For graphical reasons we prefer, showing sea ice profiles on top of sea water, which allows better comparisons of the water-sea ice interface.

L265: Are there any photos of the subglacial outflow described in L267-268? Since there is a lack of field data at this time of year I think that these would be of value.

We do have a view photos that show the sampling location of the subglacial discharge water, but the picture is not very clear since the liquid water was sampled below a layer of ice (Icing). We added the photos showing different aspects of the outflow in the supplement with a description and arrows pointing to where the sample was taken. Fig S4c is from a video that clearly shows the liquid phase of the water on top of the Aufeis after breaking the ice layer and disturbance.

[Figure]

Figure S4. Sampling site for the subglacial discharge water. a) Aufeis on land in front of the southern part of the glacier and location of the ice cave shown in b-d (red arrow). b-d) Inside the ice cave with red arrow pointing to the liquid water sampled. The liquid meltwater was mostly covered by a layer of ice. Picture credits: a,c) Josef Elster, b) Marie Sabacka, d) Tobias Vonnahme.

L283: When reading about the very high nitrate+nitrite and silicate concentrations below the ice at SG I found myself really wondering if this could be coming from the subglacial meltwater itself versus upwelling of deeper marine waters. I believe you have data of the glacial meltwater itself? You mention these samples in lines 101-102 .. and I see further on that you present this data in L295. I'd suggest re-organizing so that this comes before the marine data.

We agree that the glacial meltwater data should be shown earlier to answer this question before it arises. We moved the sentences about the subglacial outflow to the start of the paragraph.

L295: missing units for silicate in the outflow water

Thanks for spotting this omission. We add the units of $\mu$mol L$^{-1}$

L300: The definition of conservative mixing is not quite right. The sentences in lines 300- 302 are especially problematic. I see that the other reviewer has already adequately commented on this so I will defer to those comments. In the rest of the paragraph I would avoid the words "positive mixing patterns" and "positive relations". I also found the color scheme in Fig 5 (red and pink) challenging to interpretation.

Concerning the color scheme in Fig 5, we used the same colors as in the rest of the manuscript for consistency. However, we agree that the colors appear too similar in Fig 5 and added a black outline to the red circles which will help improve clarity while keeping it consistent.

Concerning the conservative mixing we changed the text in the following way as described in the response to RW 1:

"Nutrient versus salinity profiles can give indications of the endmembers (sources) of the nutrients (Fig. 5) based on a linear correlation indicating conservative mixing. A positive correlation indicates higher concentrations of the nutrients in the saline Atlantic water endmember, while a negative correlation points to a higher concentration in the fresh glacial meltwater endmember. Biological uptake and remineralisation could weaken or eliminate the correlation, indicating non-conservative mixing. In the water column at NG and IE, silicate ($R^2=0.66$, $p=0.008$), NOX ($R^2=0.62$, $p=0.01$) and phosphate ($R^2=0.69$, $p=0.005$) showed conservative positive mixing patterns with higher contributions of Atlantic Water (Fig. 5a-c). At SG silicate was negatively correlated to salinity pointing to a higher concentration in glacial meltwater ($R^2=0.86$, $p<0.0001$). The absence of correlations for NOX and PO4 indicate non-conservative mixing pointing towards the relevance of biological uptake and release measurements (Fig. 5d-f)."

L310: I echo the other reviewer that these calculations of nutrients supplied via upwelling vs the glacial meltwater should be shown.. how were these calculated? What is the error on these calculations? This paragraph needs more explanation for these values to be believed especially considering (as pointed out by the other reviewer) the single outlier values that are driving the gradient in SG samples. Also, at SG, it seems, at least from Fig 5 d-f, that the lower salinity water had higher silicate concentrations but these concentrations were much higher than those reported for the glacial meltwater above. What is the source of this silicate?

Concerning the source of silicate, we prefer to keep this as part of the discussion. (Se ch. 4.4.3 first paragraph). Briefly, the mixing calculations show that the high Si values can be attributed to the subglacial discharge water itself AND bottom water reaching the surface. So, the bottom water appears an important source.

Concerning the calculations and error estimates, we provided following response to RW1 (the error estimates will be added to the text and table 1 (See above)) that explains our methodology and the inclusion of text as appendix:

We added the following calculations to the appendix. The mentioned outlier values in SG in the UIW sample were not used for the mixing calculations as explained before. For the meltwater fraction at the surface the error related to the average IE salinity is less than 0.1 % (see comment above), the main variation of the % meltwater contribution in the surface layer of SG is related to the salinity at the surface of SG (Fig. R1). We added the error estimate of 0.1 % to the table. For nutrients, the estimation error was estimated based on the variability in the concentrations measured in the triplicates from each water type. For NOx the estimated range of contribution by upwelling is thereby 57-59 % ($\pm$ 1 %) bottom water, for Silicate 89-95 % ($\pm$ 3 %), and for phosphate 46-49 % ($\pm$ 3 %).

Equations. Mixing calculations for estimates of the fraction of meltwater (MWSal) based on salinity, and for bottom water based on nutrient concentrations (BWNuts). Sal indicates the average salinities measured at the IE (SalIE), SG at 1m depth (SalSG1m), subglacial outflow (Salglac). Nut indicates the nutrient concentrations of nitrate and nitrite (NOX), silicate (Si), and phosphate (PO4) at 1m under the sea ice at SG (Nut1mSG) and IE (Nut1mIE), the bottom water of the IE (NutBW), or subglacial outflow water (Nutglac).

$$MW_{Sal}[\%] = \frac{Sal_{IE} - Sal_{SG1m}}{Sal_{SG1m} - Sal_{glac} + Sal_{IE} - Sal_{SG1m}} * 100$$

$$MW_{Sal}[\%] = \frac{34.7\ PSU - 23.6\ PSU}{23.6\ PSU - 0\ PSU + 34.7\ PSU - 23.6 PSU} * 100 = 32\ \%$$

$$BW_{Nut}[\%] = \frac{Nut_{1mSG} - MW_{Sal}[\%] * Nut_{glac} - Nut_{1m_{IE}} + MW_{Sal}[\%] * Nut_{1m_{IE}}}{Nut_{BW} - Nut_{1m_{IE}}} * 100$$

$$BW_{NOX}[\%] = \frac{6.52\mu M - 0.32 * 2.06\ \mu M - 3.27\ \mu M + 0.32 * 3.27\ \mu M}{9.57\ \mu M - 3.27\ \mu M} * 100 = 58\ \%$$

$$BW_{Si}[\%] = \frac{4.30\ \mu M - 0.32*1.79\ \mu M - 1.59\ \mu M + 0.32*1.59\ \mu M}{4.46\ \mu M - 1.59\ \mu M} * 100 = 92\ \%$$

$$BW_{PO4}[\%] = \frac{0.41\ \mu M - 0.32*0.09\ \mu M - 0.34\ \mu M + 0.32*0.34\ \mu M}{0.67\ \mu M - 0.34\ \mu M} * 100 = 46\ \%$$

[Figure]

Figure R1. Estimated fractions of glacial meltwater in the surface layer of SG.

L333: Like the other reviewer I'm confused by the term "vertical export of Chl" – what it means, how
it was estimated, and what the errors on this estimate are.

See response to RW1 (The error is based on Chl a tripiclates and given in Fig. 6):
The vertical export flux of Chl a is based on Chl a measurements in the sediment traps. We first convert
the measured Chl concentrations (mg m-3) to mass (mg) in order to calculate the flux as the mass of
Chlorophyll a per unit area and time sedimenting to a certain depth.
Change:
This leads to higher (14 times) vertical export flux based on the sediment trap measurements than
production at IE and considerably lower (5 %) export than production at SG (Table 2).

L337: "assuming absence of grazing".. this doesn't really seem realistic?

The assumption is necessary since we did not estimate grazing rates. If grazing would be considered the
loss rate would be higher. For clarity, we added following sentence.

"As grazing was not estimated in this study, the suggested loss terms of Chl based on the sediment trap data are likely underestimations."

L348: I'd suggest explaining more fully again the goal of the "reciprocal transplant experiment" before giving the results.

See changes in the methods:

"For testing the effect of the water chemistry on phytoplankton growth, we designed a reciprocal transplant primary production experiment where the phytoplankton communities at SG and IE (1 m and 15 m) each were transplanted into sterile filtered water of both SG and IE. 50 ml of the water containing the respective original phytoplankton community were transferred into 50 ml sterile filtered (0.2 µm) seawater of SG or IE each in 100 ml polyethylene bottles. The bottles were then incubated in situ at the original depth and primary production measured as described above. The aim of the experiment is to test if water chemistry alone is sufficient to increase primary production, or if differences in algal communities, light regimes, or temperatures are more important. These samples were incubated and processed together with the other PP incubations at the adequate depths as described above."

We also added a short introduction of the experiment to the results:

"The reciprocal transplant experiment aimed to show the effect of water chemistry on primary production in the absence of effects related to different communities, temperature, or light. The results (Fig. 7) showed clearly …"

Fig 6: The quality of this figure should be improved. The numbers in the parentheses are very difficult to read.

For the final version the quality will be substantially better due to the use of vector files (pdf) instead of png (as in the current pre-print file). We will also increase the font size of the error ranges in the parentheses)

Fig 7: The x-axis with the experiment name are not clear. What does "com" stand for?

We wrote now "community" instead of "com"

Fig 8: Define UIW in the legend as you have for the other abbreviations

We wrote it now out as "Under ice water" as suggested.

L355-356: "The first [NMDS1] axis separated sea ice from water communities with no overlapping samples".. this really isn't evident in Fig 8a.. sea ice is the square and what water and under ice water samples are the triangles. These regularly are in the same ellipses, unless I'm missing something? Also, is the glacier outflow sample actually a under ice water sample? What is the salinity of this sample? I guess I'm wondering if this is a true non-marine glacial outflow sample or one that could be diluted by marine water? I think this is an important point that needs to be clarified above.

We agree that the figure needs some clarifications.

1. We agree that sea ice and water samples are not directly separated by axis 1, but by axis 1 and 2 and remove the reference: "Sea ice and water communities are clearly separated with no overlapping samples."
2. The ellipses include subglacial meltwater (Salinity=0), glacier ice (Salinity=0), surface water and sea ice at SG in 2019 and 2018, and the remaining water and sea ice samples (including deeper water samples from SG). For clarity, we colored the ellipses. In the figure caption we added following clarification: "… Groups highlighted in eclipses: glacier ice (top right), undiluted subglacial outflow (top left), surface samples (UIW, sea ice) at station SG 2019 (top blue), surface samples (1m water, sea ice) at station SG 2018 (bottom blue) and others including deeper water samples at SG (bottom). The fraction of shared OTUs (in %) are shown as lines scaled to the fraction [%] of shared OTUs.

3. We also used now a separate symbol for glacial outflow to avoid confusion about the origin (under the sea ice, or from the subglacial outflow)

4. The aim of the eclipses is to support the discussion of OTU turnover between trhe subglacial outflow and marine samples, which we use for a rough estimate of fluxes and connectivity. Since we only do the analyses for 16S samples (due to short generation time and availability of complete glacier samples), we did not show ellises for the eukaryotic communities.

L358-360: What was the stress on this NMDS? How robust is this ordination you show? I'm always weary of interpreting the axes in this manner, i.e. axes one shows X and axes 2 shows Y .. i.e. similar to how one might view a PCA. I agree that looking at Fig 8a your communities are different but I don't think you can go as far to say that axis 1 is separating ice vs water and axis 2 is separating glacial vs marine. The ordination of this NMDS would likely change each time you ran it.. maybe something to consider?

The stress values are given on top of the NMDS plots (0.07 for 16S, 0.14 for 18S and LM). The stress values are indicative of a very good to good representation in the reduced dimensions. For clarity, we added the information also in the figure caption. We removed the description of which axis separates the community. With the R function used (metaMDS) the ordinations stay the same (The plot is reproducible with the same code).

L371: "Overall the same NMDS clustering has been found as for the 16S rRNA sequencing" .. but in the 18S plot (Fig 8b) no ellipses are drawn.. does this indicate that these group divisions were not significant? The written text doesn't seem to match the figure.

The aim of the eclipses is to support the discussion of 16S OTU turnover between the subglacial outflow and marine samples, which we use to estimate fluxes and connectivity. Since we only do this analyses for 16S samples (due to short generation time and availability of complete glacier samples), we did not show ellipses for the eukaryotic communities. However, for comparability and due to descriptions of clusters in the written text, we added the ellipses for Fig. 8b and c. We tested for significance using ANOSIM and describe the significant (p<0.005) differences in the text.

Fig8c – the separation in the samples is quite striking on this NMDS. How come there are no ellipses on this plot? Were the differences shown in the NMDS not significant? Could try a perMANOVA to test the significance of differences between the groups perhaps?

For Fig 8c we prefer added the same ellipses. However, since the sampling design differs not all ellipses are present. As described in the text, differences between sea water and sea ice are significant (ANOSIM, p<0.005), but not the differences between SG surface samples, and other stations. For Fig 8c we also did following changes in the text for clarity: "Furthermore sea ice species composition at SG station differed from NG and IE (Fig. 8c)."

Discussion:

L388-391: These first few lines are a great summary and really the abstract and introduction needs to be better set-up to frame these important points: (1) evidence for subglacial upwelling at a shallow tidewater glacier under sea ice and (2) that this upwelling persists in the winter / spring and supplies nutrient-rich glacial meltwater and upwelling of bottom water: : : I actually think part of the confusion is the use of the term "upwelling" to describe the release of submarine discharge into the ocean and also the upwelling of bottom water. Perhaps a change of language throughout would be helpful – i.e. saying "submarine discharge" vs "subglacial upwelling". And as per points above the case about nutrient-rich glacial meltwater needs to be set-up and made earlier as it's really a central finding.

The referee has a good point that subglacial upwelling and submarine discharge are two different processes. We changed the terminology of submarine upwelling to submarine discharge where necessary ( e.g: "(1) evidence for submarine discharge at a shallow tidewater glacier under sea ice and (2) that this submarine discharge persists in the winter") throughout the manuscript. As mentioned above, we also moved the results description of nutrients in subglacial meltwater to the beginning of the nutrient section and added an introduction about the effect of water storage underneath a glacier over winter on the water chemistry (silicate enrichment by prolonged contact with the bedrock -> weathering, ion concentration by solute expulsion during freezing of stored meltwater)

L406: The phrasing "which does not allow basal glacial ice to melt" is unclear. The whole sentence is too long and should be made into 2, but are the authors saying that because there is not Atlantic inflow water there can be no basal ice melt? Basal ice melt can result from geothermal heat flux, overburden ice pressure, and sliding friction. Warm ocean water is not the only mechanism. I suggest looking at a textbook (e.g. the physics of glaciers) and reviews on this topic: e.g. Hubbard and Sharp, 1989

We realize that we used the wrong terminology here. We are discussing glacier terminus (glacier-marine interface) ice melt, and not basal (glacier-bedrock interface) ice melt. We corrected the terminology throughout the discussion. We also agree that the sentence can be splitted in 2.

L407: "Subglacial meltwater itself is unlikely to lead to basal ice melting due to its low salinity". This sentence is very unclear to me. I'm not sure what this sentence is saying or trying to say.

We agree that this sentence is very unclear and removed it.

L407-408: "However, basal ice melt is likely more important in systems with Atlantic water inflows: : :" as per above this seems to ignore the possibility of basal ice melt underneath temperate and polythermal glaciers. This may not be what the authors mean but as written it reads this way.

As mentioned above, we meant glacier terminus ice melt and not basal ice melt and correct the terminology.

L420: "remains from the previous melting season" is unclear. Can you specify what you mean by remains.

We refer to fresh meltwater that entered the fjord during the previous melting season (summer), remaining at the surface (due to its lower density) throughout winter due to limited mixing and advection. We added following clarification: "may be meltwater introduced during the last summer to fall melting season and remaining throughout winter."

L433: Can you specify what data you are referring to when you say "estimated bacterial growth rates". I searched for this term in the paper and did not see it previously defined. It really should be so that the basis for this calculation of doubling time is clear.

The estimated bacterial growth rate is given in table 2 as bacteria biomass production. We replaced the term "growth rate" with "biomass production" for consistency and to add a reference to table 2 in the text.

L442: Why does the supply have to be "constant" ? It seems like (from the methods) that samples for community analyses were only taken once at each station? How does a single-time point sample give an indication of the timescale of submarine discharge into the fjord? This might be a bit of a reach based on the community data alone – suggest tempering this statement.

We agree that "constant" appears to be the wrong term. We used the term "continuous" instead. The argument is that we assume that the Bacteria that are only present in subglacial outflow and surface SG water are inactive and not growing. Considering the doubling time of the entire bacteria community, these inactive not-growing bacteria would be replaced by active bacteria in the time frame of the doubling time. In addition to overgrowth, inactive bacteria would also be exposed to losses due to grazing, viral lysis, and sedimentation. We acknowledge that these assumptions are very simplified and also added some terms to show the uncertainty of this estimate: "Thus, we suggest that the presence of shared OTUs between SG and the glacial outflow may indicate a continuous supply of fresh inoculum to sustain these taxa."

L442-444: When you say the "southern part of the glacier" is this part on land or in the ocean? If it's on land you should specify. I also think that this assumption that this outflow is being released under the marine-terminating portion can be backed up by your marine data? This sentence seems out of place here.

Yes, we refer to the land-terminating part. We added the detail in the following way "land- terminating part south of the glacier".

We also agree that we have marine data to support this hypothesis (e.g. Salinity profiles). The observed subglacial outflow on land is simply an additional piece of evidence. We replaced "the clearest evidence" with "clear evidence" For clarification, we moved the observation of active subglacial outflow in the chapter before:

"Clear evidence for outflow comes also from the visual observations of subglacial outflow exiting the land-terminating part south of the glacier in October 2019, April 2018 and April 2019, which we assume also occurred under the marine terminating front. In fact, subglacial outflows in spring have been observed…"

L445- to end of paragraph: This explanation of glacier hydrology really needs to come earlier. As written this whole section on the potential magnitude of upwelling is poorly organized. Suggest first setting it up by talking about processes on the ice and then what's happening in the ocean.

We addressed this comment by 1) introducing the glacier hydrology more extensively in the introduction and 2) moving the section about glacier hydrology (442-451) to the end of chapter 4.1 since it is part of the evidence for submarine discharge and not directly for the magnitude/ flux.

L456: "Our mixing calculations estimate".. where are these calculations described?

See comment above. We added the equations and calculations to the appendix.

L457: At what depth is the submarine discharge exiting the glacier? I find myself wondering at what depth these different water masses occur (can you specify this) and how deep the DLAW is being entrained from? Is it sufficiently below the nutricline to be replete in nutrients? Also the calculated entrainment factor of 1.6, how was this calculated exactly? And you state "which pulled 1.6 times more DLAW" .. more than what? This is not clear.

Considering the estimated depth at the glacier terminus of 20 m, this would be the depth of the discharge exiting the glacier. Nutrients are depleted at the surface, but not at 15m, indicating that the discharge happens below the nutricline and has therefore the potential for upwelling.

We added this information in the following way: "Nutrients were depleted in the UIW, but not at 15 m depth, showing that the nutricline had to be shallower than 15 m. Hence, submarine discharge at a glacier terminus depth of 20 m would cause upwelling ofc nutrient rich DLAW to the surface."

The entrainment factor is the proportion of contributions from DLAW to SGO at the surface (53% DLAW: 32% SGO = 1.6 DLAW:SGO at 1m depth). We replaced "more" with "as much" for clarification. We also specified the calculation by replacing the "(53%)" by "(53 % DLAW : 32 % SGO = ratio of 1.6)" in the manuscript.

L458-459: "Fransson et al. (2020) found that 30-60% of glacier derived meltwater was incorporated in the bottom sea ice : : : again indicating that it is a widespread process at marine terminating glacier fronts" .. what is a widespread process? The release of submarine discharge and its incorporation into bottom sea ice OR the entrainment of different water masses (i.e. DLAW) as the plume rises (as discussed in the previous sentence). Again, this is a case in point of the organizational structure and lack of specificity of terms "submarine discharge" vs "upwelling of bottom waters" to be a source of confusion.

We added following clarification "… indicating that winter/ spring submarine discharge and the
resulting formation of sea ice with low porosity is a widespread process…".

L461: "Compared to the massive subglacial plumes of summer systems" .. where? This should be
specified .. different glaciers have widely different discharge fluxes. The citation seems to be from
Greenland but these glaciers will bear little resemblance to Svalbard, perhaps citing summer discharge
fluxes from Svalbard glaciers too would be useful – particularly from your study site if the intent of this
sentence is to contrast with spring discharge fluxes as seems to be the case.

We agree that the structure of the entire chapter needed improvement. Thus, we rewrote the entire chapter,
considering all comments. Concerning this specific comment, we specified the location and time of each
tidewater glacier system compared. We start with stating the conditions in our study, continue with the
most similar glacier on Svalbard, and finish with a wider picture by comparing the data to the larger and
deeper Greenland glaciers.

Changed chapter:

"To our knowledge, our study provides currently the only available estimate of subglacial upwelling in
early spring. Our study suggests that subglacial upwelling in spring causes in Billerfjorden a small volume
transport of only about >1.1 m3 m-2 month-1 (approx. 2 m3 s−1). This estimate is based on the flux of
nutrient rich bottom water needed to maintain the measured primary production assuming steady state
conditions and is therefore a rough, but conservative estimate. The most comparable estimate on the
magnitude of the upwelling is available at Kronebreen for summer. This Svalbard tidewater glacier is of
similar size and had one to two orders of magnitude higher upwelling rates compared to our study (31-
127 m3 s-1, Halbach et al., 2019). Due to their size, summer subglacial upwelling in Greenland is two to
four times higher than at Kronebreen (250-500 m3 s-1, Carroll et al., 2016). In our study about 1.6 times
as much bottom water from about 20 m (DLAW) as subglacial outflow water (SOW) reached the surface
at SG (Entrainment factor of 1.6 – see above). The entrainment factor is mostly dependent on the depth
of the glacier front (Carroll et al., 2016). In fact, the glacier terminus at SG was shallower (approx. 20 m)
than any other studied tidewater glacier on Svalbard (70 m depth at Kronebreen, Halbach et al., 2019) or
Greenland (> 100 m, Hopwood et al., 2020), explaining the higher summer entrainment factors estimated
in Kongsfjorden (3, Halbach et al., 2019) and Greenland (6 to 10, Hopwood et al., 2020) are not surprising.
Glacier terminus depth appears to be the main control of entrainment rates, likely independent of the time
of the year. However, turbulent mixing may cause increased entrainment during times of very high
subglacial discharge rates. Kronebreen is the most comparable tidewater glacier to our study area in terms
of glacier terminus depth and entrainment rate. Although the estimated entrainment factor was low at
Kronebreen (3), it substantially increased summer primary production in Kongsfjorden (Halbach et al.,
2019). Despite of the shallow depth, and the low discharge and entrainment rate of our study, subglacial
upwelling was the main mechanism to replenish bottom water with high nutrient concentrations to the
surface and substantially increased spring primary production due to; (i) submarine outflow below
(approx. 20 m) the nutricline (<15 m), (ii) the absence of any other terrestrials inputs, (iii) Atlantic water
blocked by a shallow sill (Skogseth et al., 2020), (iv) very weak tidal currents (Kowalik et al., 2015), (iv)
wind mixing blocked by sea ice in Billefjorden, and (v) undiluted subglacial meltwater having lower
nutrient concentrations than the DLAW."

The sentence mentioned by the RW was rewritten in the following way: "Our study suggests that subglacial upwelling in spring causes in Billerfjorden a small volume transport of only about >1.1 m3 m-2 month-1 (approx. 2 m3 s−1). This estimate is based on the flux of nutrient rich bottom water needed to maintain the measured primary production assuming steady state conditions and is therefore a rough, but conservative estimate. The most comparable estimate on the magnitude of the upwelling is available at Kronebreen for summer. This Svalbard tidewater glacier is of similar size and had one to two orders of magnitude higher upwelling rates compared to our study (31-127 m3 s-1, Halbach et al., 2019). Due to their size, summer subglacial upwelling in Greenland is two to four times higher than at Kronebreen (250-500 m3 s-1, Carroll et al., 2016)."

L462: "subglacial upwelling in spring is a small volume transport".. where is this data from? This study? This should be explicitly stated. Suggest re-writing this entire sentence. Also, the last part of the sentence regarding upwelling needed to maintain primary production should be a new sentence as this is a different point then the discharge flux.

The data are from this study. We agree that this should be stated. We also agree that the information "needed to maintain primary production should be moved to a seperate sentence. We rewrote the entire chapter, considering all comments. As suggested by RW1 we also converted the discharge units of the three studies (Greenland, Kongsfjorden, our study) to the same units for comparability. Concerning this comment, following changes were made:

""Our study suggests that subglacial upwelling in spring causes in Billerfjorden a small volume transport of only about >1.1 m3 m-2 month-1 (approx. 2 m3 s−1). This estimate is based on the flux of nutrient rich bottom water needed to maintain the measured primary production assuming steady state conditions and is therefore a rough, but conservative estimate."

L464: "This careful estimate".. I'd remove the word "careful".. the more so because the sentence before this one is unclear! Is this estimate of freshwater input for Billefjorden in the summer or spring? It's unclear. The estimate from the Halbach paper is I believe from the summer so you want to make sure you are comparing like with like.

As pointed out by RW1, "careful estimate" is a misleading formulation. We replaced it with "rough, but conservative". We also realized that the reason for comparing our spring study with summer values is not clear and specified that we do not know of any other spring studies with similar estimates. The study in Kongsfjorden is the most comparable estimate to our study (glacier size, terminus depth, location). We did following changes: "To our knowledge, our study provides currently the only available estimate of subglacial upwelling in early spring. …. The most comparable estimate on the magnitude of the upwelling is available at Kronebreen for summer. This Svalbard tidewater glacier is of similar size and had one to two orders of magnitude higher upwelling rates compared to our study (31-127 $m^3$ $s^{-1}$, Halbach et al., 2019)."

L465-466: The fact that you have less entrainment than the Hopwood study is really not surprising at all considering the depth of discharge and flux of discharge at the much deeper, larger glaciers in that study. I'm not sure what the purpose is of this statement? As written now it's failing to provide relevance to this study.

We agree that this fact is not surprising and rephrased the statement. We still argue that it is necessary to compare entrainment rates and state that the glacier terminus depth is typically the controlling factor, apparently independent of the time of the year.

"In our study about 1.6 times as much bottom water (DLAW) as subglacial outflow water (SOW) reached the surface at SG (Entrainment factor of 1.6 – see above) through the upwelling process. The entrainment factor is mostly dependent on the depth of the glacier front (Carroll et al., 2016). The glacier terminus at SG was shallower (approx. 20 m) than any other studied tidewater glacier on Svalbard (70 m depth at Kronebreen, Halbach et al., 2019) or Greenland (> 100m, Hopwood et al., 2020). Hence, the higher summer entrainment factors estimated in Kongsfjorden (3, Halbach et al., 2019) and Greenland (6 to 10, Hopwood et al., 2020) are not surprising. Overall, glacier terminus depth appears to be the main control of entrainment rates, likely independent of the time of the year. However, turbulent mixing may cause increased entrainment during times of very high subglacial discharge rates."

L466-467: "each volume of SGO water pulled about the same volume of DLAW with it to surface".. this is unclear.. do you mean each volume over a certain timeframe (a day? A week? A month?) .. what is the volume exactly? What was the volume of DLAW entrained? This should be stated if you are speaking about volumes here. And again the comparisons to the Hopwood study don't' seem relevant if you are comparing to large Greenland glaciers. You should specify where and what type of glaciers in the Hopwood review you are comparing too.

We refer to proportion of volumes (Vol DLAW : Vol SOW), which is a value comparable to chemical volume percentages (e.g. 70% Ethanol in MQ vol/vol). Thereby an exact volume is meaningless. To avoid confusion, we rephrased the sentence in the following way.

"In our study about 1.6 times as much bottom water (DLAW) as subglacial outflow water (SOW) reached the surface at SG (Entrainment factor of 1.6 – see above)"

We also specified the type (depth, size, location) and time (summer) of the compared studies as mentioned above.

To our knowledge, our study provides currently the only available estimate of subglacial upwelling in early spring. ….The entrainment factor is mostly dependent on the depth of the glacier front (Carroll et al., 2016). The glacier terminus at SG was shallower (approx. 20 m) than any other studied tidewater glacier on Svalbard (70 m depth at Kronebreen, Halbach et al., 2019) or Greenland (> 100m, Hopwood et al., 2020). Hence, the higher summer entrainment factors estimated in Kongsfjorden (3, Halbach et al., 2019) and Greenland (6 to 10, Hopwood et al., 2020) are not surprising. Glacier terminus depth appears to be the main control of entrainment rates, likely independent of the time of the year."

L470: This is the first mention of the depth of the discharge. As you say, 20-m is quite shallow. Are nutrient concentrations sufficiently high enough here to augment surface concentrations? In other words, is this depth below the nutricline.

As mentioned above, we now mention the depth earlier in the chapter. We also provide information on the depth of discharge in relation to nutricline (see comments above).

"The entrainment factor is mostly dependent on the depth of the glacier front (Carroll et al., 2016). The glacier terminus at SG was shallower (approx. 20 m) than any other studied tidewater glacier on Svalbard (70 m depth at Kronebreen, Halbach et al., 2019) or Greenland (> 100m, Hopwood et al., 2020)."

We also mentioned that the submarine discharge enters the fjord below the nutricline in the end of the chapter.

"In spite of the shallow depth, and the low discharge and entrainment rate of our study, subglacial upwelling appears to be the main mechanism to replenish bottom water with high nutrient concentrations to the surface and can substantially increase spring primary production due to; (i) submarine outflow below (approx. 20 m) the nutricline (<15 m), (ii) the absence of any other terrestrials inputs, (iii) Atlantic water blocked by a shallow sill (Skogseth et al., 2020), (iv) very weak tidal currents (Kowalik et al., 2015), and (iv) wind mixing blocked by sea ice in Billefjorden, and (v) undiluted subglacial meltwater having lower nutrient concentrations than the DLAW."

L473-to end of paragraph: This seems to directly contradict previous statements regarding the glacial meltwater discharge being enriched in nutrients (e.g. silicate?). Also many of the comparisons you are making are to summer discharge fluxes and summer entrainments.. the spring discharge will of course be lower but more chemically enriched from the glacial meltwater discharge? I think if you are going to use the summer values to compare, which you might have to do out of necessity and lack of other comparisons, you need to state so explicitly, and the limitations of such comparisons.

The glacial meltwater is enriched in silicate, considering its salinity (0) and compared to UIW and sea ice at NG and IE, but not compared to the bottom water. We tried to clarify it by following statement:

"…(v) undiluted subglacial meltwater having lower nutrient concentrations than the DLAW"

As mentioned above, we fully agree with the confusions about the comparisons. We rewrote the entire chapter in the following way:

[revised manuscript text omitted]

L480: The word "Surprisingly" seems to not be the right word choice here.

We removed the word "Surprisingly".

L438: "Substantial subglacial upwelling" .. I'm unclear was to what you are referring to here – is this
submarine discharge of glacial meltwater or upwelling of bottom waters? In either case the word
"substantial" seems ill-advised here given the preceding discussion and should be removed. Could it be
that you didn't observe much light limitation because the plumes were not that "massive" (compared to
summer).. i.e. you just have a much smaller discharge flux and therefore plume in the spring? This seems
likely and unsurprising.

We agree that the formulation is misleading and removed it.

L485-86: Unclear what the phrase "where light is not considered limiting" is referring too.

We specified in the following way: "where light sufficient for photosynthesis". Line 511:

"rations" should be "ratios"?

We replaced the term "rations" with "ratios".

L515: Can you really call it "deep water upwelling" if the water is being entrained from only 20-m?
This is problematic (at least for me) and needs to be clearly addressed I think.

We replaced the term "deep water" with "bottom water".

L517-519: The discussion on iron seems unrelated and as written is unconvincing.

We consider a short discussion of iron important for a comprehensive discussion. Without the
information the reader may consider iron as important micronutrient not considered and potentially
important, which would weaken the robustness of the study. By acknowledging that iron may be
imported in large amounts, but is not limiting in coastal Arctic systems, we clarify this potential question
briefly. We added following clarification and an additional reference: "However, iron limitation typically
does not occur coastal Arctic systems (Krisch et al., 2020)."

L520: "nutrient concentrations may simply be higher due to the shallower depth at SG" .. why? It's
unclear what you are trying to say. Suggest re-writing with more detail and explicity.

Nutrients are typically higher close to the sea floor due to benthic regeneration of organic matter in the
sediments. If the surface water is only 30m over the bottom, vertical mixing via diffusion or advection
needs consequently less time and/or physical forcing than at 150 m depth. We added following clarification: "nutrient concentrations may be higher due to less physical forcing and time needed for vertical mixing at the shallower SG compared to IE.

L529: Was the Frasson study done at this same site?

No the study was done at the neighboring fjord. We added the information in the following way: "The role of bedrock derived minerals and particles for composition of sea ice chemistry have been described in detail in the neighboring fjord (Tempelfjorden) by Fransson et al. (2020)."

L530: "The values" .. vague.. specify what kind of values you are referring to.

We replaced "The values", with "Silicate concentrations"

L535: Paragraph ending here is rambling and needs to be re-written. Suggest taking out the iron since you have no data on this to compare.

We agree and removed the last sentence about iron.

L536: "related".. what do you mean by this word? Specify.

We added following clarification: "…which was introduced via subglacial upwelling in Kongsfjorden…"

L538: Were are you proposing this nitrification is occurring? In the ocean or in the glacial meltwater? Could the high nitrate come from the subglacial waters itself? See papers by Beaton et al. in Greenland, Jemma Wadham, Boyd et al., 2011 (AEM) and Wynn et al., 2007 (Chemical Geology). Do you have measurements of the outflow un-diluted by seawater so you can rule this possibility out?

We propose the nitrification to happen in the UIW. We added the following information: "Ammonium regeneration and subsequent nitrification under the sea ice…". We disregard high nitrate inputs from the glacial meltwater itself since we did not measure high nitrate concentration in our samples from the outflow of undiluted meltwater (see Table 1). For clarification we added the following statement: "Nitrate can be supplied through the subglacial meltwater itself (Wynn et al., 2008), however we did not find high nitrate concentrations in the undiluted subglacial outflow water in our study."

L566: Were you able to resolve any low-light level species in your molecular community composition data to back this statement up?

In general, diatoms are know to be quite well adapted to low light levels. Diatoms were also the most common taxon of the UIW phytoplankton community (based on light micsroscopy, which is more quantitative). We added a statement of the capability of diatoms to grow under low light conditions. "In particular diatoms, the most common taxa of under ice phytoplankton blooms (von Quillfeldt, 2000, this study) are known to be well adapted to low light conditions (Furnas, 1990)."

Furnas MJ (1990) In situ growth rates of marine phytoplankton: approaches to measurement, community and species growth rate. J Plankton Res 12:1117–1151

L581: "their" .. unclear what this is referring to.

We replaced "their production" with "primary production"

L646: "In winter and spring, this would result in the lack of subglacial upwelling".. but with more melt there would be longer melt seasons and presumably more submarine discharge and associated upwelling
– at least in the shorter term?

We added following information: "In the shorter term, a longer melt season and presumably increased submarine discharge may lead to increased subglacial upwelling in winter and spring. However, on longer time scales , tidewater glaciers will retreat and transform towards land terminating glaciers (Błaszczyk et al., 2009), which would result in the lack of subglacial upwelling and systems more similar to the IE with less nutrients and light available for phytoplankton."

---

## Referee Report (RR1)

The authors have thoughtfully and thoroughly responded to comments on the previous draft and that effort is appreciated. This is a valuable contribution regarding early spring discharge at a tidewater glacier front, which convincingly combines a number of lines of evidence to support their conclusions of higher primary productivity under the ice in such fjords. The purpose and conclusions of the manuscript are now much clearer and, as such, the manuscript is overally greatly improved. I believe that the manuscript is now much improved and close to being ready for submission.

However, I do have some specific suggestions below to once again, aid clarity, for the authors' consideration.

L14: the meaning of the word "sufficient" is unclear. Do you mean the flux is sufficient or that the associated nutrient upwelling is sufficient? It would be good to be precise here.

L18: change to "we still observed... and primary production at this time of year"

L18-19: "subglacial meltwater" .. suggest changing / specifying "submarine discharge"?

L20: the reason for the two-fold higher under-ice irradiance is not clear from the preceding sentences.

L22: "The nutrient supply increased primary production" .. how are you disentangling nutrient supply from the effects of stratification and irradiance?

L22: increased primary production by 30% where? In the seawater? In the sea ice? Both?

L23: the meaning of the phrase "limiting the inhabitable place" is unclear

L22-23: You switch here to talking about the sea ice communities I believe, and in then in the next sentences are back (?) to talking about the under ice marine communities.. it might be good to keep the discussion about the water together and then talk about the sea ice.

L26-28: This last sentence (arguably an important one since it's the end of the abstract) is unclear. Suggest re-writing. Perhaps split into – making the part after "while sea ice" a new sentence. There are just too many qualifiers re: the sea ice to take in to come away with the overall take-home message as currently written.

Also, for the first part of the sentence regarding the retreat of tidewater glaciers, I think the authors should instead emphasize the possibility of longer, more extensive melt seasons with climate change and thus the higher likelihood for more early season spring discharge and what that effect is on primary production. The study of this spring early-season discharge is really the novel aspect of this paper – so there's no need (in my opinion) to focus on the eventual retreat of tidewater glaciers – which is further into the future, and involves more complicated considerations of bed slopes.

L38: change "upwelling" to "upwelled"

L39 & L43: while I appreciate the authors adding in approximate distances, it seems strange that the region where primary productivity is low in front of the glaciers is in the same range as the region at some distance out where it is higher – particularly since the authors cite the same study. In L39 suggest refining the estimate for low primary productivity in close proximity to the glacier front according to the figure referenced in Halbach et al., 2019.

L41: add "type" after glacial bedrock

L42: change "surface increase" to "surface can increase"

L48: change to "an overall low discharge flux"

L49: change to "However, the limited amount"

L50: change to "quantification of **spring** subglacial outflow" and "on **both** sea ice.."

L57: change to "Glacier terminus melt rates of basal ice at the glacier-marine initerface.."

L58: change to "seasonal subglacial outflow flux..."

L59: suggest change "terminus" to "**basal**" and "than in" to "**compared to the**" Also, was their evidence for winter upwelling in the Moon et al. (2018) study? Was the flux high enough to permit this? It's unclear as written but if this was the case I would suggest explicitly stating so.

L60: change depth to "depths", change "deep terminus melt" to "basal ice melt"

L63: I wonder why the authors are specifically mentioning cold-based glaciers here – especially as because later they emphasize that their results are extendable to warm or polythermal ice masses. Suggest deleting the specific mention of cold-based here and instead emphasize that this has been observed at several warm and polythermal ice masses in Svalbard.

L72: change "subglacial" to "submarine"?

L75: change "to the summer situation" to "in the summer".

L80: change "nutrient ion enrichment" to "nutrient and ion enrichment"?

L82-84: Sentence beginning with "Especially" is very unclear and seems a tad redundant (at least as written).

L88-90: Would suggest deleting this sentence and focusing instead of impact of warmer and longer meltseasons – no need to extend to glacier retreat which is really beyond the scope of the study – see comments above for abstract.

L93: delete "ice algae start growing" and also L94: delete" within sea ice"

L97: change "subglacial upwelling" to "subglacially induced upwelling)

L101: extra space between "to" and "wind"

L111-112: Re-write the last sentence: We suggest that even though subglacial upwelling is diminished in the spring, compared to the summer, in the absence of wind mixing, the enriched nutrient may enhance algal growth at this time of year.

L114: The way this sentence is written begs the question – and what about the other glacier front.. why not describe that too?

L146: I have to admit to still being confused as to why additional water was added to the sea ice cores for melting. Why not just directly melt the core?

L247: Do you mean you transferred the phytoplankton communities into their respective environments? Sentence is a bit unclear.

L251-252: Last sentence is unclear. Is "adequate" the correct word here?

L321: Change therefor to therefore

L325: extra space after 4 µmol/L

L345: It's unclear in this paragraph what site you are referring too? Is this for all the sites>

L349-350: Is this estimate for inorganic nutrients valid in light of the non-conservative mixing discussed in the paragraph prior?

L437: Add phrase "there was no" after "In addition"

L444: change to "below the freezing point"

L452: change to glacier meltwater contributions

L462: change to have **enriched** silicate concentrations

L464: Note emphasis here on warm or polythermal-based glaciers – see comment above regarding line in introduction on cold-based glaciers

L467: change "last" to late

L468: change "melting" to **melt season which as remained** throughout the winter. (delete extra period)

L473: change "substantially" to much

L475: change "productions" to **production**

L505: move phrase "in Billerfjorden" to the end of the sentence

L515: make second part of the sentence beginning with "explaining the higher summer …" a new sentence or re-write to make it follow from the first part of the sentence. As written, it doesn't.

L519: is "it" referring to submarine upwelling? Suggest clarifying.

L547: With regards to stratification observed at the SG site, would the physical mechanism of upwelling disturb this stratification?

L604: Typo in "studiy"

L607: Change "studies" to study periods

L682: Change "Last" to Lastly

L698: Change "most of" to much and "different evidence" to different lines of evidence

L704: systems more similar to NG or IE? NG seems appropriate to refer to here since this is actually an adjacent land-terminating site correct?

L705: change to "but would result in higher biomass.."

L706: Sentence beginning with "The pelagic ..." is unclear and a fragment. Suggest re-writing.

L720: What is the depth of the effect of wind-induced vertical mixing?

---

## Author Response (AR2)

The authors have thoughtfully and thoroughly responded to comments on the previous draft and that effort is appreciated. This is a valuable contribution regarding early spring discharge at a tidewater glacier front, which convincingly combines a number of lines of evidence to support their conclusions of higher primary productivity under the ice in such fjords. The purpose and conclusions of the manuscript are now much clearer and, as such, the manuscript is overally greatly improved. I believe that the manuscript is now much improved and close to being ready for submission. However, I do have some specific suggestions below to once again, aid clarity, for the authors'consideration.

We want to thank the reviewer sincerely for the positive evaluation and very constructive and detailed last feedback which helped to improve the manuscript. We corrected the manuscript as outlined below or provide detailed explanations to more general questions of the reviewer. We also corrected some additional minor grammatical issues.

We also realized that Fig. 5, submitted with the last version was still missing the asterisks indicating significant linear regression and had values with too many digits behind the comma. We also realized that we plotted Nutrients against brine salinities, while a plot of Nutrients against bulk salinities are more meaningful (while showing the same trends). Thus, we now uploaded the corrected figure. This change has no implications on any conclusion drawn in the paper, but we believe it provides better illustration of the salinity versus nutrient relationship to the reader. In fig 1, we realized that we plotted CTD profiles starting from the water surface while it is much more meaningful to show data starting at the sea ice-water interface and we adjusted the plot accordingly.

L14: the meaning of the word "sufficient" is unclear. Do you mean the flux is sufficient or that the associated nutrient upwelling is sufficient? It would be good to be precise here.

We specified that the flux is sufficient in the following way: "We hypothesized that submarine discharge under sea ice is present in early spring and that its flux is sufficient to increase phytoplankton primary productivity."

L18: change to "we still observed... and primary production at this time of year"

We changed the sentence accordingly.

L18-19: "subglacial meltwater" .. suggest changing / specifying "submarine discharge"?

We changed "subglacial meltwater" to "subglacial discharge" as suggested.

L20: the reason for the two-fold higher under-ice irradiance is not clear from the preceding sentences.

We added following detail: "…a two-fold higher under-ice irradiance due to a thinner snow cover…"

L22: "The nutrient supply increased primary production" .. how are you disentangling nutrient supply from the effects of stratification and irradiance?

We disentangled the effects of nutrients from stratification and irradiance based on the results of the reciprocal transplant experiment, where the addition of sterile filtered surface water from SG lead to higher primary production in both IE and SG systems, while filtered seawater from the IE had the opposite effect. We added following details: "Reciprocal transplant experiments showed that nutrient supply increased primary production by approximately 30 %."

L22: increased primary production by 30% where? In the seawater? In the sea ice? Both?

In seawater. We added the information: "…increased phytoplankton primary production by approximately 30 %."

L23: the meaning of the phrase "limiting the inhabitable place" is unclear

With a lower brine volume fraction the brine channels become smaller and fewer leading to place limitation. We added following specification: "inhabitable brine channel space"

L22-23: You switch here to talking about the sea ice communities I believe, and in then in the next sentences are back (?) to talking about the under ice marine communities.. it might be good to keep the discussion about the water together and then talk about the sea ice.

We talk about seawater in the beginning before adding one sentence about sea ice and finishing with a sentence about both sea ice and seawater combined. We tried now to clarify it by either using the term "sea ice" or "seawater", or "phytoplankton" to make sure about what we are talking about in the different sentences.

L26-28: This last sentence (arguably an important one since it's the end of the abstract) is unclear. Suggest re-writing. Perhaps split into –making the part after "while sea ice" a new sentence. There are just too many qualifiers re: the sea iceto take in to come away with the overall take-home messageas currently written. Also, for the first part of the sentence regarding the retreat of tidewater glaciers, I think the authors should instead emphasize the possibility of longer, more extensive melt seasons with climate change and thus the higher likelihood for more early season spring discharge and what that effect is on primary production. The study of this spring early-season discharge is really the novel aspect of this paper –so there's no need (in my opinion) to focus on the eventual retreat of tidewater glaciers –which is further into the future, and involves more complicated considerations of bed slopes.

We split the sentence as suggested. Regarding the first sentence, we prefer to keep the take home message. One main message of our paper is also that the submarine discharge is somewhat decoupled from snowmelt, which makes the argument of earlier and increased discharge with climate change a bit out of place in the context of our paper. Furthermore, it is not clear if the increased discharge can compensate the negative effects of a decreasing grounding line depth and a consequently decreasing plume dilution factor. Thus, we argue that tidewater glacier retreat to land is most important in the context of our study.

L38: change "upwelling" to "upwelled"

We changed the term accordingly.

L39 & L43: while I appreciate the authors adding in approximate distances, it seems strange that the region where primary productivity is low in front of the glaciers is in the same range as the region at some distance out where it is higher –particularly since the authors cite the same study.

The ranges specify a light inhibited area close to the glacier in contrast to an upwelling fertilized area further away. We tried to specify this in the following way: "Primary production is typically low in direct proximity to the glacier front (within hundreds of meters to kilometres from the glacier front …increase summer primary production at some distance ("more than hundreds of meters to kilometres away from the glacier front"

In L39 suggest refining the estimate for low primary productivity in close proximity to the glacier front according to the figure referenced in Halbach et al., 2019.

We added following details: "Primary production and biomass is typically low (e.g. 0.6 ± 0.3 mg Chl a m$^{-3}$, Halbach et al., 2019) in direct proximity to the glacier front"

L41: add "type" after glacial bedrock

We added "type" as suggested.

L42: change "surface increase" to "surface canincrease"

We changed the statement accordingly.

L48: change to "anoverall lowdischarge flux"

We changed the sentence accordingly.

L49: change to "However, the limited amount"

We changed the sentence accordingly.

L50: change to "quantification of spring subglacial outflow" and "on bothsea ice.."

We changed the sentence accordingly.

L57: change to "Glacier terminus melt rates of basal ice at the glacier-marine initerface.."

We changed the sentence accordingly.

L58: change to "seasonal subglacial outflow flux..."

We changed the sentence accordingly, but did not include the word seasonal, since it seems a bit misplaced.

L59: suggest change "terminus" to "basal" and "than in" to "compared to the"Also, was their evidence for winter upwelling in the Moon et al. (2018) study? Was the flux high enough to permit this? It's unclear as written but if this was the case I would suggest explicitly stating so.

We changed the sentence accordingly. We also added a sentence putting the iceberg metwater fluxes into context to other main sources. "The freshwater flux from these icebergs exceeds summer river runoff and reaches values of early summer (June-July) subglacial discharge (Moon et al., 2018), which may allow winter upwelling."

L60: change depth to "depths", change "deep terminus melt" to "basal icemelt"

We changed the sentence accordingly.

L63: I wonder why the authors are specifically mentioning cold-based glaciers here – especially as because later they emphasize that their results are extendable to warm or polythermal ice masses. Suggest deleting the specific mention of cold-based here and instead emphasize that this has been observed at several warm and polythermal ice masses in Svalbard.

We agree and removed the specific reference to cold-based glaciers.

L72: change "subglacial" to "submarine"?

We changed the sentence accordingly.

L75: change "to the summer situation" to "in the summer".

We changed the sentence accordingly.

L80: change "nutrient ion enrichment" to "nutrient andion enrichment"?

We changed the sentence accordingly.

L82-84: Sentence beginning with "Especially" is very unclearand seems a tad redundant (at least as written).

We shortened the sentence and merged it with the sentence before to avoid redundance and increase clarity: "We suggest that these nutrients can significantly increasing increase primary production in front of tidewater glaciers compared to similar fjords without these glaciers especially after nutrients supplied via winter mixing are used up (Leu et al., 2015)."

L88-90: Would suggest deleting this sentence and focusing instead of impact of warmer and longer meltseasons –no need to extend to glacier retreat which is really beyond the scope of the study –see comments above for abstract.

As suggested above, we prefer to keep the description of tidewater glacier retreat. A main message of our paper is also that the submarine discharge is somewhat decoupled from snowmelt, which makes the argument of earlier and increased discharge with climate change less relevant than the shallowing grounding line depth. We do have a rather detailed discussion on the effect of the glacier grounding line depth effect on plume dilution, which benefits from an introduction of retreating and shallowing tidewater glacier. Furthermore, it is not clear if the increased discharge can compensate the negative effects of a decreasing grounding line depth and a consequently decreasing plume dilution factor. Thus, we argue that tidewater glacier retreat to land is most important in the context of our study.

L93: delete "ice algae start growing" and also

We changed the sentence accordingly.

L94: delete" within sea ice"
We changed the sentence accordingly.

L97: change "subglacial upwelling" to "subglacially inducedupwelling)

We changed the sentence accordingly.

L101: extra space between "to" and "wind"

We removed the extra space.

L111-112: Re-write the last sentence: We suggest that even though subglacial upwelling is diminished in the spring, compared to the summer, in the absence of wind mixing, the enriched nutrient may enhance algal growth at this time of year.

We changed the sentence accordingly.

L114: The way this sentence is written begs the question –and what about the other glacier front.. why not describe that too?

We added the suggested details: "We used the natural conditions in a Svalbard fjord as a model system contrasting the biological response at two glacier fronts. Only one of the glacier fronts supplies submarine freshwater discharge during the winter/spring (early spring) transition period while a fast ice cover was present. In contrast, the other glacier front is mostly land-terminating."

L146: I have to admit to still being confused as to why additional water was added to the sea ice cores for melting. Why not just directly melt the core?

Melting ice cores in filtered seawater is a common and generally accepted approach in marine sea ice biology. If the ice core would melt directly, organisms living in the brine channels in Salinities commonly reaching 60PSU would be subject to very low bulk salinities (brine channel liquid + melted ice) mostly about 5 PSU. Consequently the organisms would experience osmotic shock and especially flagellates have been described to experience osmolysis, in addition to overall lower primary production

estimates due to osmotic stress. Hence, filtered seawater is added to reach salnities of the melted ice core of about 20 PSU (50% SW with 35 PSU + 50% sea ice of 5 PSU), which leads to less stress and survival of flagellates. We suggest that the reference that we already added gives detailed information about this approach and problem of direct melting of ice cores, which would be beyond the scope of this paper.

L247: Do you mean you transferred the phytoplankton communities into their respective environments? Sentence is a bit unclear.

We added following clarifications: "50 ml of the water containing the phytoplankton community communities of SG or IE were transferred into 50 ml sterile filtered (0.2 μm) seawater of SG or IE in 100 ml polyethylene bottles. The bottles with IE communities were then incubated under the ice at the IE station and the SG communities under the ice at the SG station."

L251-252: Last sentence is unclear. Is "adequate" the correct word here?
We replaced "adequate" with "respective".

L321: Change therefor to therefore

We corrected the typo.

L325: extra space after 4 μmol/L

We removed the extra space.

L345: It's unclear in this paragraph what site you are referring too? Is this for all the sites>

We added the specification that these calculations were done at the SG site.

L349-350: Is this estimate for inorganic nutrients valid in light of the non-conservative mixing discussed in the paragraph prior?

Inorganic nutrients behave conservatively in the seawater endmember (linear relationship with salinity), which allows us to use them for these mixing calculations. We added following sentence for clarification: "Inorganic nutrients behaved conservatively at the IE reference (Fig. 5a-c), which allows similar mixing calculation of the bottom water fraction."

L437: Add phrase "there was no" after "In addition"

We added the phrase as suggested.

L444: change to "below the freezing point"

We changed the sentence accordingly.

L452: change to glacier meltwater contributions

We changed the sentence accordingly.

L462: change to have enriched silicate concentrations

We changed the sentence accordingly.

L464: Note emphasis here on warm or polythermal-based glaciers –see comment above regarding line in introduction on cold-based glaciers

We changed the statement in the introduction as suggested above.

L467: change "last" to late

We changed the sentence accordingly.

L468: change "melting" to melt season which as remained throughout the winter. (delete extra period)

We changed the sentence accordingly.

L473: change "substantially" to much

We changed the sentence accordingly.

L475: change "productions" to production.

We changed the sentence accordingly.

L505: move phrase "in Billerfjorden" to the end of the sentence

We changed the sentence accordingly.

L515: make second part of the sentence beginning with "explaining the higher summer ..." a newsentence or re-write to make it follow from the first part of the sentence. As written, it doesn't.

We split the sentence in two: "In fact, the glacier terminus at SG was shallower (approx. 20 m) than any other studied tidewater glacier on Svalbard (70 m depth at Kronebreen, Halbach et al., 2019) or Greenland (> 100 m, Hopwood et al., 2020). Hence, the higher summer entrainment factors estimated in Kongsfjorden (3, Halbach et al., 2019) and Greenland (6 to 310, Hopwood et al., 2020) are not surprising."

L519: is "it" referring to submarine upwelling? Suggest clarifying.

Yes it is. We added the information.

L547: With regards to stratification observed at the SG site, would the physical mechanism of upwelling disturb this stratification?

Considering the strength of the stratification due to salinity changes from around 35 to 5 within 1-2 m, and the low plume dilution factor of 1.6, we would not expect the upwelling to be strong enough for substantial physical disturbance.

L604: Typo in "studiy"

We corrected the typo.

L607: Change "studies" to study periods

We changed the sentence accordingly.

L682: Change "Last" to Lastly

We changed the term accordingly.

L698: Change "most of" to much and "different evidence" to different lines of evidence

We changed the sentence accordingly.

L704: systems more similar to NG or IE? NG seems appropriate to refer to here since this is actually an adjacent land-terminating site correct?

We agree and changed the sentence accordingly.

L705: change to "but would result in higher biomass.."

We changed the sentence accordingly.

L706: Sentence beginning with "The pelagic .." is unclear and a fragment. Suggest re-writing.

We rewrote the sentence in the following way: "Considering the increased sedimentation rate at IE, we expect the pelagic/sympagic benthic coupling to become stronger"

L720: What is the depth of the effect of wind-induced vertical mixing?

The depth effect is highly dependent on the water column stratification (mostly salinity of the surface water) with strong seasonal and spatial variation. Thus, we cannot give any exact depth, but prefer to stay with this simplified statement of increased wind leading to increased mixing.

---

## Author Response (AR3)

Dear Evgeny Podolskiy,

We want to thank you sincerely for the last technical corrections and acceptance of our manuscript.

We did all corrections as suggested in the commented pdf file that you attached.

The term "submarine discharge" was suggested by reviewer 2, but we agree that in many places it can be misleading because it is unclear that this submarine discharge is coming from underneath the glacier. Thus, we changed the term to "subglacial discharge", whenever "submarine discharge" would be unclear (including the title).

Best regards

Tobias R Vonnahme